



# Quantifying the agricultural footprint on the silicon cycle: Insights from silicon isotopes and Ge/Si ratios

Sofía López-Urzúa[1], Louis Derry[1,2], Julien Bouchez[1]

[1]Université Paris Cité, Institut de physique du globe de Paris, CNRS, F-75005 Paris, France

[2]Department of Earth and Atmospheric Sciences, Cornell University, Ithaca, NY, USA

*Correspondence to*: Sofía López-Urzúa (slopezurzua@gmail.com)

**Abstract**. Silicon (Si) is essential for ecosystem function, supports primary productivity, and is intricately linked to the carbon cycle, which regulates Earth's climate. However, anthropogenic activities, such as agriculture, deforestation, and river damming, have disrupted the natural Si cycle, altering biogenic and dissolved Si fluxes in soils and rivers. Despite the

importance of understanding and quantifying human impacts on Si cycling at local and global scales, few studies address these disruptions, leaving a critical knowledge gap. Here, we analyzed the Si isotope composition ($\delta^{30}Si$) and germanium-silicon (Ge/Si) ratio dynamics across various Critical Zone compartments—soil, bedrock, water and plants—within the Kervidy-Naizin agricultural catchment observatory, France. Our findings reveal a vertical gradient in $\delta^{30}Si$ across the water pool in the Critical Zone, from lighter groundwater ($\delta^{30}Si = 0.56 \pm 0.25‰$) to heavier soil solutions ($\delta^{30}Si = 1.50 \pm 0.22‰$).

This gradient reflects distinct processes: in deep groundwater, weathering and clay precipitation control $\delta^{30}Si$ signatures, while at shallower depths, progressive plant uptake and crop removal further enrich $\delta^{30}Si$ in soil solutions. Using a mass balance combining $\delta^{30}Si$ and Ge/Si ratios, we quantified Si export from the catchment as plant material, both natural and harvested. Additionally, we assessed Si export from agricultural harvesting using two independent approaches: an elemental mass balance based on riverine chemistry and suspended sediments, and a method incorporating isotope fractionation factors

and soil Si loss indices. Plant material export, including natural and harvested material, emerged as the largest Si export flux from the catchment, accounting for ~74% of the Si solubilized from rock and exceeding dissolved Si export by 3.2 to 5.4 times. Through two independent approaches, we estimated that $37 \pm 10\%$ to $50 \pm 19\%$ of total Si export occurs through harvesting, depending on crop species, with the harvesting flux being 1 to 4 times greater than the dissolved Si flux. Reduction in dissolved Si exports because of agriculture may have significantly impacted downstream ecosystems, where Si

availability directly influences primary productivity. Our study highlights how human activities have reshaped the Si cycle in agricultural landscapes.

## 1 Introduction

Silicon (Si) is the second most abundant element in the Earth's crust (28%) and plays a crucial role in both geochemical and biological systems. In the geosphere, Si is a fundamental component of minerals, while in the biosphere, it serves as a vital



nutrient for many organisms. In its dissolved form as ortho-silicic acid ($H_4SiO_4$), Si supports both marine and terrestrial ecosystems. In the ocean, dissolved Si (DSi) is absorbed by diatoms to construct their siliceous frustules (Yool and Tyrrell, 2003). On land, Si provides physiological and structural benefits to vascular plants (Epstein, 1999; Guntzer et al., 2012; Ma et al., 2001). DSi is absorbed by roots from soil solutions, transported through the xylem via transpiration, and eventually precipitated as opal phytoliths, primarily in leaves but also distributed throughout other tissues (Hodson et al., 2005;

Strömberg et al., 2016; Trembath-Reichert et al., 2015).

    Anthropogenic activities, such as land use, deforestation, and river damming, have significantly altered the natural Si cycle, introducing new and complex feedback loops (Vandevenne et al., 2012). In coastal zones, the biogeochemical cycling of Si is highly sensitive to human pressure, particularly river damming and global temperature rise (Laruelle et al., 2009). Empirical evidence from deforestation experiments at the Hubbard Brook Experimental Forest supports these findings,

showing that land clearance can elevate dissolved Si levels, which is coupled with increased losses of amorphous silica (ASi) contained in phytoliths through erosion (Conley et al., 2008).

    The conversion of natural ecosystems to agricultural land and the subsequent harvesting of crops significantly alters the Si cycle, as land use influences Si mobilization and storage. For instance, in European temperate forests that have been cultivated for over 250 years, land use is identified as the most significant factor controlling Si mobilization, reducing

baseflow Si delivery by two- to threefold (Struyf et al., 2010). Similarly, Clymans et al. (2011) showed that forests exhibit approximately double the ASi levels compared to disturbed land, such as arable fields, pastures, and grazed forests. They estimated that human disturbance has caused a 10% decline in total ASi storage in soils since the onset of agricultural development around 3000 BCE.

    Terrestrial plants produce approximately 84 Tmol of biogenic Si (BSi) annually, with agricultural crops accounting for about

35% of this production due to their large biomass and high Si concentrations (Carey and Fulweiler, 2016). Crop plants such as rice, maize, wheat, barley, and sugarcane (all in the Poacea family) accumulate significant Si, exceeding concentrations of 10 mg Si $g^{-1}$ (of dry weight; Datnoff et al., 2001). Consequently, cultivation and harvest lead to the rapid depletion of the phytolith pool (Keller et al., 2012), and the export of straw can deplete the soil phytolith pool within a decade (Ma and Takahashi, 2002). Global agricultural activities are estimated to remove approximately 7.8 Tmol of Si per year from

landscapes (Matichenkov and Bocharnikova, 2001), a quantity almost equivalent to the 8.1 Tmol of dissolved Si transferred from continents to oceans via rivers and groundwater (Sutton et al., 2018). Over recent decades, the annual production of BSi by the top 10 global crops has tripled, adding 39 Gmol of Si each year. This increase, referred to as human-appropriated biogenic Si (HABSi), is projected to rise by 22–35% by 2050 due to population growth and land cover changes (Carey and Fulweiler, 2016). Therefore, understanding how land use influences the Si cycle—by locally increasing or decreasing the

dissolved Si flux in rivers —is crucial for predicting repercussions in phytoplankton proliferation, carbon cycling, and availability of Si for ecosystem.

    Technological advancements in the measurement of stable Si isotopes ($\delta^{30}Si$) and germanium-to-silicon (Ge/Si) ratios have significantly improved our understanding of the Si dynamics in terrestrial and aquatic ecosystem (Blecker et al., 2007;





Cornelis et al., 2010, 2011; Derry et al., 2005; Ziegler et al., 2005a). Si isotopes are now widely used as tracers in Critical
Zone research to quantify the effects of chemical weathering and nutrient cycling (Frings et al., 2021a, b; Guertin et al.,
2024). Silicon has three naturally occurring stable isotopes — $^{28}$Si (92.23%), $^{29}$Si (4.67%) and $^{30}$Si (3.10%) — whose
fractionation at low temperatures is mainly driven by water-rock interactions and biological processes. Significant Si
fractionation occurs during secondary mineral formation (Ziegler et al., 2005a, b), the adsorption of Si to Fe and Al oxides
(Delstanche et al., 2009; Oelze et al., 2014; Opfergelt et al., 2009), and the precipitation of amorphous silica (Fernandez et
al., 2019; Roerdink et al., 2015; Stamm et al., 2019). Vascular plants absorb Si through specialized root transporters, such as
Lsi1 and Lsi2 (Ma et al., 2007; Ma and Yamaji, 2006), with lighter Si isotopes preferentially incorporated into the plant
(Ding et al., 2005; Frick et al., 2020). A second fractionation step occurs during phytolith formation through silica
polymerization, where lighter isotopes are again preferentially incorporated. This selective uptake leads to further
fractionation within plant tissues, with the extent of fractionation varying among species, as demonstrated by studies on
banana (Opfergelt et al., 2006), rice (Ding et al., 2005), bamboo (Ding et al., 2008), or wheat (Frick et al., 2020; Ziegler et
al., 2005a).

Because of the similar directions of Si isotope fractionation during water-rock interactions and biological cycling, other
tracers are required to resolve potential ambiguity during identification of processes through which Si isotope fractionate.
Ge/Si ratios can serve as such an additional tracer (Derry et al., 2005; Fernandez et al., 2022; Kurtz et al., 2002; Scribner et
al., 2006). During secondary mineral formation, Ge is preferentially incorporated relative to Si, resulting in high Ge/Si ratios
in the mineral phase (Kurtz et al., 2002; Murnane and Stallard, 1990). Additionally, Ge partitioning is influenced by its
adsorption onto hydrous iron oxides and Al oxyhydroxides (Anders et al., 2003; Pokrovsky et al., 2006). Ge/Si ratios have
been also used to trace biological processes, with studies suggesting that Ge is discriminated against Si during vascular plant
uptake (Blecker et al., 2007; Delvigne et al., 2009; Derry et al., 2005; Lugolobi et al., 2010; Meek et al., 2016).

Hence, combining measurements of Ge/Si ratios and $\delta^{30}$Si offers a powerful tool for investigating the impact of agriculture
on the Si cycle. Studies using these tracers have reached significant findings: Vandevenne et al. (2015) documented a
depletion of light Si isotopes in soil water from intensive croplands and managed grasslands ($\delta^{30}$Si = 1.3 - 1.8‰) compared
to native forests ($\delta^{30}$Si = -0.1 - 1.4‰). In the Kaveri River basin (India), Sarath et al. (2022) attributed an enrichment of
approximately +1.06‰ in the $\delta^{30}$Si river signature to uptake by plants and the return of Si-depleted flow from irrigated
agriculture. In the Amazon, the conversion of forest to cropland was found to increase water percolation in the soil, leading
to enhanced weathering of deeper soil layers and a greater contribution of mineral weathering to stream waters (Ameijeiras-
Mariño et al., 2018).

Despite the significance of these findings, quantitative studies on the impact and quantification of land use on the Si cycle
remain scarce, and there is still a substantial gap in our understanding and quantification on how land use, particularly
agriculture, affects Si cycling on a local and global scale. This study aims to address this gap by investigating Si isotopes and
Ge/Si ratios in a small agricultural catchment in Brittany, France. We aim to quantify the fraction of silica removed from the
catchment through crop harvesting. By integrating Si concentration, Ge/Si ratio, and $\delta^{30}$Si data from different compartments



of the Critical Zone, we employ an elemental and isotopic mass balance model to determine the fractions of Si exported through soil erosion, chemical weathering as dissolved Si, and agricultural harvesting. This study offers critical insight into the broader implications of land use on the Si cycle, enhancing our understanding of how these processes influence Si availability, ecosystem health, and global biogeochemical cycles.

## 2 Study site and sampling

### 2.1 The agricultural Kervidy-Naizin catchment

Located in Brittany, France, the study site encompasses the 4.8 km² headwater Kervidy-Naizin catchment, which is part of the ORE ArgHys observatory (Fovet et al., 2018). The ORE ArgHys observatory is integrated into the National Research Infrastructure (RI) of the French Critical Zone initiative, OZCAR-RI (Observatoires de la Zone Critique–Application et Recherche; Gaillardet et al., 2018). This catchment is characterized by intensive agriculture, including mixed cropping systems primarily consisting of maize and wheat, as well as indoor dairy and pig farming.

The catchment is situated on fissured and fractured Brioverian schist (530 Ma), comprising a monotonous and rhythmically bedded succession of dominant siltstones and sandstones (Thomas et al., 2009, Pellerin et al., 1998, Van Vliet-Lanoe et al., 1998). The schist is primarily composed of quartz, muscovite, chlorite, and, to a lesser extent, K-feldspar and plagioclase. Pyrite is the main accessory mineral below a depth of 7 meters, with secondary solid phases including illite, smectite, and Fe(II) hydroxides (Pauwels et al., 1998). Regional catchment-scale (829 km²) denudation rates, estimated through measurements of beryllium-10 ($^{10}$Be), are $10 \pm 1.7$ m Ma$^{-1}$ (Evel catchment; Malcles et al., 2023).

An unconsolidated layer of weathered material, up to 30 meters thick, overlays the bedrock and supports a shallow aquifer. Three pools of groundwater are identified in the catchment (Dia et al., 2000; Molenat et al., 2008). Deep Groundwater (DGW), flowing through the fissured schist below the 30 meters of unconsolidated and weathered material, is characterized by low nitrate and dissolved organic carbon (DOC), eventually emerging near the stream. Shallow Groundwater (SGW), hosted in the unconsolidated material and weathered schist, occupies depths between 0 and 30 meters and exhibits high nitrate concentrations. Finally, Wetland Groundwater (WGW), also hosted in the in the weathered schist and the unconsolidated layer but closer to the stream and wetland is distinguished by its high DOC concentrations.

Soils in the study area can be categorized into three distinct systems following the classification by Water and Curmi (1998): i) well-drained soils, ii) intermediate soils and iii) hydromorphic soils (Fig. 1). The upland regions are characterized by well-drained soils (Brunisols and luvisolic Brunisols) typically exhibiting depths of 60 to 80 cm. There, the water table usually remains a few meters below the surface. Downslope, the water level rises to just a few centimetres below the surface, producing intermediate soils. In the riparian zone, soils are highly hydromorphic, and the water table usually reaches the surface during the wet season, resulting in the development of redoxisols and reductisols. These hydromorphic soils are characterized by a depletion of Fe-oxyhydroxides, high organic matter content, and hydrolytic degradation of clays (Duchaufour, 1982).



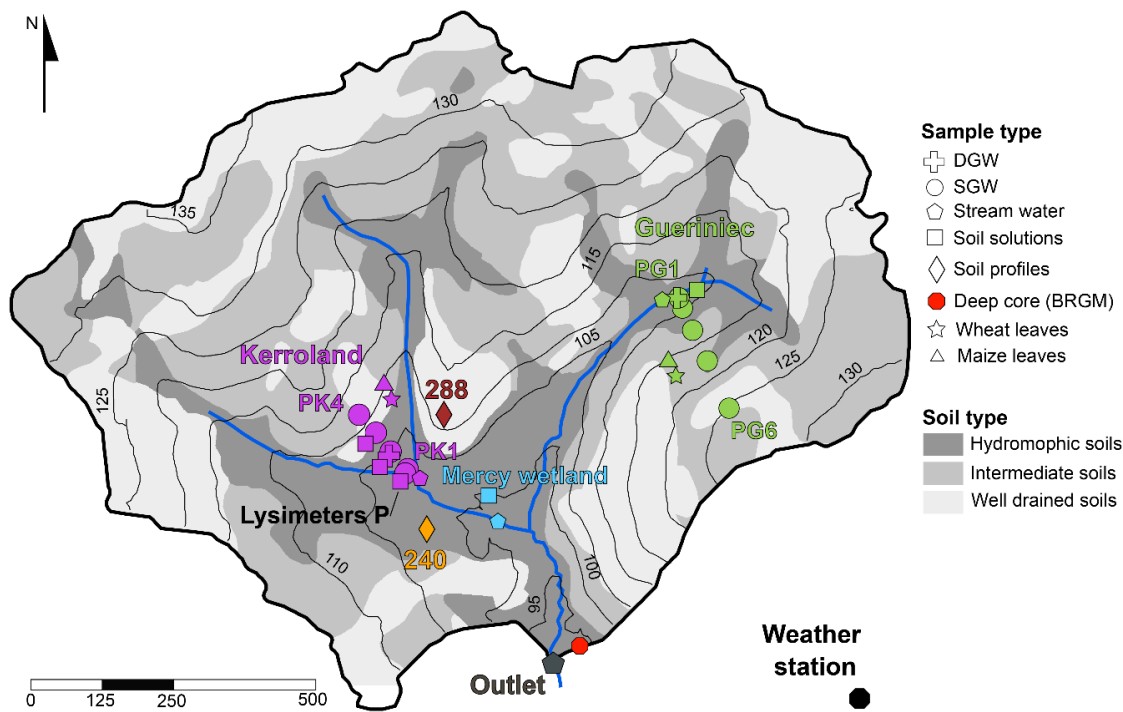

**Figure 1. Soil map of the Kerdivy-Naizin catchment showing sampling site locations. Symbols denote sample types, including deep groundwater (DGW), shallow groundwater (SGW), stream water, soil solutions, soil profiles, deep core "BRGM" (in red), maize leaves, and wheat leaves. Sample colours indicate specific locations: purple (Kerroland), green (Gueriniec), cyan (Mercy Wetland), and gray (outlet). Soil profiles are color-coded: orange for hydromorphic soils and burgundy for well-drained soils. Black lines represent elevation contours (in meters), and blue lines show the stream network.**

Pig farming, followed by dairy farming, is the most widespread agricultural operation in the basin (Akkal-Corfini et al., 2014). The crops produced in the basin are intended for bovine and porcine animal feed. In 2020, the most abundant crops, representing 70% of the agricultural area, were maize (35.8%), cereals (wheat and barley; 22.6%), and pastures grasses (11.9%). Generally, bovine manure is used on site to fertilize the fields. However, pig slurry and poultry manure are partly exported off the farms. The management of crop residues depends on each type. During the harvest of grain maize, the stalks and roots are returned to the soil. In the case of silage maize and cereal straw, the entire above-ground part of the plant is exported out of the basin, and only the roots are left in place.

### 2.2 Sampling

Sampling was conducted over a period of nine years from different compartments of the Critical Zone, including soil solutions, groundwater, stream water, bedrock, soil, and plants. The catchment is equipped with piezometers and lysimeters along two transects (Fig. 1 and Fig. A1 in Appendix A). The "Gueriniec transect" (PG), situated in the eastern part of the catchment is equipped with six piezometers and one lysimeter at 50 cm depth. The "Kerroland transect" (PK), located in the southwest part and at a lower elevation than PG, is equipped with six piezometers and six lysimeters for sampling soil



solution at different depths from 10 to 100 cm. The Mercy wetland is equipped with a lysimeter sampling pore water at depths of 10 cm and 50 cm depth. Stream water was collected from four locations: at the outlet ("Riv Ex"), near the Mercy wetland ("Riv M"), and at the foot of each transect ("Riv K" and "Riv G"). Additionally, in 2023 a new set of lysimeters, named "P", was temporarily installed at the bottom of the Kerroland transect near the stream. Depending on the field campaign, soil solutions, groundwater, and stream samples were collected to cover different hydrological periods.

Regolith and soil samples were taken from two soil profiles, named "240" and "288" (Fig. 1). The profile 240 was excavated
to a depth of 100 cm in hydromorphic soils, while the profile 288, located in the well-drained soils, reached a depth of 90 cm. Four samples, each corresponding to a soil horizon, were collected in each profile. It was not possible to access the drill cuttings form the installation of the piezometers in the catchment. As a proxy for local bedrock composition, a fresh schist fragment found in the vicinity of the Gueriniec transect was collected and analyzed. Additional data on bedrock chemistry reported in Dia et al. (2000) was used in this study. The data includes major and trace element concentrations from drill
cuttings of a deep core within the catchment, named "BRGM", which intercepted fresh schist at a depth of 28 m.

Finally, maize and wheat leaves were collected near both transects, with one sample of each species taken from each transect.

## 3 Methods

### 3.1 Elemental analyses

The chemical composition of soil and regolith samples was determined by X-ray fluorescence spectrometry using an X Epsilon 3XL Panalytical spectometer equipped with a silver anticathode, with uncertainties for Al, Si, Ti, and Fe concentrations ≈ 1%. The mineralogy of soil and bedrock samples was determined through X-Ray Diffraction (XRD) using an Empyrean Panalytical diffractometer equipped with a copper anticathode (wavelength Cu kα1 = 1,54 Å; 45 keV) at the Chemistry Department of Université Paris Cité. Quantitative mineral analysis was conducted using Maud software
(Lutteroti, 2000). Organic matter content in soil samples was determined using the dry combustion method, which converts all carbon present in the sample into carbon dioxide. The sample is heated to approximately 1000°C in the presence of oxygen, and after chromatographic separation, the amount of carbon dioxide produced is quantified using a catharometer (thermal conductivity detector). The organic matter content is then calculated by multiplying the organic carbon by 1.72, based on the assumption that soil organic matter contains 58% carbon (Pribyl, 2010). The measurements were conducted at
the Laboratoire d'Analyses des sols, INRAE, France.

Water samples were analyzed for major cations, anions, dissolved silica, dissolved organic carbon (DOC), and trace metals. Anion ($Cl^-$, $NO_3^-$, $SO4^{2-}$) concentrations were analyzed by ion chromatography (DIONEX DX 100) with uncertainties ≈ 1%. Major cations, trace metals, and total dissolved Si concentrations were measured using an Agilent 7900 Inductively Coupled Plasma Mass Spectrometer (ICP-MS); spectral interferences on mass 28 for Si were mitigated by the use of an integrated
collision cell with He. DOC concentrations were determined with a Shimadzu TOC-VCSH Total Organic Carbon analyzer.



All the measurements of dissolved species were performed at the PARI platform of the Institut de Physique du Globe de Paris (IPGP).

## 3.2 Extraction of oxide phases

We measured the presence of iron, aluminum (Al), and Si in both amorphous and crystalline oxide forms to assess the absorption of Si into oxide minerals, as this process induces Si isotopic fractionation. These extractions were carried out following two specific protocols, at the Laboratoire d'Analyses des Sols, INRAE. First, we employed an extraction with oxalic acid and ammonium oxalate solution buffered to a pH of 3 to assess amorphous Al-Si phases (Tamm, 1922). Approximately 1.25 g of soil ground to 250 µm was agitated for 4 hours in the presence of 50 ml of reagent, at 20°C in the dark. The second method involved extraction at elevated temperatures with sodium citrate as a complexing agent, sodium

bicarbonate to buffer the pH, and sodium dithionite as a reducing agent (Mehra & Jackson, 1960). According to Jeanroy (1983), this method allows for extraction of all the iron present as oxides and oxyhydroxides and less than 5% of the iron present in ferriferous silicate minerals. Approximately 0.5 g of soil ground to 250 µm was mixed with 25 ml of the extraction solution. After adding 1.5 ml of the reducing solution, the mixture was heated to 80°C in a water bath for 30 minutes with intermittent agitation. After cooling, the volume was adjusted to 50 ml, homogenized, and filtered. For both extraction

methods, the quantities of Al, Si, and Fe in the filtered extracts were measured by Inductively Coupled Plasma Atomic Emission Spectrometry (ICP-AES) at the Laboratoire des Analyses de Sol, INRAE.

## 3.3 Digestion

Before digestion, separation of the clay fraction for two samples was achieved through a centrifugation method, which followed the procedure outlined by the US Geological Survey (USGS) based on Stokes' Law (U.S. Geological Survey,

200  2001).

To digest clays, soil, rocks, and leaf samples, we followed an alkaline fusion method adapted from Georg et al. (2006). Prior to alkaline fusion, soil and leaf samples were ashed to eliminate organic matter. The samples were heated in ceramic crucibles at 200°C for 2 hours, followed by an additional heating for 4 hours at 600°C.

For each sample we weighed between 5 mg and 10 mg of powdered material in Ag crucibles and added NaOH pellets

(analytical grade, Merck) while maintaining a < 1:10 sample/NaOH weight ratio. The crucibles were heated at 720 °C for 12 minutes in a muffle furnace, and left to cool down to 300 °C. Subsequently, the content of the crucibles was transferred into a Teflon vial with 15 mL of MilliQ® water, closed and left for 24 hours at room temperature. The solution was then transferred into LDPE bottles, and the Teflon vial containing the crucible was filled with 10 mL of MilliQ® water and sonicated for 15 minutes before transferring the solution into the same bottle. This step was repeated three times to ensure

complete recovery from the crucible. Finally, the sample was diluted to a Si concentration of less than 30 ppm (to avoid precipitation of silica) and adjusted to a matrix of 0.2N HNO$_3$, ready for Si purification. For clay fractions, elemental





chemistry was determined after alkaline fusion using an Agilent 7900 Inductively Coupled Plasma Mass Spectrometer (ICP-MS).

### 3.4 Si isotopes

We employed a Si purification column chemistry protocol described in detail in López-Urzúa et al. (2024). Prior to Si purification, samples with a [DOC]/[Si] >1 were subject to UV photolysis to decompose DOC and prevent matrix effects during Si isotope measurements (López-Urzúa et al., 2024).

The Si isotope composition was measured with a Thermo Finnigan Neptune Plus MC-ICP-MS (IPGP, France). Measurements were performed in 'wet' plasma mode using a Stable Introduction System (SIS) composed of a quartz spray 220 chamber, with an uptake rate of ~ 100 µL/min. Isotopes $^{28}Si$, $^{29}Si$ and $^{30}Si$ were simultaneously collected in three Faraday cups (L4, L1 and H2, respectively) each equipped with $10^{11}$ Ω amplifiers. The 'medium' resolution entrance slit (M/ΔM 5-95% peak height ≥ ~4000) was employed to effectively separate polyatomic interferences. The samples were analyzed at a Si concentration of 1 ppm, with typical signal intensities for $^{28}Si^+$ ranging from 9 to 11 V. Si isotopic compositions are reported in the standard per mil notation. $δ^{30}Si$ (‰) and $δ^{29}Si$ (‰) were calculated by standard-sample bracketing using NBS-28 as 225 the bracketing standard. One measurement consisted of 25 cycles of 4.194 s integration time. Each sample was measured multiple times and the reported delta values correspond to the average of the n consecutive measurements (n = 4 to 8). Uncertainties on the $δ^{30}Si$ values are reported as the 95% confidence interval (CI 95%) using Student's law.

The three-isotope plot of Si shows that all the data obtained in the present study plot along the mass-dependent fractionation line (refer to Fig S.1 in the supplement). The long-term accuracy and precision were checked with routine measurements of 230 the certified USGS basalt standard (BHVO-2), and the Diatomite and Big Batch standards during the different analytical sessions. The $δ^{30}Si$ obtained on BHVO-2 (-0.29 ± 0.12‰, 2 SD, n = 237), Diatomite (1.30 ± 0.17‰, 2 SD, n = 155) and Big Batch (-10.59 ± 0.20‰, 2 SD, n = 69) agree with previously published values (BHVO-2: Delvigne et al., 2021; Jochum et al., 2005; Big Batch and Diatomite: Reynolds et al., 2007).

### 3.5 Ge/Si ratios

Ge concentrations were measured using isotope-dilution hydride generation ICP-MS. Samples were prepared by mixing approximately 10 ml of sample with an enriched solution of $^{70}Ge$ spike ($^{70}Ge/^{74}Ge$ = 161) to target a $^{70}Ge/^{74}Ge$ ratio near 9.5. Spiked samples were left to sit at room temperature for at least 24 h to allow sufficient equilibration of the spike with the sample. Germanium was analyzed using a CETAC HGX200 hydride generation system coupled online to a HR-ICP-MS Element II (ThermoScientific) at IPGP. Quantification was performed by isotope dilution using the $^{70}Ge/^{74}Ge$ ratio with 240 corrections for mass bias and signal drift from sample-standard bracketing. At the same time, response curves were established by analysis of Ge standards at 2, 5, 10, 20, 50, 100, and 200 ng L$^{-1}$. Results from isotope dilution calculations were cross-checked against the standard response curves measured at m/z=74. Ge/Si ratios for the reference materials NBS-28 and Diatomite are consistent with published values. The Ge/Si ratio obtained for Diatomite (0.70 ± 0.02 µmol mol$^{-1}$ n = 3)





and NBS-28 (0.36 ± 0.01 µmol mol⁻¹, n = 3) agree with values reported by Frings et al. (2021b) (0.66 ± 0.02 µmol mol⁻¹ and
0.38 ± 0.05 µmol mol⁻¹, respectively).

## 4. Results

A summary of Si concentration, δ³⁰Si and Ge/Si for bedrock, bulk soil, clay fraction plants and water samples are given in
Table 1 and Fig. 2. Detailed data for individual samples, including water, soil, bedrock, and plants, on are available at
https://doi.org/10.5281/zenodo.14615156 (López-Urzúa et al., 2025).

To ascertain the provenance of the fresh schist fragment, we compared the insoluble elemental ratio Ti/Al in our sample with
those reported for the deep BRGM core (Dia et al., 2000). The Ti/Al ratio for the deep schist (0.044) from the core and the
rock fragment (0.053) are similar, indicating that the sampled fragment may indeed be representative of the fresh bedrock.
The schist bedrock displays a δ³⁰Si value of -0.13 ± 0.05‰ (Table 1), and a Ge/Si ratio of 1.33 µmol mol⁻¹, consistent with
abundant quartz (66%, with Ge/Si ≤ 1 µmol mol⁻¹) and muscovite (22% with Ge/Si= 2.5 to 3.0 µmol mol⁻¹) (Aguirre et al.,
2017; Evans and Derry, 2002).

**Table 1. Summary of Si concentration, δ³⁰Si, Ge/Si, and Si/Al ratios for the different Critical Zone compartments of the Kervidy-Naizin catchment.**

| | Si (µmol L⁻¹) | | δ³⁰Si (‰) | | Ge/Si (µmol mol⁻¹) | | Si/Al (molar ratio) | |
| --- | --- | --- | --- | --- | --- | --- | --- | --- |
| | Mean ± SD | n | Mean ± SD | n | Mean ± SD | n | Mean ± SD | n |
| **Bedrock** | 81.7% [a] | 1 | -0.13 ± 0.05 | 1 | 1.33 | 1 | 7.54 | 1 |
| **Bulk soil** | 72.6 ± 4.4% [a] | 8 | -0.37 ± 0.05 | 8 | 1.80 ± 0.19 | 8 | 5.53 ± 1.1 | 8 |
| **Clay fraction** | 35.9 ± 6.1% [a] | 2 | -0.75 ± 0.05 | 2 | 4.08 ± 1.09 | 2 | 1.32 ± 0.19 | 2 |
| **Secondary clay** [b] | | | -1.30 ± 0.01 | 2 | 6.64 ± 2.43 | 2 | 0.67 ± 0.05 | 2 |
| **Plants** | | | 0.16 ± 0.5 | 4 | 2.48 ± 1.11 | 4 | - | |
| **Wheat** | | 2 | 0.51 ± 0.46 | 2 | 3.27 ± 0.86 | 2 | 45.1 ± 5.08 | 2 |
| **Maize** | | 2 | -0.21 ± 0.29 | 2 | 1.61 ± 0.42 | 2 | 7.83 ± 2.27 | 2 |
| **Stream water outlet** | 158 ± 15 | 8 | 1.12 ± 0.13 | 8 | 0.33 ± 0.17 | 5 | 323 ± 124 | 8 |
| **Soil solutions** | 201 ± 35 | 25 | 1.50 ± 0.22 | 25 | 0.66 ± 0.19 | 16 | 212 ± 299 | 25 |
| **Shallow groundwater** | | | | | | | | |
| **Kerroland** | 194 ± 40 | 27 | 1.05 ± 0.14 | 27 | 0.65 ± 0.19 | 15 | 862 ± 920 | 27 |
| **Gueriniec** | 161 ± 73 | 16 | 0.72 ± 0.18 | 16 | 0.88 ± 0.20 | 3 | 640 ± 1757 | 16 |
| **Deep groundwater** | | | | | | | | |
| **Kerroland** | 448 ± 14 | 5 | 0.74 ± 0.08 | 5 | 0.88 ± 0.42 | 3 | 10018 ± 2881 | 5 |
| **Gueriniec** | 445 ± 11 | 3 | 0.28 ± 0.01 | 3 | 0.19 | 1 | 6671 ± 5354 | 3 |

[a] SiO₂ (%), [b] clay fraction corrected by the contents of primary minerals (muscovite, microcline and quartz), refer to Appendix B.

The bulk soil samples displayed consistent values across both soil profiles, δ³⁰Si values are consistently low (-0.37 ±
0.05‰), while high Ge/Si ratios are relatively high (1.80 ± 0.19 µmol mol⁻¹) compared to the bedrock (Table 1), both
signatures being consistent with weathering and formation of secondary clays.





The clay fractions show even lighter δ³⁰Si values and higher Ge/Si ratios than the bulk soil, with mean δ³⁰Si values of -0.75 ±

0.07‰ and Ge/Si ratios of 4.08 ± 1.09 µmol mol⁻¹ in both soil profiles (well-drained and hydromorphic). As the bedrock

contains muscovite, it is likely that the separated clay fraction includes primary minerals in addition to secondary clays. The

high K/Al ratio (0.14 - 0.15, Table B) in the clay fraction indicates the presence of muscovite (K/Al = 0.33) alongside with

secondary minerals.

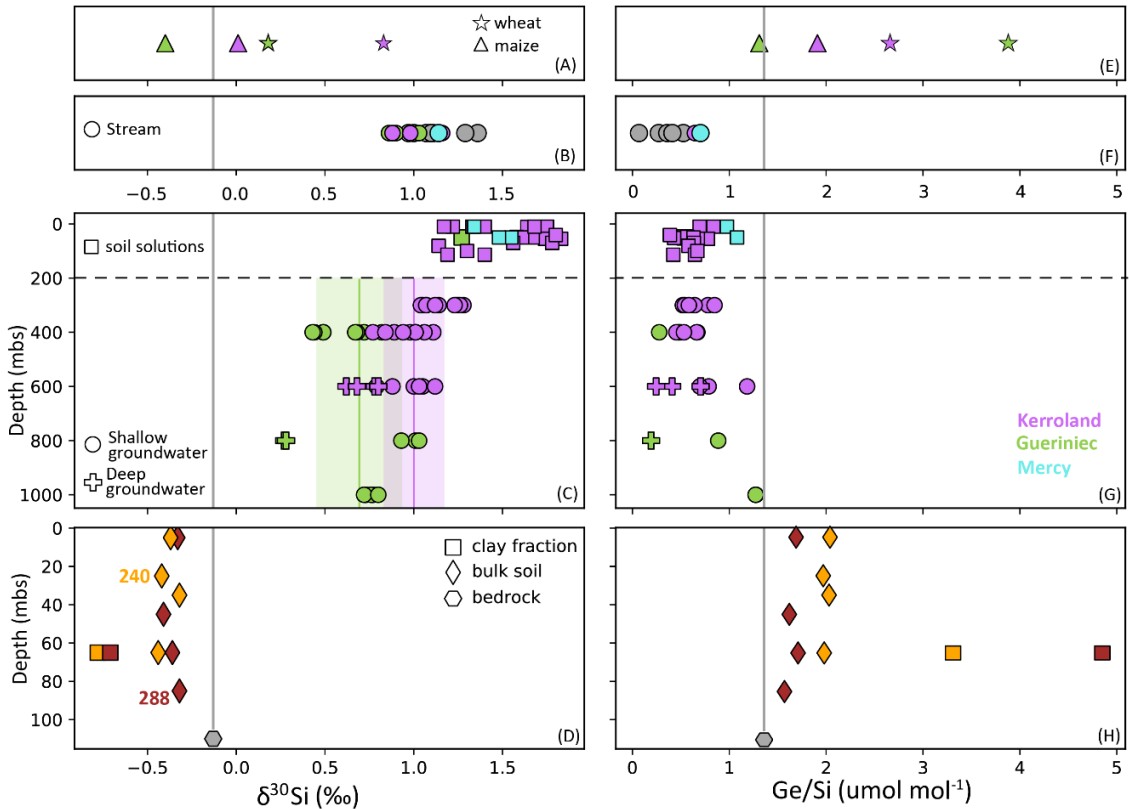

**Figure 2.** Data of δ³⁰Si and Ge/Si ratios across different Critical Zone compartments of the Kervidy-Naizin catchment. Sampling
sites are color-coded as follows: Kerroland (purple), Gueriniec (green), Mercy wetland (cyan), and the stream outlet (grey).

Samples from soil profiles 288 and 240 are highlighted in burgundy and yellow, respectively. Panels A and E present data from
plant leaves, panels B and F from stream water, panels C and G from soil solutions and groundwater, and panels D and H for rock
and regolith samples. In Panel C, vertical lines and shaded areas in purple and green denote the mean and one standard deviation
(1 SD) of groundwater data for each transect.

To correct the values obtained for the clay fraction and isolate the δ³⁰Si and Ge/Si signatures of secondary minerals

(kaolinite), we used a simple mixing model (Appendix B). This model assumes that the mineralogy of the clay fraction is

composed of quartz ($SiO_2$), kaolinite ($Al_2Si_2O_5(OH)_4$), muscovite ($KAl_2(AlSi_3O_{10})(OH)_2$) and microcline ($KAlSi_3O_8$) (see

details Appendix B). After correction, the values for secondary minerals show a mean δ³⁰Si values of -1.30 ± 0.01‰ and a

Ge/Si ratios of 6.64 ± 2.43 µmol mol⁻¹, which are comparable to those reported for clays in other catchments, such as




tropical granitic (4.8 - 6.1 µmol mol$^{-1}$) and temperate humid granitic (6.1 - 6.3 µmol mol$^{-1}$) environments (Cornelis et al., 2010; Lugolobi et al., 2010).

Water samples exhibit a broad range of Si concentrations (95 to 465 µmol L$^{-1}$), with the highest concentrations found in deep groundwater (429 to 464 µmol L$^{-1}$). In contrast, shallow groundwater, stream, and soil solutions display similar average Si concentrations, which fall between 161 to 201 µmol L$^{-1}$, without clear differentiation among these types of water.

The $\delta^{30}$Si signatures in water samples exhibit significant variability, ranging from 0.27 to 1.83‰. Two notable trends in $\delta^{30}$Si signatures emerge. First, groundwater from the Gueriniec transect consistently exhibits lighter $\delta^{30}$Si values, with a mean of 0.64‰, compare to the Kerroland transect, which has heavier signatures averaging 0.99‰. Second, a vertical gradient in $\delta^{30}$Si is observed across both groundwater and soil solutions (Fig. 2). $\delta^{30}$Si values progressively increase from deeper to shallower piezometers, reaching their highest values in soil solutions. Deep groundwater (samples PG1 and PK6) display the lightest $\delta^{30}$Si values, with a mean of 0.56‰, while shallow groundwater averages 0.92‰. Soil solutions exhibit the heaviest $\delta^{30}$Si values, averaging 1.50‰. These depth gradients are consistently observed within a given location when comparing nested piezometers and soil solutions (Appendix C; Fig. C1).

The Ge/Si ratios in water samples exhibit a broad range, from 0.07 to 1.27 µmol mol$^{-1}$. However, unlike $\delta^{30}$Si, no clear vertical gradient is observed. The highest Ge/Si values are found in groundwater from PG5 (mean: 1.27 µmol mol$^{-1}$) and soil solutions from the Mercy wetland (mean: 1.02 µmol mol$^{-1}$). Stream samples at the outlet exhibit the lowest Ge/Si ratios, with a mean of 0.33 ± 0.17 µmol mol$^{-1}$.

Plant samples reveal contrasting patterns in $\delta^{30}$Si signatures and Ge/Si ratios between maize and wheat leaves across the two transects. Maize leaves from both Kerroland and Gueriniec transects exhibit lighter $\delta^{30}$Si values and lower Ge/Si ratios compared to wheat leaves (Fig. 2, Table 1). A consistent difference in $\delta^{30}$Si signatures between the transects is observed, with leaves from Kerroland—regardless of the plant species—showing heavier $\delta^{30}$Si values compared to those from Gueriniec.

# 5 Discussion

## 5.1 Controls on $\delta^{30}$Si and Ge/Si dynamics in the Kervidy-Naizin catchment

The three main processes affecting $\delta^{30}$Si and Ge/Si signatures in the Critical Zone are mineral dissolution and/or neoformation of clays (Froelich et al., 1992; Kurtz et al., 2002; Ziegler et al., 2005a, b); Si incorporation into Fe and Al oxyhydroxides (Anders et al., 2003; Delstanche et al., 2009; Oelze et al., 2014; Opfergelt et al., 2009; Pokrovsky et al., 2006); and plant uptake and phytolith formation (Blecker et al., 2007; Delvigne et al., 2009; Derry et al., 2005; Ding et al., 2005; Frick et al., 2020; Ma et al., 2007; Ma and Yamaji, 2006; Meek et al., 2016; Opfergelt et al., 2006). In the following subsections, we investigate the influence of these three processes on Si dynamics within the Kervidy-Naizin catchment.





### 5.1.1 Mineral dissolution and neoformation of clays in groundwater

The combined analysis of $\delta^{30}Si$, Ge/Si and Si/Al ratios enables the tracking of clay formation and dissolution processes (Fig. 3). Assuming congruent dissolution of the bedrock, the initial fluid after release by primary mineral dissolution should reflect the Si/Al ratio (7.54), $\delta^{30}Si$ (-0.13 ± 0.05‰), and a Ge/Si ratio (1.33 µmol mol$^{-1}$) characteristic of the rock. However, groundwaters consistently exhibits higher Si/Al and $\delta^{30}Si$ values, along with lower Ge/Si ratios, compared to the presumed initial compositions (Fig. 3 and Table 1), reflecting fractionation driven by clay mineral formation.

Deep groundwater shows the highest Si/Al ratio (mean = 8762) consistent with clay precipitation. However, samples from the shallow groundwater exhibit lower Si/Al and higher $\delta^{30}Si$ values, suggesting the influence of additional processes, such as plant uptake or precipitation of iron oxyhydroxides. The observed decrease in Si/Al in the SGW is attributed to a sharp reduction in Si concentration, dropping from a mean value of 446 µmol L$^{-1}$ in DGW to 181 µmol L$^{-1}$ in SGW, while Al concentrations exhibit only minor variations. The higher Si concentration in the DGW may be due to its transit through

plagioclase- and primary mineral-rich schist, which are prone to weathering.

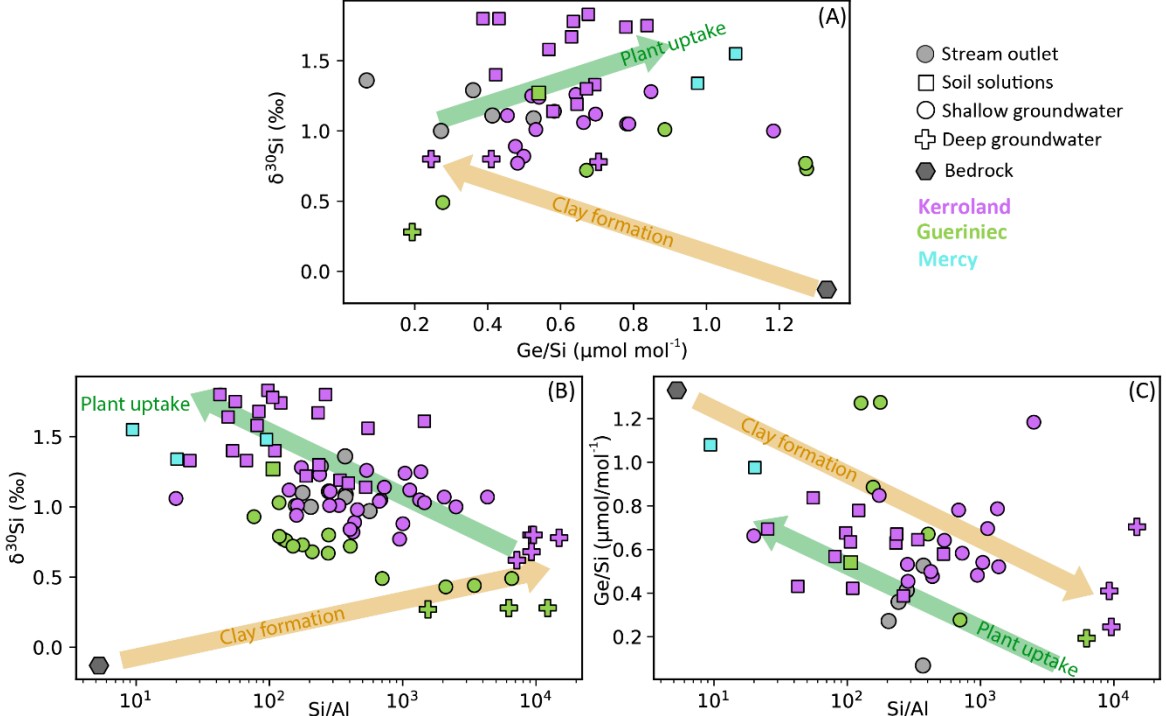

**Figure 3. Cross plots of dissolved (A) $\delta^{30}Si$ and Ge/Si (B) $\delta^{30}Si$ and Si/Al ratios (C) Ge/Si and Si/Al ratios. Coloured arrows illustrate the presumed trends for clay formation and plant uptake in the water compartments. The slopes of the arrows represent general trends and are not meant to indicate precise relationships.**

The lower Si concentration in the SGW may result from the depletion of easily weatherable minerals in the upper horizons. Saturation index calculations show oversaturation of kaolinite and muscovite, and near-equilibrium for quartz, pointing to the absence of easily weatherable minerals in these shallow horizons. A potential source of Si in SGW could be the





dissolution of clay minerals facilitated by organic ligands (Berggren and Mulder, 1995; Cama and Ganor, 2006; Drever and Stillings, 1997; Wieland and Stumm, 1992). However, given the relatively low DOC content in SGW and DGW (0–6.2 ppm) this process is unlikely. Additionally, clay dissolution would likely decrease $\delta^{30}Si$ in SGW, which contradicts the observed increase (Fig. 3).

### 5.1.2 Si adsorption onto Fe and Al oxyhydroxides in the soil

In the Kervidy–Naizin catchment, fluctuations in water levels lead to a series of redox processes in the soils of the riparian area (Walter and Curmi, 1998). During the dry season, oxidizing conditions promote the precipitation of iron oxides, while the wet season reducing conditions lead to the release of Fe. During Si adsorption onto Fe and Al oxyhydroxides, lighter isotopes are preferentially removed, resulting in an increase in the $\delta^{30}Si$ in the solution (Delstanche et al., 2009; Oelze et al., 2014; Opfergelt et al., 2009). A similar pattern is observed for Ge/Si ratios, with previous studies reporting the preferential adsorption of Ge onto Al and Fe oxyhydroxides (Anders et al., 2003; Pokrovsky et al., 2006).

If redox processes explained the $\delta^{30}Si$ and Ge/Si dynamics in soil solutions, a correlation between Fe and the tracers would be expected. However, for the full dataset of soil solutions ($n = 25$), no significant correlation between $\delta^{30}Si$ values and Fe concentrations ($R^2=0.002$) is observed (Appendix C; Fig. C2). In contrast, when analyzing a subset of lysimeters P ($n = 4$) collected simultaneously on the same day, a negative correlation emerges between $\delta^{30}Si$ values and Fe concentrations ($R^2=0.85$, after excluding one outlier with the highest $\delta^{30}Si$ value). This observation suggests the dissolution of iron oxides and the subsequent release of adsorbed Si may be occurring at this site. Similarly, for the full dataset of soil solutions, Ge/Si ratios show a positively correlated with Fe concentrations ($R^2 = 0.72$), indicating that Ge concentrations in the soil solutions may be influenced by adsorption/desorption onto Fe-oxyhydroxides, followed by subsequent dissolution (Appendix C; Fig. C2).

However, the contribution of a lighter $\delta^{30}Si$ pool from poorly crystalline silica and iron oxides is expected to be negligible, as our leaching experiments indicate that these pools account for less than 0.6% of total Si (Table 2; López-Urzúa et al., 2025). Similarly, the contribution of a high-Ge oxide pool is expected to be minimal, as observed by Scribner et al. (2006) in their study along a climate gradient in Hawaii. The authors concluded that Al-Si phases are the primary host for soil Ge, rather than Fe oxyhydroxides, which do not significantly impact the Ge/Si mass balance.

### 5.1.3 Plant uptake and removal of crops

The heavy $\delta^{30}Si$ signatures observed in our soil solutions as well as in SGW may result from the prolonged uptake of lighter Si isotopes by plants, followed by their removal through harvesting, as in agricultural catchments. This biomass removal prevents the recycling of the phytoliths, unlike in unaltered systems (Derry et al., 2005). In the Kervidy-Naizin catchment, water levels are generally shallow (30–100 cm depth), especially near the riparian zone (PK1-2-5-6 and PG1-2-3) during the growing season, allowing plant root access to groundwater (water level database: https://agrhys.fr/BVE/vidae/). The vertical gradient for $\delta^{30}Si$ values from DGW to SGW to soil solutions suggests that plants have direct access to groundwater, leading





to *in situ* fractionation of Si isotopes. Alternatively, this trend may reflect the percolation of δ³⁰Si-depleted soil solutions into the saturated zone. Further evidence of plant uptake and harvesting is the low Si concentration in soil solutions and SGW. This decrease in concentration, compared to DGW, could be attributed to crop harvesting and the subsequent removal from the system, resulting in reduced DSi levels (Conley et al., 2008).

The current understanding of Ge-Si during plant uptake suggests that plants generally discriminate against Ge, resulting in
lower Ge/Si ratios in plants compared to soil solutions (Blecker et al., 2007; Cornelis et al., 2010; Delvigne et al., 2009; Derry et al., 2005; Lugolobi et al., 2010; Meek et al., 2016). However, our analyses of whole leaf tissue from maize and wheat do not show the expected discrimination of Ge over Si. On the contrary, the Ge/Si ratios measured in the leaves of both crops were higher than those found in groundwater, soil solution and stream water, implying a lack of Ge discrimination during maize and wheat uptake. This finding aligns with the observation from recent studies indicating that
when analyzing bulk plant material, plants do not consistently discriminate against Ge during uptake (Frings et al., 2021b; Kaiser et al., 2020). For instance, Delvigne et al. (2009) reported elevated Ge/Si ratios in bulk plant biomass following lithium-borate fusion. Similarly, Frings et al. (2021b) found high Ge/Si ratios in bulk plants (including grasses, spruce and pine trees) from three catchment following NaOH alkaline fusion. Furthermore, experiments tracking Ge and Si in hydroponic growth solutions for horsetail, banana and wheat found no resolvable discrimination during plant uptake
(Delvigne et al., 2009; Rains et al., 2006) while *in situ* imaging studies showed that Ge is often more localized in roots compared to shoots (Sparks et al., 2011). However, in a temperate forest, Meek et al. (2016) observed low Ge/Si ratios in both leaves and sap water of maples (*Acer*) and oaks (*Quercus*), along with enriched Ge/Si ratios in soil solutions during the growing season. These contrasting findings indicate that Ge discrimination during plant uptake may vary significantly depending on plant type and environmental conditions. Our results further support the notion that plant uptake of Ge relative
to Si is more complex than previously thought and may vary significantly between plant species or functional type.

## 5.2 Determining Si export from the catchment as biomass using an isotopic mass balance of Ge/Si and δ³⁰Si: $e_{org}^{Si}$

Assuming steady state and congruent weathering of bedrock, a set of mass balance equations can be formulated for any element and tracer within the weathering zone. The fraction of Si solubilized from the rock can follow three pathways: particulate erosion as secondary mineral, particulate erosion in biogenic material (including natural and harvested biomass),
or exported as dissolved Si in the stream. This approach, previously applied for both Si isotopes and Ge/Si ratios (Baronas et al., 2018; Bouchez et al., 2013; Frings et al., 2021a, b; Steinhoefel et al., 2017), is used here to determine the proportions of Si exported as dissolved species ($w_{iso}^{Si}$), as secondary minerals ($e_{sec-iso}^{Si}$), and as organic material ($e_{sec-iso}^{Si}$) as shown in the three following catchment-scale mass balance equations:

$$\delta^{30}Si_{rock} = e_{sec-iso}^{Si} * \delta^{30}Si_{sec} + e_{org-iso}^{Si} * \delta^{30}Si_{org} + w_{iso}^{Si} * \delta^{30}Si_{diss} \qquad (1)$$

$$\left(\tfrac{Ge}{Si}\right)_{rock} = e_{sec-iso}^{Si} * \left(\tfrac{Ge}{Si}\right)_{sec} + e_{org-iso}^{Si} * \left(\tfrac{Ge}{Si}\right)_{org} + w_{iso}^{Si} * \left(\tfrac{Ge}{Si}\right)_{diss} \qquad (2)$$



$$1 = e_{sec-iso}^{Si} + e_{org-iso}^{Si} + w_{iso}^{Si} \tag{3}$$

In these equations, the subscripts *rock*, *sec*, *org*, and *diss* refer to the Ge/Si ratios and $\delta^{30}$Si signatures of bedrock, secondary clay minerals, organic material, and stream, respectively. The term "iso" in the parameters ($w_{iso}^{Si}$, $e_{sec-iso}^{Si}$, and $e_{sec-iso}^{Si}$) indicates that these values were determined using isotopic signatures of $\delta^{30}$Si and Ge/Si in the mass balance. This system of three equations allows us to solve for the three unknowns: $w_{iso}^{Si}$, $e_{clay-iso}^{Si}$, and $e_{org-iso}^{Si}$. We used a Monte Carlo approach to account for uncertainties in Si isotopes and Ge/Si ratios. Three scenarios were run, assigning normal distributions to Ge/Si ratios and $\delta^{30}$Si, for secondary clays, bedrock and stream (Table 1). Each scenario varied the signatures for the organic fraction: in Scenario 1, we applied a uniform distribution based on the whole range of measured plant isotopic signatures to capture their high variability; in Scenario 2, we used a normal distribution based on measurements made on wheat leaves only; and in Scenario 3, we applied a normal distribution derived from measurements on maize only (details in Appendix D). The results between the three scenarios show similar trends in Si partitioning (Table 2 and Fig. 4). In all scenarios, the fraction $e_{clay-iso}^{Si}$ represents the least significant pool, with values ranging from 0.04 to 0.08. The fraction of $w_{iso}^{Si}$ presents values is around 0.15 to 0.22. The most significant pool of exported Si is found in organic material. Scenario 1 shows the highest fraction, with a mean $e_{org-iso}^{Si}$ of 0.81 ± 0.11. In contrast, scenario 2 and 3, which use the isotopic signature only from wheat and maize leaves, respectively yielded slightly lower $e_{org-iso}^{Si}$ values of 0.71 ± 0.19, and 0.71 ± 0.14 respectively. However, the three scenarios produced similar $e_{org-iso}^{Si}$ values within uncertainty range.

**Table 2. Parameters and results of the Si isotopic and Ge/Si mass balance used to determine $e_{sec-iso}^{Si}$, $e_{org-iso}^{Si}$, and $w_{iso}^{Si}$ under the three scenarios. SD represents the standard deviation, while Q1 and Q3 indicate the first (25th percentile) and third (75th percentile) quartiles of the data distribution.**

| Scenario | 1 | | | 2 | | | 3 | | |
|---|---|---|---|---|---|---|---|---|---|
| Samples used to constrain Ge/Si and $\delta^{30}$Si ranges | Combined maize and wheat | | | Wheat | | | Maize | | |
| Distribution | Uniform | min | max | Normal | mean | SD | Normal | mean | SD |
| Parameters | $\delta^{30}$Si$_{org}$ (‰) | -0.40 | 0.83 | $\delta^{30}$Si$_{org}$ (‰) | 0.51 | 0.46 | $\delta^{30}$Si$_{org}$ (‰) | -0.21 | 0.29 |
| | Ge/Si$_{org}$ (µmol mol$^{-1}$) | 1.31 | 3.88 | Ge/Si$_{org}$ (µmol mol$^{-1}$) | 3.27 | 0.86 | Ge/Si$_{org}$ (µmol mol$^{-1}$) | 1.61 | 0.42 |
| Monte Carlo results | | | | | | | | | |
| Fluxes | $w_{iso}^{Si}$ | $e_{clay-iso}^{Si}$ | $e_{org-iso}^{Si}$ | $w_{iso}^{Si}$ | $e_{clay-iso}^{Si}$ | $e_{org-iso}^{Si}$ | $w_{iso}^{Si}$ | $e_{clay-iso}^{Si}$ | $e_{org-iso}^{Si}$ |
| Mean | 0.15 | 0.04 | 0.81 | 0.21 | 0.08 | 0.71 | 0.22 | 0.07 | 0.71 |
| Median | 0.15 | 0.03 | 0.83 | 0.20 | 0.05 | 0.73 | 0.22 | 0.05 | 0.72 |
| SD | 0.08 | 0.05 | 0.11 | 0.13 | 0.09 | 0.19 | 0.11 | 0.06 | 0.14 |
| Q1 | 0.09 | 0.01 | 0.77 | 0.10 | 0.02 | 0.62 | 0.14 | 0.03 | 0.62 |
| Q3 | 0.19 | 0.05 | 0.88 | 0.30 | 0.11 | 0.84 | 0.30 | 0.09 | 0.81 |





Our calculations indicate that particulate Si export via plant material (natural and anthropogenic) is the largest contributor to Si export from the catchment, with estimated fractional losses approximately from $0.74 \pm 0.19$ to $0.81 \pm 0.11$. This represents values 3.2 to 5.4 times greater than the total dissolved Si export. These contributions are also higher than those observed in three other non-agricultural catchments, where $e_{org-iso}^{Si}$ ranged from $0.20 \pm 0.12$ to $0.42 \pm 0.23$ (Frings et al., 2021b) and in a

global river compilation with $e_{org-iso}^{Si}$ of $0.32 \pm 0.12$ (Baronas et al., 2018). These findings highlight the significant impact of harvesting on the Si cycle, which can thus sharply reduce dissolved Si fluxes in streams.

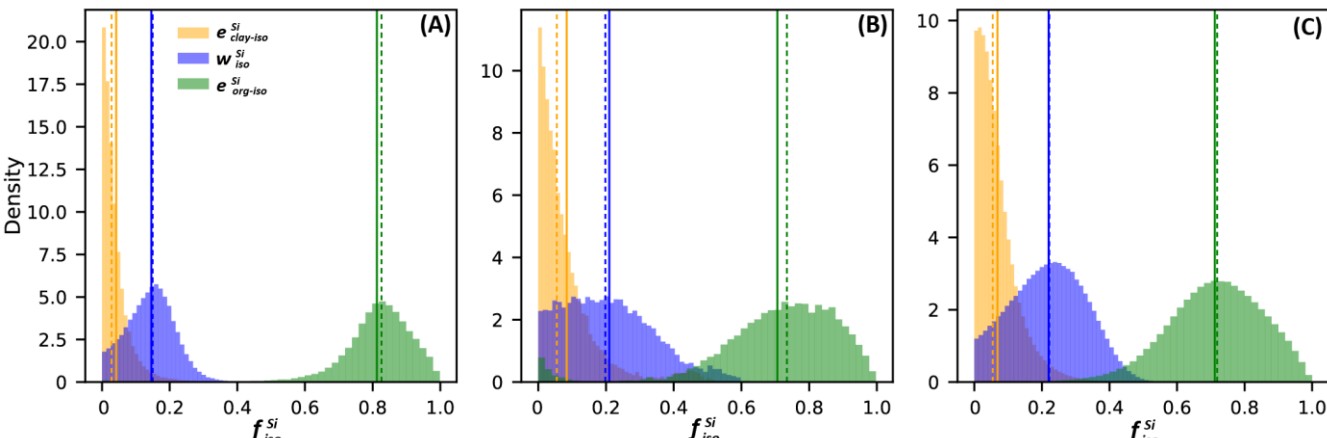

**Figure 4. Distribution of the fraction of Si export as $e_{clay-iso}^{Si}$, $e_{org-iso}^{Si}$, and $w_{iso}^{Si}$ calculated using the combined δ³⁰Si and Ge/Si isotopic mass balance for three scenarios with varying signatures in the organic fraction. Solid lines represent the mean of each**
**fraction, while dashed lines indicate the median. (A) Scenario 1 assumes a uniform distribution for the plant signatures, incorporating all measured leaf values. (B) Scenario 2 uses a normal distribution using only the wheat signatures. (C) Scenario 3 applies a normal distribution using only the maize signatures.**

### 5.3 Quantifying Si removal from the system by harvesting

Silica harvest estimation can be determined by assessing the biomass and type of grain exported by each farmer, following
the global estimation methodologies of Carey & Fulweiler (2016) and Matichenkov & Bocharnikova (2001). However, in our study area, robust estimates are unavailable, and a portion of the harvested biomass is reincorporated into the watershed as manure. This recycling complicates the calculation of net Si export, as Si cycling between crops and manure varies depending on farming practices and crop type. In the following sections, we estimate silica harvest from the catchment using two independent approaches based on the framework developed by Bouchez et al. (2013). This approach has been used to
quantify nutrient uptake with a variety of metal stable isotope systems (Charbonnier et al., 2020, 2022; Schuessler et al., 2018; Uhlig et al., 2017) and more specifically with Si isotopes (Baronas et al., 2018; Frings et al., 2021a). To determine the export of Si by harvesting we first perform an elemental mass balance using direct flux estimates based on riverine chemistry and suspended sediments ($h_{river}^{Si}$); second, we use an estimate of the Si isotope fractionation factor associated to plant uptake together with indexes of Si depletion in soils ($h_{regolith}^{Si}$).




## 5.4 Quantifying Si export through harvesting using riverine Si concentrations and suspended sediments: $h_{river}^{Si}$

In this approach, the catchment is considered as a steady-state open flow-through reactor, where dissolved Si is derived from partial rock dissolution and subsequently partitioned into various compartments. Figure 5 shows a schematic diagram representing the compartments and fluxes of the model and Table 3 presents the definitions of all the parameters and symbols used. At steady state, the total Si denudation rate accounts for the export of both dissolved and particulate Si, expressed as:

$$D * [Si]_{rock} = W^{Si} + E^{Si} \tag{4}$$

where $D$ is the total denudation rate (kg m$^{-2}$ yr$^{-1}$) equal at steady-state to the conversion of rock into regolith material, $[Si]_{rock}$ is the Si concentration in the bedrock (mol kg$^{-1}$), $W^{Si}$ is the dissolved Si exported from the system (mol m$^{-2}$ yr$^{-1}$), and $E^{Si}$ is the export of Si contained in river solid particles (mol m$^{-2}$ yr$^{-1}$). The solid-phase export of Si, $E^{Si}$ includes Si located in secondary minerals $E_{sec}^{Si}$, organic matter $E_{org}^{Si}$ and primary minerals $E_{prim}^{Si}$:

$$E^{Si} = E_{sec}^{Si} + E_{prim}^{Si} + E_{org}^{Si} \tag{5}$$

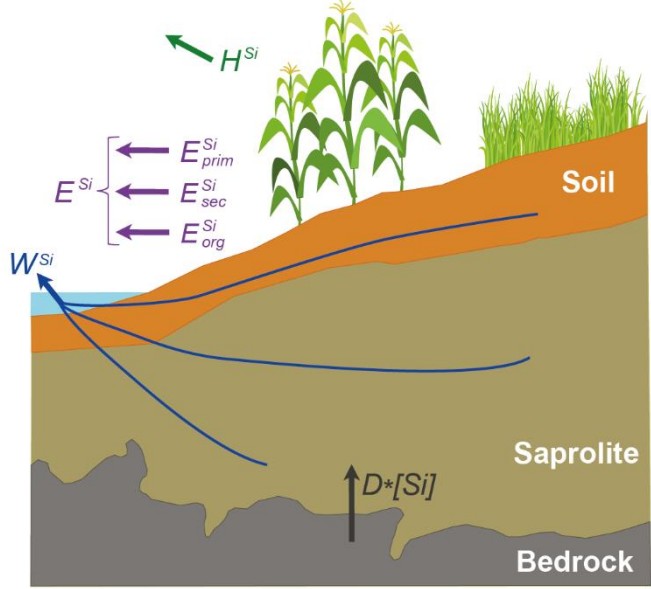

**Figure 5. Schematic diagram representing the compartments and fluxes in the mass balance model**

In a natural system, the Si removed by plants returns to the weathering zone as litter and then can be redissolved or exported as particulate organic matter. However, in an agricultural catchment, a fraction of the Si removed by plants does not return as litter but is removed due to harvesting $H^{Si}$ (mol m$^{-2}$ yr$^{-1}$). Thus Eq. (4) becomes:

$$D * [Si]_{rock} = W^{Si} + E^{Si} + H^{Si} \tag{6}$$



where $H^{Si}$ is the flux of Si exported in plants from the catchment due to agriculture (mol m⁻² yr⁻¹). Dividing both sides of Eq. (6) by $D * [Si]_{rock}$, we obtain the relative proportion of all fluxes with respect to the denudation flux:

$$1 = w^{Si} + e^{Si} + h^{Si} \qquad (7)$$

If $w^{Si}$ and $e^{Si}$ are known, we can estimate the fraction of Si being exported due to harvesting.

In the following sections, we determine $w^{Si}$ leveraging stream gauging and chemical data ($w^{Si}_{river}$) and estimate $e^{Si}_{river}$ based on suspended sediments data from the stream (Vongvixay et al., 2018).

**Table 3. Summary of symbols used in the equations of the mass balance model.**

| | |
|---|---|
| **Total mass fluxes (kg m⁻² yr⁻¹)** | |
| $D$ | Denudation (*e.g.* estimated through measurements of beryllium-10 in quartz) |
| **Elemental fluxes (mol m⁻² yr⁻¹)** | |
| $W^{Si}$ | Dissolved export of Si |
| $E^{Si}$ | Particulate export of Si including erosion of primary and secondary minerals and organic solid material |
| $E^{Si}_{prim}$ | Particulate export of Si contained in primary minerals |
| $E^{Si}_{sec}$ | Particulate export of Si contained in secondary minerals |
| $E^{Si}_{org}$ | Particulate export of Si contained in organic matter |
| $H^{Si}$ | Removal of Si due to harvesting of crops |
| $E^{Si}_{river}$ | Particulate export of Si including erosion from primary and secondary minerals and organics (determined by particulate material export estimation and stream gauging) |
| $W^{Si}_{river}$ | Dissolved export of Si (determined from stream gauging and chemical monitoring data) |
| $H^{Si}_{river}$ | Removal of Si due to crop harvesting (determined from stream gauging, chemical monitoring data and particulate material export estimation) |
| $H^{Si}_{regolith}$ | Removal of Si due to crop harvesting determined by Si loss indexes and isotopic fractionation factors |
| **Normalized elemental fluxes to the regolith production (non-dimensional, 0 to 1)** | |
| $w^{Si}_{river}$ | Dissolved export of Si determined from stream gauging and chemical monitoring data normalized to the Si flux associated with regolith production |
| $e^{Si}_{river}$ | Particulate export of Si including erosion from primary and secondary and organic (determined by particulate material estimations and stream gauging) normalized to the Si flux associated to regolith production |
| $h^{Si}_{river}$ | Removal of Si due to harvesting crops (determined from stream gauging, chemical monitoring data and particulate material export estimation) normalized to the Si flux associated to regolith production |
| $e^{Si}_{sec}$ | Particulate export of Si contained in secondary minerals normalized to the Si flux associated to regolith production |
| $e^{Si}_{org}$ | Particulate export of Si contained in organic matter normalized to the Si flux associated to regolith production |
| $h^{Si}_{regolith}$ | Removal of Si due to harvesting crops determined by Si loss indexes and isotopic fractionation factors, normalized to the Si flux associated to regolith production |
| **Weathering indexes (non-dimensional)** | |
| $\tau_{prim}$ | Mass transfer coefficient of Si from primary minerals |
| $CDF$ | Chemical depletion fraction (fraction of mass loss by weathering between regolith material and parent rock) |




### 5.4.1 Si dissolved fluxes based on riverine dissolved Si: $w_{river}^{Si}$

We determined the annual $w^{Si}$ over a period of 22 years using the measured dissolved Si river flux at the catchment ($W_{river}^{Si}$). The dissolved annual river fluxes were calculated as:

$$W_{river}^{Si} = \sum_{i=1}^{8760} \frac{[Si]_{riv} * Q}{A} \qquad (8)$$

where $[Si]_{riv}$ represents hourly Si concentration in the stream, $i$ refer to the hours within a hydrological year, $Q$ is the stream water hourly discharge, and $A$ is the catchment area ($m^2$). Hourly discharge data are available from 2001 to 2022 at https://agrhys.fr/BVE/vidae/. Daily Si concentration measurements are available from fall 1999 until summer 2000, with additional samples in 2003, 2006, 2007, and during three flood events in 2021 and 2022 at an hourly resolution. This dataset includes 844 samples, covering various seasons and hydrological conditions (Table 4; López-Urzúa et al., 2025).

To better estimate an average stream dissolved Si concentrations over 22 years using a more continuous (hourly) discharge record (Appendix E), we derived a relationship between Si concentration and discharge in a logarithmic space (C-Q relationship). We used IsoplotR's maximum likelihood estimator (Vermeesch, 2018) to determine best parameters of a power law equation describing the relationship between Si (µmol L$^{-1}$) and Q (L s$^{-1}$) as follows:

$$[Si] = k \times Q^b \qquad (9)$$

yielding $b$ = -0.09 ± 0.014 (SE, standard error) and $k$ = 195.9 ± 1.06 (SE). Using Eq. (9), we estimated hourly Si concentrations for each hour over the 22-year period. We then calculated the annual $W_{river}^{Si}$ for each hydrological year from 2000 to 2022 (Table 5; López-Urzúa et al., 2025) using Eq. (8). For additional details on the estimation of riverine Si flux and the calculation of $W_{river}^{Si}$ see Appendix E.

The average $W_{river}^{Si}$ over the 22-year period is 0.042 ± 0.02 mol m$^{-2}$ yr$^{-1}$ (SD) (Table S1 in the supplement). The non-
dimensional dissolved Si flux ($w_{river}^{Si}$; Table S1 in the supplement) is derived by dividing this value by the total Si denudation rate ($[Si]_{rock} * D$). Using a denudation rate of 10 ± 1.7 m Ma$^{-1}$ (Malcles et al., 2023) and a bedrock Si concentration of 82 wt. % SiO$_2$ (Table 1), the total Si denudation rate yields a value of 0.345 ± 0.059 mol m$^{-2}$ yr$^{-1}$. The mean fraction of Si exported from the catchment in the dissolved form is 0.12 ± 0.06, which aligns with the values calculated from the isotopic mass balance. This fraction represents only a small proportion of the total Si export and suggests that particulate
erosion and harvesting are major contributors to Si export.

### 5.4.2 Si particulate erosion flux: $e_{river}^{Si}$

To estimate $e_{river}^{Si}$, we used the correlation between annual suspended sediments concentration and annual water specific flux at the Kervidy-Naizin catchment of Vongvixay et al. (2018). This relationship is based on a study period spanning 9 hydrological years (2005 to 2014) during which turbidity was recorded every 10 minutes. Subsequently, suspended sediment
concentration (SSC) was computed using corrected turbidity values as an index data, alongside an SSC–turbidity rating



curve with a correlation coefficient of approximately 0.75 established from over 300 stream samples. Vongvixay et al. (2018) established a linear relationship between suspended sediment concentrations and the annual water specific flux. The regression equation describing their relationship is *SSC = 0.036Q−1.72* with $R^2$ value of 0.787, where *SSC* is the annual suspended sediment flux (t km$^{-2}$) and *Q* is the annual runoff (mm). Leveraging these established relationships, we calculated the annual suspended sediment for our 22 years with discharge data. Subsequently we calculated the annual value of $e^{Si}$ by multiplying the particulate erosion flux by the Si concentration we measured in the topsoil. Our calculations show that the fraction of Si exported as solid particles $E_{river}^{Si}$ averaged $0.12 \pm 0.07$ mol m$^{-2}$ yr$^{-1}$ (SD) with a mean estimated $e_{river}^{Si}$ fraction of $0.36 \pm 0.21$.

### 5.4.3 Fraction of Si exported by harvesting: $h_{river}^{Si}$

After determining the mean $e_{river}^{Si}$ and $w_{river}^{Si}$ over the 22-year period, we calculate the fraction of Si exported from the catchment $h^{Si}$ for each year using Eq. (7) (Table S1 in the supplement). To derive the mean $h_{river}^{Si}$ and account for uncertainties, we performed a Monte Carlo approach with 10,000 simulations assuming normal distributions for the three parameters (Fig. 6). The mean $h_{river}^{Si}$ from this simulation is $0.50 \pm 0.19$ (SD), indicating that between 20 to 70% of the Si release from bedrock is exported from the catchment through harvesting practices. As the uncertainty on the input parameters partially stems from the temporal variability in river fluxes, the resulting uncertainty in $h_{river}^{Si}$ is closely related to annual flux variations. Years with significant rainfall increase erosion, leading to increased export of Si as sediments. In this context, the mean over 22 years serves as a valuable estimator, as it integrates data across diverse hydrological conditions, making it more comparable to denudation estimates that relate for longer time scales.

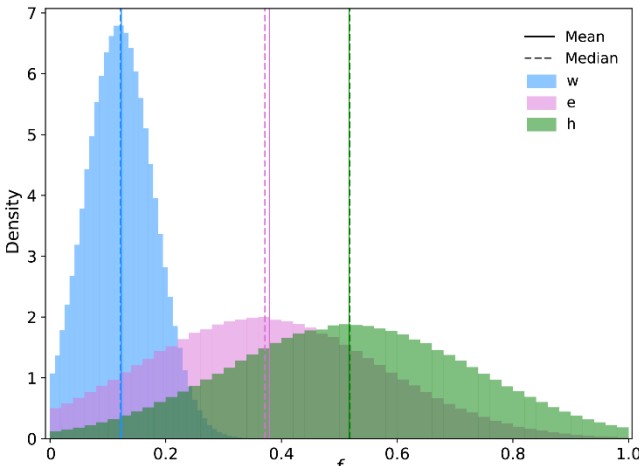

**Figure 6. Distribution of the fraction of Si export as $w_{river}^{Si}$, $e_{river}^{Si}$, and $h_{river}^{Si}$, calculated using riverine Si concentrations, suspended sediments and the mass balance. The solid line represents the mean, while the dashed line indicates the median.**



### 5.5 Quantifying Si export through harvesting using Si loss indexes and isotopic fractionation factors: $h_{regolith}^{Si}$

The isotopic composition of each compartment within the Critical Zone can be related to elemental fluxes and isotope fractionation factors. Because at Kervidy-Naizin a portion of organic particulate erosion originates from harvesting, we reformulate Eq. (5e) and Eq. (15) from Bouchez et al. (2013). Then by normalizing to the total Si flux (detailed derivation is provided in Appendix F), we determined the fraction of Si harvested as:

$$h_{regolith}^{Si} = \frac{(\delta^{30}Si_{diss} - \delta^{30}Si_{rock}) * \tau_{prim}^{Si} - e_{sec}^{Si} * \varepsilon_{prec}^{Si} - e_{org}^{Si} * \varepsilon_{upt}^{Si}}{\varepsilon_{upt}^{Si}} \tag{10}$$

Here, $h_{regolith}^{Si}$ represents the fraction of Si harvested, calculated as a function of normalized Si fluxes ($e_{sec}^{Si}$ and $e_{sec}^{Si}$; Table 3), isotope fractionation factors for secondary mineral precipitation ($\varepsilon_{prec}^{Si}$) and plant uptake ($\varepsilon_{upt}^{Si}$), and the mass transfer coefficient of Si from primary minerals ($\tau_{prim}^{Si}$) defined as:

$$\tau_{prim} = \frac{[Si]_{prim/reg} * [Ti]_{rock}}{[Si]_{prim/rock} * [Ti]_{reg}} - 1 \tag{11}$$

where $[Si]_{prim/reg}$ and $[Si]_{prim/rock}$ represents the Si concentration associated with primary mineral in the regolith and rock, respectively, and the $[Ti]_{reg}$ and $[Ti]_{rock}$ terms corresponds to the Ti concentration in the rock and regolith.

### 5.5.1 Si isotope fractionation associated with vegetation and clay formation: $\varepsilon_{prec}^{Si}$ and $\varepsilon_{upt}^{Si}$

To estimate the fractionation factors for secondary mineral precipitation and plant uptake, we used the isotopic signatures from the stream ($\delta^{30}Si_{diss}$), plants ($\delta^{30}Si_{org}$) and secondary minerals ($\delta^{30}Si_{sec}$). The fractionation factors were determined as follows (Bouchez et al., 2013):

$$\varepsilon_{prec}^{Si} = \delta^{30}Si_{sec} - \delta^{30}Si_{diss} \tag{12}$$

$$\varepsilon_{upt}^{Si} = \delta^{30}Si_{org} - \delta^{30}Si_{diss} \tag{13}$$

Using the mean $\delta^{30}$Si from the stream and the clay-corrected fraction (Table 1), we calculated $\varepsilon_{prec}^{Si}$ of -2.42 ± 0.03‰ (Table 4). This clay neoformation fractionation factor is similar to that estimated for other environments dominated by kaolinite as the secondary mineral (Frings et al., 2021a).

**Table 4. Si isotopic fractionation factor determined for secondary clay precipitation and plant uptake. Uncertainty obtained through gaussian error propagation.**

| Fractionation factors (‰) | |
|---|---|
| $\varepsilon_{sec}^{Si}$ | -2.42 ± 0.07* |
| $\varepsilon_{upt}^{Si}$ | -0.96 ± 0.26* |
| $\varepsilon_{upt-wheat}^{Si}$ | -0.62 ± 0.16* |
| $\varepsilon_{upt-maize}^{Si}$ | -1.32 ± 0.24* |






For the plant uptake fractionation factor ($\varepsilon_{upt}^{Si}$), we derived a value of -0.96 ± 0.26‰ using the mean δ³⁰Si from dissolved Si in the stream and leaf samples. We further assessed the fractionation factors specific to each crop species, maize and wheat. $\varepsilon_{upt-wheat}^{Si}$ is -0.62 ± 0.07‰, similar to values reported by Frick et al. (2020) for spring wheat (-0.43‰). $\varepsilon_{upt-maize}^{Si}$ is -1.32 ± 0.07‰, consistent with previously documented values by Ziegler et al. (2005a) and Sun et al. (2017), who found -1.0 and -

1.1‰ respectively.

### 5.5.2 Si loss index: $\tau_{prim}^{Si}$

We first performed a quartz correction to the Si concentrations in bedrock and soil samples to account for the non-stoichiometric dissolution of the primary mineral assemblage where Si is preferentially released from more reactive plagioclase, muscovite and chlorite whereas quartz is assumed to by inert.

To calculate $\tau_{prim}^{Si}$, we assessed the content of Si located in primary minerals based on the mineralogical composition and stoichiometry derived from our bedrock and soil profiles (Table S2 in the supplement). The Si associated with each mineral was calculated using stoichiometry, and $\tau_{prim}$ was determined as follows:

$$\tau_{prim}^{Si} = \frac{([Si]_{muscovite}+[Si]_{albite}+[Si]_{microcline}+[Si]_{chlorite})_{soil}*[Ti]_{rock-qtz}}{([Si]_{muscovite}+[Si]_{albite}+[Si]_{microcline}+[Si]_{chlorite})_{rock}*[Ti]_{soil-qtz}} - 1 \tag{14}$$

where the concentrations on Ti and Si in soil samples were corrected from quartz. The Si concentration terms $[Si]i$ in Eq.

(14) represents the Si content associated with each primary mineral $i$ identified by XRD and were calculated based on their mineralogical abundance and stoichiometry for each horizon sample (Table S2 in the supplement). From our analysis across both profiles (Table S2 in the supplement), we calculated an average $\tau_{prim}^{Si}$ value of -0.42 ± 0.15 (SD, $n = 6$).

### 5.5.3 Particulate erosion fluxes: $e_{org}^{Si}$ and $e_{sec}^{Si}$

The particulate erosion fluxes, $e_{sec}^{Si}$ and $e_{org}^{Si}$, can be expressed as follows (Bouchez et al., 2013):

$$e_{sec}^{Si} = (1 - CDF) * \frac{[Si]_{sec}}{[Si]_{rock-qtz}} \tag{15}$$

$$e_{org}^{Si} = (1 - CDF) * \frac{[Si]_{org}}{[Si]_{rock-qtz}} \tag{16}$$

Here, $[Si]_{sec}$ and $[Si]_{org}$ represents the Si concentrations associated to secondary minerals and particulate organic material in the sediments, respectively, while $[Si]_{rock-qtz}$ is the Si concentration in the rock corrected for quartz. The chemical depletion fraction (CDF) (Riebe et al., 2001), is given by:

$$CDF = 1 - \frac{[Ti]_{rock-qtz}}{[Ti]_{soil-qtz}} \tag{17}$$



The CDF represents the overall fractional mass loss occurring during weathering and export of dissolved elements. Across our profiles (Table S2 in the supplement), we calculated an average CDF value of $0.29 \pm 0.13$ (SD, $n = 6$).

To determine the Si concentration associated with secondary minerals $[Si]_{sec}$ we followed a similar approach as for $\tau_{prim}$, focusing on the percentage and stoichiometry of kaolinite and vermiculite in each sample. For estimating the Si

concentration in organic matter $[Si]_{org}$, we relied on measured organic matter content in the upper soil horizons (0 to 30 cm of each profile, Table S2 in the supplement), assuming that 2.3% of this organic matter consists of Si, as reported for wheat (Hodson et al., 2005). The term $e_{sec}^{Si}$ yielded a value of $0.087 \pm 0.042$ (SD, n = 6), and the term $e_{org}^{Si}$ a value of $0.0064 \pm 0.00032$ (SD, n = 2).

### 5.5.4 Determining $h_{regolith}^{Si}$

After determining all the parameters in Eq. (10), we calculated $h_{regolith}^{Si}$ for three different scenarios, each one applying a different fractionation factor for plant uptake. To solve Eq. (10) and obtain associated uncertainties, we conducted a Monte Carlo simulation for each scenario (Table 5).

**Table 5. Parameters and results of the isotopes fractionation factors used to determine $h_{regolith}^{Si}$ under the three scenarios. SD represents the standard deviation, while 5% and 95% indicate the 5th and 95th percentiles of the data distribution.**

| Scenario | 1 | | | 2 | | | 3 | | |
|---|---|---|---|---|---|---|---|---|---|
| Samples used to constrain fractionation factor ranges | Combined maize and wheat | | | Wheat | | | Maize | | |
| Distribution | Uniform | min | max | Normal | mean | SD | Normal | mean | SD |
| Fractionation factor (‰) | $\varepsilon_{upt-all}^{Si}$ | -1.52 | -0.29 | $\varepsilon_{upt-wheat}^{Si}$ | -0.62 | 0.16 | $\varepsilon_{upt-maize}^{Si}$ | -1.32 | 0.24 |
| **Monte Carlo results: $h_{regolith}^{Si}$** | | | | | | | | | |
| **Mean** | 0.38 | | | 0.46 | | | 0.26 | | |
| **Median** | 0.34 | | | 0.45 | | | 0.25 | | |
| **SD** | 0.23 | | | 0.25 | | | 0.15 | | |
| **5%** | 0.06 | | | 0.07 | | | 0.04 | | |
| **95%** | 0.84 | | | 0.90 | | | 0.54 | | |


In Scenario 1, we adopted a uniform distribution for $\varepsilon_{upt}^{Si}$ using the minimum and maximum values of the fractionation factors using all plant samples ($n = 4$). The overall fractionation factor of harvested plants will vary from year to year depending on the species grown and is calculated as the weighted mean of the different crop fractionation factors, proportionate to their prevalence in the basin. By assuming a uniform distribution, we attribute the same probability to each

plant species across the range of possible fractionation factors, from the minimum to the maximum values derived from our samples. This approach allows us to capture the full spectrum of potential fractionation factors, ultimately providing a more comprehensive assessment of Si export due to harvesting practices in the catchment. In contrast, for scenarios 2 and 3 we





employed a normal distribution for each species, characterized by their respective average and standard deviation. Specifically, in Scenario 2, we used the wheat fractionation factor $\varepsilon_{upt-wheat}^{Si}$ while in Scenario 3, we use the maize

fractionation factor $\varepsilon_{upt-maize}^{Si}$.

The results show averages values of $h_{regolith}^{Si}$ ranging from 0.26 to 0.46 depending on the scenario (Table 5 and Fig. 7). The highest fraction of harvested Si is reached using the wheat fractionation factor, yielding a mean of $0.46 \pm 0.25$, while the lowest value is observed in scenario 3, using the maize fractionation factor with a value of $0.26 \pm 0.15$. Scenario 1 yielded a mean value of $0.38 \pm 0.23$.

Given that maize and wheat are the most prevalent cultivars in the catchment, Scenario 1 appears to be the most suitable for estimating Si export due to harvesting. However, our plant sampling strategy, which focused solely on leaves, restricts our ability to accurately determine the overall plant fractionation factor. During transpiration, lighter isotopes are incorporated into phytoliths, resulting in an enrichment of the remaining solution with heavier isotopes. As a result, the phytoliths in the upper parts of the plant (leaves) may exhibit a heavier isotopic signature than those in lower parts, such as stems and roots.

As we only measured leaves, the overall plant fractionation factor is likely to be heavier. Therefore, the most accurate estimate of $h_{regolith}^{Si}$ is likely between Scenario 1 (0.38), which accounts for the presence of different crops and Scenario 2 (0.46), which reflects a lower fractionation factor.

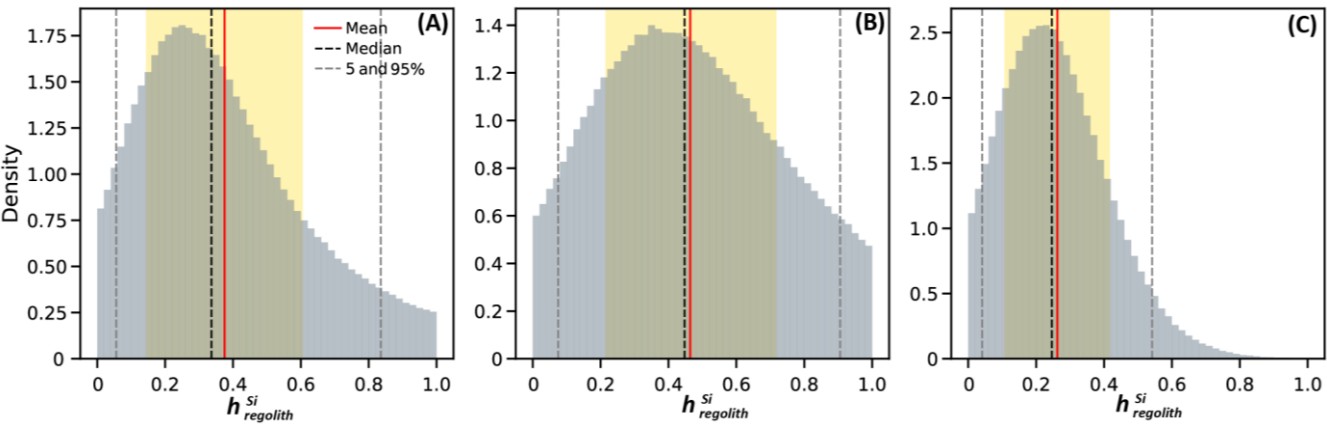

**Figure 7. Distribution of the fraction of harvested Si ($h_{regolith}^{Si}$) calculated using Si loss indexes and isotopic fractionation factors for three scenarios with varying plant isotope fractionation factor. Red lines indicate the mean, black dashed lines represent the median, and grey dashed lines mark the 5th and 95th percentiles. Yellow shaded areas denote the mean ± one standard deviation (1 SD). (A) Scenario 1 assumes a uniform distribution for the plants isotope fractionation factor, incorporating all fractionations factors determined in leaves. (B) Scenario 2 uses a normal distribution based only on the wheat fractionation factor. (C) Scenario 3**
**applies a normal distribution based only on the maize fractionation factor.**



### 5.6 Uncertainties in H$^{Si}$ and implications of harvesting on the Si cycle

Our estimates indicate that Si flux from harvesting, depending on crop type, accounts for 1.2 to 4.2 times the dissolved Si flux at the agricultural catchment of Kervidy-Naizin, highlighting the strong impact that harvesting can have on the biogeochemical cycle of Si.

Our two independent methods for quantifying Si export due to harvesting produced consistent results. Using the first method, based on an elemental mass balance of riverine solute chemistry and suspended sediments over a 22-year period, we estimated that the highest fraction of Si exported through harvesting, $h_{river}^{Si}$ was 50 ± 20% of the total Si exported of the catchment, amounting to 0.179 ± 0.076 mol m$^{-2}$ yr$^{-1}$ of Si losses. We assume that the imbalance between denudation and particulate erosion and weathering represents the Si exported by harvesting. However, this approach has two limitations.

The first limitation is the mismatch between the short time scale of flux measurements (22 years for particulate erosion and weathering) and the much longer integration time (20 kyr and 200 kyr) associated with the estimation of denudation rates based on beryllium isotopes (Malcles et al., 2023). The beryllium-based denudation used in this study reflects an average value and may not have remained constant throughout the Late Pleistocene-Holocene. Instead, it likely fluctuated in response to climatic changes. As we are comparing short-term data with a long-term average, the actual differences from the onset of

agriculture at the site could be either larger or smaller, depending on whether the current denudation rate is higher or lower than its historical mean over the past ~200 kyr. If confirmed, this discrepancy could affect our calculated $h_{river}^{Si}$. The second limitation is the potential omission of unaccounted Si pools, such as large plant debris (Uhlig et al., 2017), which were not sampled in this study. This limitation is particularly relevant during flood events, when large amounts of organic material, including plant debris, are mobilized and contribute to the overall Si flux. Consequently, our $h_{riv}^{Si}$ estimate may be an

overestimate due to the exclusion of these Si pools.

Our second approach based on isotopic mass balance using fractionation factors and Si loss indices ($h_{regolith}^{Si}$) estimated that Si export by harvesting represents 37 ± 10% of the total Si export, equating to 0.127 ± 0.035 mol m$^{-2}$ yr$^{-1}$ of Si losses. The primary limitation of this method is the heterogeneity of the catchment soils and bedrock. The bedrock is characterized by a succession of siltstones and sandstones of varying composition, complicating efforts to determine the parent material of soil

profiles accurately. For instance, rocks sampled near the regolith profile may not accurately represent those drained by the stream (Bouchez and Von Blanckenburg, 2021), introducing uncertainty in our $h_{regolith}^{Si}$ calculations.

Another potential limitation of our approach involves the geochemistry of manure. A considerable portion of harvested biomass is used for livestock feed, with some manure returned to catchment. In both isotopic methods, we assumed that the isotopic signatures (δ$^{30}$Si and Ge/Si ratios) of manure match those of plants. However, Si fractionation may occur in grazers

since Si plays a role in connective tissues, especially in bone and cartilage, in warm-blooded animals (Carlisle, 1988). Nevertheless, studies indicate that the amount of Si retained in animals after plant ingestion is negligible (Jones and Handreck, 1967). Research on hippos found no significant difference between δ$^{30}$Si in plant forage (−0.42 ± 0.89‰) and their feces (−0.52 ± 0.71‰)(Schoelynck et al., 2019). Still, if fractionation occurs, the large fractionation factor used in our



scenarios could also account for potential differences between manure and plant isotopic signatures. Regarding germanium,
as its behavior closely mirrors that of Si, the Ge/Si signature in manure is expected to largely reflect the original Ge/Si ratio
of the plants consumed by animals. However, further research is needed to support this assumption.

Nevertheless, despite these limitations, our estimates remain robust, providing valuable insights into the role of harvesting in Si export and its implications for the biogeochemical cycle of Si in the catchment.

## 6 Conclusions

This study demonstrates that agricultural activities significantly alter the Si cycle in terrestrial landscapes, particularly within the Critical Zone of agricultural catchments. Our analysis of $\delta^{30}$Si and Ge/Si ratios across soil, bedrock, water, and plants in the Kervidy-Naizin catchment revealed a vertical gradient in $\delta^{30}$Si, influenced by distinct weathering and biological processes. In deeper groundwater, $\delta^{30}$Si is regulated by weathering and clay precipitation, while in shallower soils, biological uptake and agricultural harvesting lead to a progressive enrichment in $\delta^{30}$Si and decrease on Si concentrations.

Notably, our analyses challenge the conventional view of Ge discrimination during plant uptake for the specific crops examined, as we found no significant Ge discrimination relative to Si in bulk maize and wheat tissue. Instead, elevated Ge/Si ratios in maize and wheat leaves, compared to groundwater, soil solutions, and stream water, align with recent studies suggesting that plants do not consistently discriminate against Ge in bulk material. Our findings suggest that Ge discrimination likely varies significantly among plant types, emphasizing the need for caution when generalizing

fractionation patterns. Grass, such as maize and wheat, may exhibit different Ge/Si fractionation behaviors compared to woody plants.

The quantification of Si export as harvested plant material underscores the substantial impact of agricultural harvesting, which accounted for a significant fraction of the Si export catchment, ranging from approximately 3 to 5 times the dissolved Si flux. Our findings reveal that agricultural harvesting alone represents a primary pathway for Si removal from terrestrial

ecosystems, emphasizing its role in reducing Si availability downstream. The two independent approaches we employed—an elemental mass balance based on riverine chemistry and suspended sediments, which estimated Si export at $50 \pm 19\%$ of the total Si, equivalent to $0.179 \pm 0.076$ mol m$^{-2}$ yr$^{-1}$, and an isotopic mass balance integrating fractionation factors, which estimated $37 \pm 10\%$ or $0.127 \pm 0.035$ mol m$^{-2}$ yr$^{-1}$—produced consistent values, supporting the robustness of these methods in assessing Si fluxes.

Overall, this study underscores the extent to which human activities reshape the natural Si cycle in agricultural settings. As anthropogenic influence over terrestrial ecosystems continues to expand, further research is essential to identify and quantify unforeseen ways in which human actions alter global Si biogeochemistry.



## Appendix A

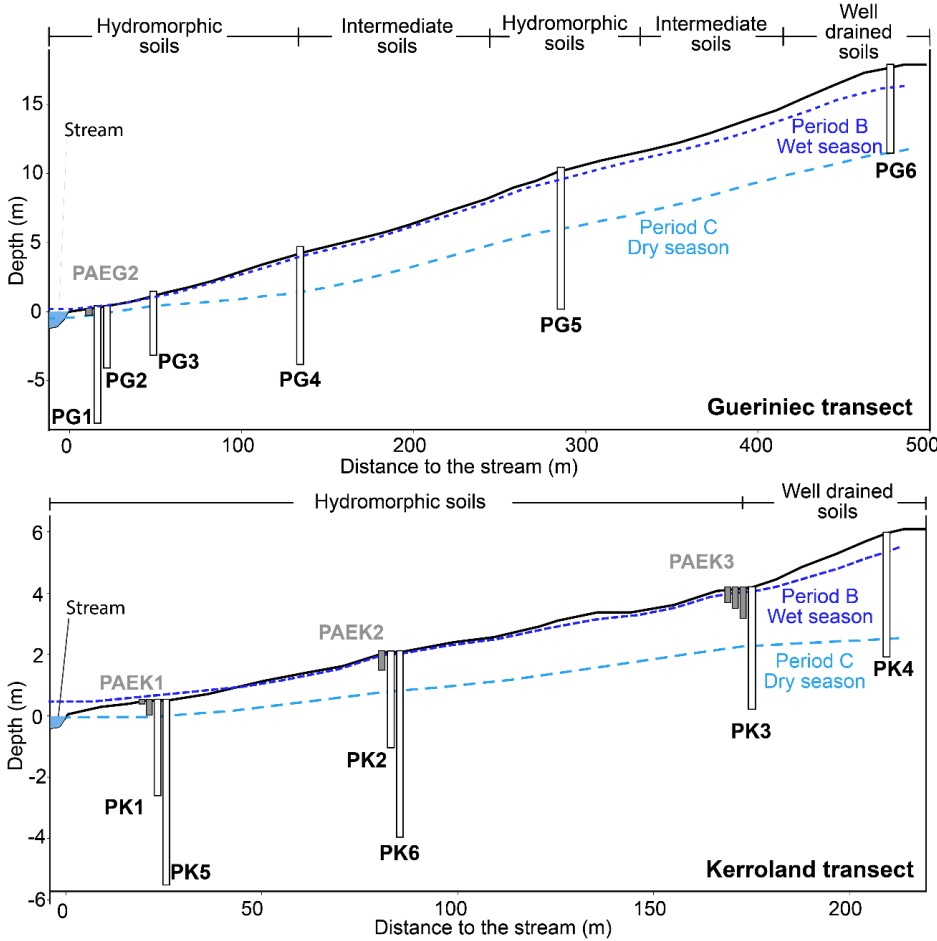

**Figure A1. Cross sections of the Gueriniec and Kerroland transects. Black rectangles represent piezometers and grey rectangles lysimeters. The dark blue dotted line represents the piezometer level during high flow in the wet season (Period b, Humbert et al., 2015; Molenat et al., 2008) and the light blue dashed line the water level during recession in the dry season (Period C).**

## Appendix B: Mineralogy and correction of clay fraction

To isolate the composition of secondary phases in the clay sized fraction, we applied a correction to account for contamination by clay-sized primary minerals. First, we determined the mineralogy of the clay fraction by inverting a simple mixing model (Aguirre et al., 2017), assuming that the mineral composition consists of quartz ($SiO_2$), kaolinite ($Al_2Si_2O_5(OH)_4$), muscovite ($KAl_2(AlSi_3O_{10})(OH)_2$) and microcline ($KAlSi_3O_8$). This can be represented in matrix form as: $Ax=b$, where $A$ is the matrix of mineral chemistry, $b$ is the composition clay fraction (Table B1), and $x$ is the vector of mineral fractions, such that $f_{muscovite} + f_{kaolinite} + f_{quartz} + f_{microcline} = 1$. For both samples we found that approximately 50% of the clay fraction is composed of kaolinite, with the remainder attributed to primary minerals (Table B1).



Once the mineral fractions were determined, we correct the Si and Al concentration to account for the contribution from kaolinite. To adjust the $\delta^{30}Si$ and and Ge/Si ratios, we assumed that primary minerals retained the isotopic signature of the bedrock. By considering the fractions of each mineral, we were able to isolate the $\delta^{30}Si$ and Ge/Si signatures specific to kaolinite. The $\delta^{30}Si$ values for the secondary clays were -1.31 ± 0.04‰ for the 240 soil profile clay sample and -1.29 ± 685 0.04‰ for the 288 soil profile clay sample, with corresponding Ge/Si ratios of 4.92 and 8.36, respectively.

**Table B1 Mineral fractions in the clay fraction as determined by the inversion model. Corrected Si and Al concentrations, along with $\delta^{30}Si$ and Ge/Si ratios, are shown to reflect only the secondary mineral content, specifically kaolinite.**

| | \multicolumn{4}{c}{Fraction of each mineral from inversion model} | | | |
|---|---|---|---|---|
| samples | Quartz | Muscovite | Kaolinite | Microcline |
| | $SiO_2$ | $KAl_2(Si_3Al)O_{10}(OH)_2$ | $Al_2Si_2O_5(OH)_4$ | $KAlSi_3O_8$ |
| 240 clay | 0.21 | 0.19 | 0.55 | 0.05 |
| 288 clay | 0.30 | 0.05 | 0.50 | 0.15 |

| \multicolumn{7}{c}{Clay fraction chemistry} | | | | | | |
|---|---|---|---|---|---|
| | Al (ppm) | Si (ppm) | K (ppm) | Si/Al ($\mu$mol L$^{-1}$/$\mu$mol L$^{-1}$) | $\delta^{30}Si$ (‰) | Ge/Si ($\mu$mol mol$^{-1}$) |
| 240 clay | 119,886 | 151,749 | 24,226 | 1.18 | -0.78± 0.07 | 3.31 |
| 288 clay | 124,435 | 193,524 | 27,747 | 1.45 | -0.71 ± 0.07 | 4.85 |

| \multicolumn{7}{c}{Correction, concentration and signatures associated to kaolinite} | | | | | | |
|---|---|---|---|---|---|
| | Al (ppm) | Si (ppm) | K (ppm) | Si/Al ($\mu$mol L$^{-1}$/$\mu$mol L$^{-1}$) | $\delta^{30}Si$ (‰) | Ge/Si ($\mu$mol mol$^{-1}$) |
| 240 clay | 30,063 | 22,887 | - | 0.71 | -1.31 ± 0.04 | 4.92 |
| 288 clay | 11,043 | 7,554 | - | 0.64 | -1.29 ± 0.04 | 8.36 |



**Appendix C**

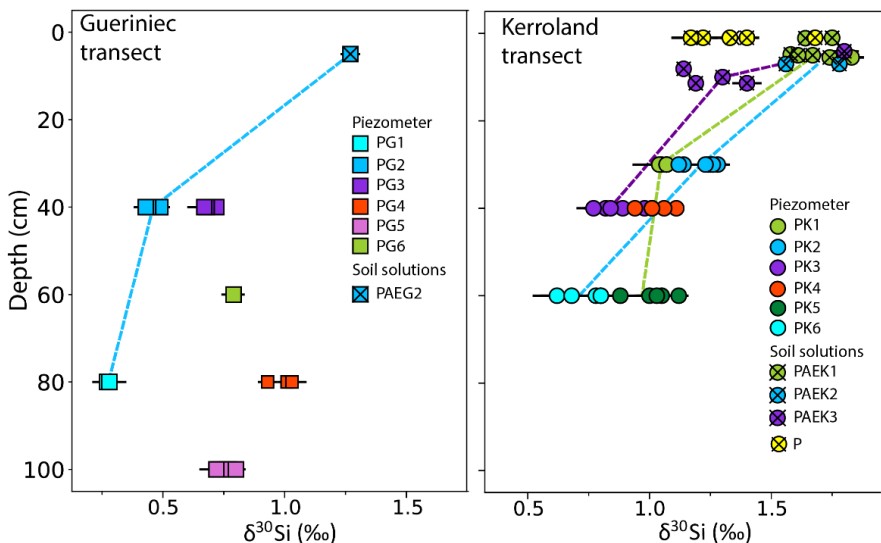

690

**Figure C1. Subsurface water δ³⁰Si signature vs. sampling depth for the Gueriniec (left) and Kerroland (right) transects. The colored lines join the samples collected at the same location but at different depths.**





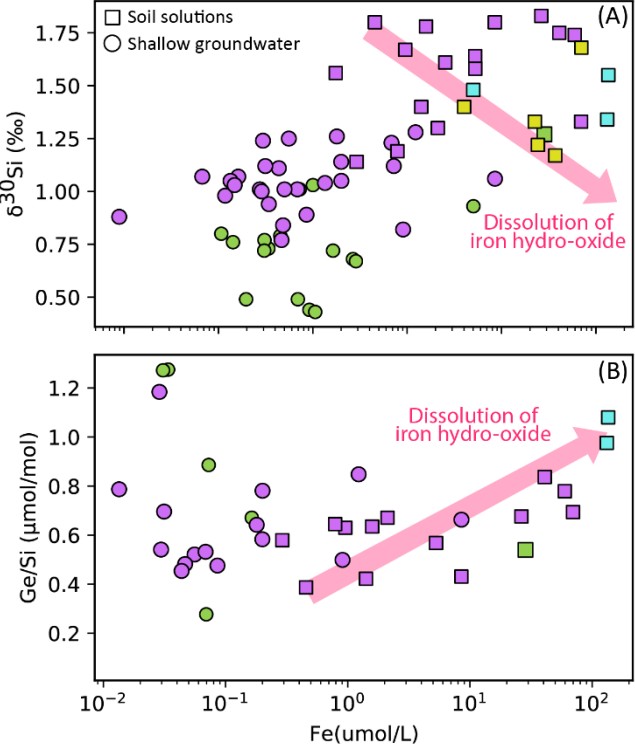

**Figure C2. Cross plots of dissolved (A) δ³⁰Si and Fe concentration and (B) Ge/Si and Fe concentration. Arrows illustrate the trends expected for iron oxyhydroxide precipitation.**

## Appendix D: Monte Carlo simulation of Si export in biomass

To solve Eq. (1), Eq. (2), and Eq. (3), we expressed them in matrix form and used Gaussian elimination to calculate $w_{iso}^{Si}$, $e_{clay-iso}^{Si}$ and $e_{org-iso}^{Si}$. To account for uncertainties in the measured Si isotopes and Ge/Si ratios, we applied a Monte Carlo approach across three distinct scenarios. Normal distributions were assigned in all scenarios using the mean and standard deviations of the Ge/Si ratios and δ³⁰Si measured in secondary clays ($n$ =2), bedrock ($n$ = 1) and stream samples($n$ = 8 for δ³⁰Si and $n$ = 5 for Ge/Si ratios), except for the leaf samples (Table 1).

In Scenario 1, we applied a uniform distribution to $\delta^{30}Si_{org}$ and $\left(\frac{Ge}{Si}\right)_{org}$, using the minimum and maximum isotopic signatures from plant samples ($n$ = 4) to capture the high variability in the plant data. In scenario 2, we used a normal distribution based solely on wheat leaves ($n$ = 2), while in Scenario 3 we applied parameters obtained from measurements on maize ($n$ = 2).

Model solutions where $w_{iso}^{Si}$, $e_{clay-iso}^{Si}$ and $e_{org-iso}^{Si}$ exceeded 1 or were less than 0 were excluded. Additionally, iterations with negatives Ge/Si ratios or secondary clay Ge/Si values below 3.31 (the minimum measured clay fraction) were also




removed. Out of the 6 million iterations, only 78,658 were valid for Scenario 1, 13,524 for Scenario 2, and 1,541,432 for Scenario 3.

## Appendix E: Estimation of riverine Si concentration and fluxes

To determine the export of dissolved Si ($W^{Si}$) by the stream, we used Si concentration data alongside discharge measurements at the catchment outlet. The discharge data, spanning from 2000 to 2023, enabled the estimation of annual fluxes over 22 years. This data set is available in https://agrhys.fr/BVE/vidae/, recorded in liter per second (L s$^{-1}$), which we have recalculated to liter per hour (L h$^{-1}$). During this period, there were 536 days with missing discharge data, averaging 24 days per year, with a peak of 59 total missing days in one year. This absence of data could potentially lead to an underestimation of Si fluxes.

Daily Si concentration measurements are available from fall 1999 until summer 2000, with additional samples collected in 2003, 2006, 2007, and during three flood events in 2021 and 2022 at an hourly resolution. This dataset includes 844 samples, covering various seasons and hydrological conditions (Table 4; López-Urzúa et al., 2025). Daily samples (from 1999 to 2007) were collected during the daytime hours of 15:00 to 18:00, and the corresponding discharge values were calculated as the average discharge during these hours.

To estimate Si concentrations over the 22 years with discharge data, we fitted a linear relationship between Si concentration [Si] and discharge (Q) in logarithmic space. Assuming a 5% independent error for both Si concentration (µmol L$^{-1}$) and discharge (L s$^{-1}$) measurements, we utilized IsoplotR's maximum likelihood estimator (Vermeesch, 2018) to determine the power law-equation [Si]=k*Q$^b$ describing the relationship between Si and Q yielding b being -0.09 ± 0.014 (SE, standard error) and the constant k being 195.9 ± 1.06 SE.

We estimate the Si concentration and associated standard error for each hour using the following criteria: if Q (L s$^{-1}$) < 3: [Si] = 154.77 (µmol L$^{-1}$); otherwise, [Si]= 195.9*Q$^{-0.09}$. This approach is used because, at very low Q values, the calculation can yield unrealistically high Si concentrations that have never been measured in the stream. Once the Si concentration was estimated for each hour, we calculate the annual the Si flux $W_{river}^{Si}$ and its associated error for each hydrological year from 2000 to 2022 (Table 5; López-Urzúa et al., 2025). Then to calculate $w_{river}^{Si}$ we divided $W_{river}^{Si}$ in mol m$^{-2}$ yr$^{-1}$ by the Si concentration of the bedrock multiplied by denudation.

## Appendix F: Derivation of equations for determining $h^{Si}_{regolith}$

The isotopic composition of each compartment within the Critical Zone can be related to elemental fluxes and isotope fractionation factors (Bouchez et al., 2013). Because at Kervidy-Naizin a portion of organic particulate erosion originates from harvesting, we reformulate Eq. (5e) from (Bouchez et al., 2013) as follows:



$$\delta^{30}Si_{diss} = \delta^{30}Si_{rock} \; - \frac{(E_{sec}^{Si}*\varepsilon_{prec}^{Si} + E_{org}^{Si}*\varepsilon_{upt}^{Si} + H_{regolith}^{Si}*\varepsilon_{upt}^{Si})}{S_{rock}^{Si}+S_{prim}^{Si}} \tag{F1}$$

Here, $H_{regolith}^{Si}$ represents the flux of Si harvested, determined as a function of Si fluxes and isotope fractionation factors for secondary mineral precipitation ($\varepsilon_{prec}^{Si}$) and plant uptake ($\varepsilon_{upt}^{Si}$). The terms $S_{rock}^{Si}$ and $S_{prim}^{Si}$ represents the flux of Si released during primary mineral dissolution prior to secondary mineral formation, occurring at the weathering front and in the regolith, respectively, expressed in mol m$^{-2}$ yr$^{-1}$. This dissolution is assumed to be congruent. These terms can be calculated using Eq. (15) from Bouchez et al. (2013):

$$\frac{S_{rock}^{Si}+S_{prim}^{Si}}{D*[Si]_{rock}} = -\tau_{prim}^{Si} = 1 - \frac{[Si]_{prim/reg}*[Ti]_{rock}}{[Si]_{prim/rock}*[Ti]_{reg}} \tag{F2}$$

$\tau_{prim}^{Si}$ represents the loss of Si through the primary mineral dissolution between rock and erodible material. This is distinct from the definition of $\tau$ by Chadwick et al. (1990), which is a net loss and thus includes the effects of both the dissolution of primary minerals and the precipitation of secondary minerals. By combining Eq. (A3) and (A4), and normalizing to the total Si flux, we can determine the fraction of Si harvested ($h_{regolith}^{Si}$) as:

$$h_{regolith}^{Si} = \frac{(\delta^{30}Si_{diss} - \delta^{30}Si_{rock})*\tau_{prim}^{Si} - e_{sec}^{Si}*\varepsilon_{prec}^{Si} - e_{org}^{Si}*\varepsilon_{upt}^{Si}}{\varepsilon_{upt}^{Si}} \tag{F3}$$

*Data availability.* Supplementary tables are available in the Supplement and the dataset including the hydrological data and geochemistry of water, soil, bedrock, and plants are available at https://doi.org/10.5281/zenodo.14615156 (López-Urzúa et al., 2025).

*Author contributions.* Sofía López-Urzúa: Conceptualization, data curation, formal analysis, methodology, investigation, Visualization, Writing – original draft. Louis Derry: Formal analysis, methodology, investigation, Supervision, Funding acquisition, writing – review and editing. Julien Bouchez: Formal analysis, methodology, investigation, writing – review and editing.

*Competing interests.* The authors declare that they have no conflict of interest.

*Acknowledgements.* We extend our sincere gratitude to Pierre Burckel for measuring trace elements and Si concentrations by ICP-MS Agilent 7900, Caroline Gorge for anion analysis by ion chromatography, and Sophie Nowak for XRD and FRX analyses. Special thanks to Johanne Lebrun Thauront and Samuel Abiven for providing the soil samples and extraction experiments data. We are particularly grateful to Ophélie Fovet, Laurent Jeanneau, Yannick Hamon, Mikaël Faucheux, and Patrice Petitjean from the ORE ArgHys Critical Zone Observatory for their exceptional dedication and support. Their



contributions, including discharge data, stream chemistry data, plant and water samples, and invaluable field assistance, were crucial to the success of this project.

*Financial support.* This study was financially supported by the Agence Nationale de la Recherche (ANR) through the 770 "Investissements d'avenir" program under the project CZTOP (ANR-17-MPGA-0009). Geochemical analyses presented here were supported by the IPGP multidisciplinary program PARI and the Île-de-France region SESAME grant no. 12015908.

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
