# Peer review of "Quantifying the agricultural footprint on the silicon cycle: Insights from silicon isotopes and Ge/Si ratios"

_EGUsphere, 2025_

## Author Comment (AC1)

Reviewer 1:

Review of manuscript egusphere-2025-78 submitted to EGUsphere by Sofía López-Urzúa and colleagues: Quantifying the agricultural footprint on the silicon cycle: Insights from silicon isotopes and Ge/Si ratios

With apologies to the authors and editor for this late review.

López-Urzúa and colleagues present the results of a comprehensive Si (isotope) budget for a small agricultural budget in France. Using different mass-balance approaches to quantify the amount of Si exported from the catchment in harvested crops, they find that it exceeds by a large amount the export of dissolved Si in streamwater, providing a demonstration of anthropogenic impacts on catchment Si cycling.

In general, I find this a solid manuscript worthy of publication after minor revisions. It is well written with clear figures and appropriate referencing, and deals with a topic that I think will be interesting to many in the community. The methods used are appropriate and the data seem of good quality. I have some suggestions or questions the authors may wish to consider in a revised version of the manuscript, that I detail in rough order of appearance.

**Response:** We thank Reviewer 1 for their thoughtful and constructive review. We greatly appreciate this recognition of the manuscript's overall quality, and we welcome the opportunity to improve our work further.

The two main concerns raised were:

- Inconsistencies across the manuscript regarding the assumption of congruent versus incongruent dissolution.

- The assumption that all primary minerals share the same Si isotope and Ge/Si signatures.

To address these points, we have revised the manuscript to consistently assume incongruent bedrock dissolution, considering quartz as inert. This change required correcting the Si isotope composition ($\delta^{30}$Si) and Ge/Si ratio of the initial solution after mineral dissolution, which in turn necessitated the use of mineral-specific signatures. Because the bedrock in our catchment consists of fine-grained siltstone, it was not possible to physically separate and analyze individual minerals to determine their specific signatures. Therefore, we compiled a mineral-specific dataset of $\delta^{30}$Si and Ge/Si values from the literature, that will be included in the Supplementary Material. These mineral-specific signatures were also used to correct the signatures of secondary clays minerals in the clay-sized fraction and were consistently applied throughout both the elemental and isotopic mass balance calculations. To assess the impact of estimates of mineral-specific isotope signatures on our flux calculations, we conducted a sensitivity analysis, that will be included in the appendix.

The revised results show that, although we now obtain a slightly lower fraction of Si exported via harvesting using the isotope-based mass balance model and a similar one using the regolith-based mass balance model , our main conclusion remains unchanged: Si export through crop harvesting continues to be an important flux—exceeding the dissolved Si flux and comparable to the erosion flux—underscoring the strong anthropogenic imprint on the silicon cycle in agricultural catchments.

Below, we provide point-by-point responses to the reviewer's detailed comments and clarify our methodology and interpretations accordingly.

Perhaps the weakest part of the dataset – as acknowledged by the authors (e.g. around L595) is the small number of total plant and clay samples, and that they are limited to only the leaves and not the full plant biomass. Much of the data interpretation relies on the plant and clay Si isotope fractionations/differences between fractionations for the difference species, but I feel these are not so well constrained. If there is the possibility to provide more data here this would greatly help strengthen the paper.

**Response:** We appreciate the reviewer's thoughtful comment and fully acknowledge the limitations of our dataset, particularly the small number of leaf (n = 4) and clay (n = 2) samples, as noted in the manuscript. Despite these constraints, we have taken several measures to integrate uncertainty and ensure our interpretations remain robust, particularly with regard to plant Si isotope signatures.

- Using Method 1 (section 5.2: Determining Si export from the catchment as biomass using an isotopic mass balance of Ge/Si and δ³⁰Si: $e_{org}^{Si}$), we implemented a conservative approach in Scenario 1 by applying a uniform distribution across the full range of measured plant δ³⁰Si values (−0.40‰ to +0.83‰) as an input for our Monte Carlo estimates of uncertainty. This ensures that the modeled outcomes reflect the full range of natural variability observed in our dataset.
- Using Method 3 (section 5.5: quantifying Si export through harvesting using Si loss indexes and isotopic fractionation: $h_{regolith}^{Si}$), we calculated the isotopic fractionation for each type of leaf sample. These are consistent with published values in literature. To incorporate uncertainty, we again applied a uniform distribution, assigning equal probability to the full range of observed fractionation values among species.

While our analysis includes only two maize and two wheat samples, these represent the dominant crops grown in the catchment. Moreover, Si isotopic fractionation is more strongly influenced by plant functional type than by intraspecific variability (Frick et al., 2020). Thus, we believe the selected species capture the relevant functional variability for the purpose of this study.

Regarding the clay samples, we analyzed two samples from contrasting soil conditions—one from a hydromorphic soil and one from a well-drained soil. Both exhibited very similar δ³⁰Si signatures, suggesting that the isotopic composition of secondary clays remains relatively consistent across variable soil environments in the studied catchment. This observation supports our use of a representative value for clays contributing to the erosion flux.

We agree that expanding the dataset—especially with more comprehensive plant and soil sampling—would further strengthen our conclusions. Nonetheless, we are confident that our conservative modeling strategy, use of scenario testing, and reliance on published values for comparison provide robust and defensible constraints on Si isotope behavior and fluxes in this catchment. In the revised manuscript, we will include a brief discussion before introducing the scenarios to clarify that the modeled outcomes capture the full range of natural variability observed in our dataset.

Related – in some cases the uncertainty propagation seems unrealistically small, in particular for the clay fractionation (Table 4 gives it as ±0.07‰; presumably 1sd?), but I can't make this fit with the data from table 1. Also, an uncertainty of only 0.01‰ is used for the secondary clay itself, but this is after a series of corrections for the 'contamination' of the clay size fraction with primary minerals. How is it possible that this correction process (detailed in appendix B) results in a narrower uncertainty? And is it justifiable that a single clay sample taken at ca. 60cm depth (Fig. 2) is representative of the clay that will eventually be eroded?

**Response:** We thank the reviewer for pointing out the issue with the uncertainty propagation in Table 4. There was an error in the original calculation. As pointed out earlier, in the revised manuscript we will apply mineral-specific signatures to correct the clay-sized fraction from the contribution of primary minerals, which will result in the updated values in Table 4. Additionally, to better capture uncertainty in this correction process, we revised our approach as follows:

- We first estimated the mineralogical composition of the clay-sized fraction through a mixing model and implemented a Monte Carlo simulation to propagate uncertainties. We assumed normal distributions for Si, Al, and K concentrations with a standard deviation of 5%, consistent with the long-term analytical precision of elemental analysis (<5%).
- We then corrected Si and Al concentrations to account for kaolinite contributions through Gaussian uncertainty propagation.

- Finally, for δ³⁰Si and Ge/Si ratio corrections, we performed a second Monte Carlo simulation using mineral-specific δ³⁰Si and Ge/Si values compiled from the literature for primary minerals (quartz, muscovite, and microcline). This updated procedure significantly larger and more realistic uncertainties (e.g., ±0.17‰ for δ³⁰Si and ±0.74 for Ge/Si) compared to the previously underestimated ±0.04‰:

| Sample | δ³⁰Si (‰) | Ge/Si (µmol mol-1) |
|---|---|---|
| 240 clay | -1.20 ± 0.16 | 4.78 ± 0.41 |
| 288 clay | -1.25 ± 0.17 | 8.31 ± 0.74 |

Regarding Table 1, the previously reported uncertainty of ±0.01‰ for secondary values is now updated to ±0.04‰. This uncertainty corresponds to the standard deviation of the mean calculated from the two corrected δ³⁰Si values (−1.20 ± 0.16‰ and −1.25 ± 0.17‰) but it does not incorporate the uncertainties from the correction procedures applied in Appendix B. To avoid confusion, we have added a footnote in Table 1 to clarify that the reported uncertainty represents the descriptive variation (mean ± SD) between corrected values, not the total analytical or propagated uncertainty used in modeling.

Concerning the representativeness of the clay sample taken at ~60 cm depth: we analyzed two clay samples from contrasting pedological conditions—one from a hydromorphic soil and one from a well-drained soil. Both yielded very similar δ³⁰Si values, suggesting minimal isotopic variation in secondary clays across soil types and depths. Moreover, the mineralogical and geochemical profiles of the sampled soils are relatively homogeneous, supporting the assumption that these clays are representative of those being mobilized via erosion from more superficial horizons. While we agree that expanding the dataset would be ideal, we are confident that the revised methodology and expanded uncertainty analysis now provide a more accurate and defensible treatment of the clay corrections and their influence on the mass balance model.

The mass-balances approaches detailed here explicitly or implicitly require steady-state, but I wonder how justifiable that is for this heavily anthropgenised catchment. E.g. the Clymans et al. reference that is cited details how the soil pools of Si change over decadal to centennial tiemscales in response to land cover change. This is a bit of an easy criticism to make but perhaps some discussion on how transient increases or decreases in the size of internal soil pools of Si (phytoliths, amorphous Si, clays, …) might impact the interpretation would be warranted?

**Response:** Thank you for raising this point. We acknowledge that strict steady-state conditions may not always hold, particularly in a catchment with significant anthropogenic influence. Transient changes in soil Si reservoirs —such as phytoliths, amorphous silica, and clays—could indeed influence short-term mass-balance interpretations. However, we argue that a quasi-steady-state framework is appropriate for our study for several reasons:

1) Temporal integration and isotopic consistency of dissolved Si export across multi-year time scales: Our silicon isotope dataset spans multiple hydrological years and seasons (2015, 2017, 2021, 2022, 2023; see Zenodo dataset). Despite differences in sampling time and discharge conditions, δ³⁰Si values in soil solution, groundwater, and river water remain consistent in both absolute values and isotopic trends. This temporal coherence indicates that our estimates of the Si element and isotope dissolved efflux averages over short-term (i.e., seasonal and inter-annual) represents a steady state. Our reasoning aligns with that of Bouchez et al. (2013), who argue that time-integrated datasets (e.g., sediment depth profiles, multi-year average fluxes) provide a solid basis for applying a quasi-steady-state assumption.

2) Long-term (multi-decadal) changes in Si content and isotope composition in secondary phases: Over long time scales, the additional removal of Si through harvesting will likely progressively deplete all "solid" compartments of the Critical Zone in Si, in particular in the lightest Si isotopes. As a result, today's soil solutions are isotopically heavier than in the past. However, solid phases such as clay minerals and organic matter formed partly under pre-agriculture conditions still retain an "inherited", lighter isotopic composition. Consequently, clay minerals sampled today as a whole integrate inherited Si isotope composition that is lighter than what would be expected if they had formed solely from present-day water-rock

interactions. A similar reasoning might apply to soil organic matter. Note that it does not apply to the plant pool, which in this environment is harvested yearly, leaving the possibility for this pool to "reset" its isotopic composition as time passes and as water itself evolves. If this legacy effect were fully corrected for (i.e., if solids were in equilibrium with today's heavier waters), the isotopic signature of soil solutions would likely be even heavier, further reinforcing our conclusion that harvesting is a major driver of light Si isotope export.

In light of these points, we recognize that the system may not be in perfect steady state, but we consider a quasi-steady-state approach both justifiable and robust for our mass-balance interpretation. In the revised manuscript, we will include a brief discussion before Section 5.2 to explicitly acknowledge these limitations and clarify the assumptions underlying our approach.

Regarding the vertical gradients in [Si] and d30Si, there doesn't seem to be much discussion of a simple mixing between Si-deplete rainwater and Si-rich 'weathering' water. Could this be part of the interpretation?

**Response:** While mixing between Si-depleted rainwater and Si-rich weathering-derived water could explain some of the vertical gradients in [Si], it cannot fully account for the observed $\delta^{30}Si$ signatures. To our knowledge, no silicon isotope values for rainwater have been reported in this region. If mixing were the dominant process, it would require rainwater to have a significantly heavier $\delta^{30}Si$ signature to explain the enrichment in soil solutions. Instead, the observed $\delta^{30}Si$ enrichment is best explained by biological processes, particularly plant uptake and subsequent harvesting, which preferentially remove lighter Si isotopes from the system. We will add a sentence in the manuscript to clearly explain that rain water cannot be a significant Si input to the soil system and thus cannot contribute to the establishment of the Si concentration and isotope composition profiles.

The authors assume that the bedrock is dissolving congruently (e.g. L311, but somewhat contradicted on L541), and that all primary minerals have the same Si isotope signature (e.g. Appendix B, L682). But how justifiable are these assumptions? A growing body of work demonstrates that minerals have specific d30Si signatures. Probably of minor importance here, but perhaps worth considering.

**Response:** Thank you for highlighting this important point. As noted earlier, we acknowledge an inconsistency in our original approach. In Section 5.1 (L311), we initially assumed congruent bedrock dissolution—i.e., that the water released reflects the bulk rock Si/Al ratio (7.54), $\delta^{30}Si$ (−0.13 ± 0.05‰), and Ge/Si ratio (1.33 µmol mol⁻¹). However, in our calculation of $h_{regolith}^{Si}$ we treated quartz as inert, implying incongruent dissolution.

To address this, we revised the manuscript to consistently reflect incongruent bedrock dissolution, assuming that only muscovite, albite, and chamosite actively dissolve. We also now consider mineral-specific $\delta^{30}Si$ and Ge/Si signatures, based on a literature compilation:

| Mineral | $\delta^{30}Si$ (‰) | Comments | Ge/Si (µmol mol⁻¹) | comments |
|---|---|---|---|---|
| Quartz | -0.06 ± 0.10 n=11 | Mean of data heavier than -0.13 (bedrock) | 0.72 ± 0.30 n=5 | Mean of all quartz values |
| Microcline and albite | -0.29 ± 0.14 n=17 | Mean for feldspars and plagioclase | 2.55 ± 0.77 n=8 | Mean for feldspars and plagioclase |
| Muscovite | -0.49 ± 0.11 | Estimated as mean between biotite and feldspar | 2.13 ± 0.21 n=3 | Mean of available values |
| Chamosite | -0.68 ± 0.18 n=6 | Estimated using biotite values | 4.55 ± 1.24 n= 6 | Estimated using biotite values |

For $\delta^{30}Si$, we selected only quartz data with heavier values than the bedrock average (−0.13 ± 0.05‰), since including lighter quartz values would not reproduce the bulk bedrock signature, given that other minerals exhibit even lower $\delta^{30}Si$. This "filtered" literature compilation yielded a

quartz δ³⁰Si of –0.06 ± 0.10‰. For microcline and albite, we adopted the average value reported for feldspars and plagioclases (–0.29 ± 0.14‰). For chamosite, a member of the chlorite group, we used values from biotite (–0.68 ± 0.18‰), and for muscovite, where direct data are lacking, we applied the average between biotite and feldspar (–0.49 ± 0.11‰), based on known trends relating δ³⁰Si fractionation to polymerization (Douthitt, 1982; Savage et al., 2014) and interlayer cation effects (Méheut et al., 2009; Méheut and Schauble, 2014).

For Ge/Si ratios, we used 0.72 ± 0.30 µmol mol⁻¹ for quartz, 2.55 ± 0.77 µmol mol⁻¹ for albite and microcline, 2.13 ± 0.21 µmol mol⁻¹ for muscovite, and 4.55 ± 1.24 µmol mol⁻¹ for chamosite.

We acknowledge that there are uncertainties in the mineral-specific δ³⁰Si and Ge/Si values, and that we rely on literature data compiled from other sites. However, this is currently the best approach available, as we only have one bulk bedrock sample and mineral-specific data are scarce. These kinds of assumptions are common in geochemical modeling, and we have taken care to test that our main outcomes are not overly sensitive to these choices.

Using these revised values, we recalculated the composition of the dissolving fluid, obtaining a Si/Al = 1.8, δ³⁰Si = –0.44 ± 0.08‰ and Ge/Si = 2.8 µmol mol⁻¹. These corrections do not affect the main trends or interpretations of our δ³⁰Si and Ge/Si dynamics in the catchment.

There are three different approaches applied here: 1) a d30Si+Ge/Si mass balance, 2) a mass balance based on river Si fluxes, and 3) a mass balance based on soil geochemistry. Although they are designed to predict slightly different aspects of Si export, I was surprised not to see a more explicit comparison (e.g. in a table or a figure).

**Response:** We appreciate the reviewer's suggestion to more explicitly compare the three different approaches used to estimate Si export. We agree that a clearer side-by-side comparison would strengthen the manuscript. To address this, we will move Tables 2 and 5 to the appendix and include a new summary table in the main text that directly compares the estimates from all three approaches.

The fractional export value for e_Si in approach 2 (stream water + sediment based) is 0.36 (L498). As far as I understand, this includes E_org, E_sec and E_prim - but is this inconsistent with a bedrock dominated by quartz? (which they assume elsewhere to be inert, e.g. 541 – if quartz is not dissolving then a minimum value for e_Si would be the quartz fraction of the bedrock)

**Response:** We thank the reviewer for this thoughtful observation. It is correct that if quartz is assumed to be inert, the minimum theoretical value for $e_{river}^{Si}$ should reflect the quartz fraction of the bedrock. Given that quartz comprises approximately 62% of the bedrock, this suggests a lower bound for $e_{river}^{Si}$ around 0.62.

The lower value we obtained (0.36) using the stream water + sediment approach likely reflects methodological limitations. Our "gauging" estimate is based on turbidity-derived suspended sediment concentrations, which primarily capture the fine sediment fraction. Coarse and dense minerals such as quartz are more likely to be transported as bedload and are thus underrepresented in turbidity-based estimates. This grain-size bias in suspended sediment sampling has been well documented (Bouchez et al., 2011; Lupker et al., 2012). In addition, it is also possible that some portion of the quartz is retained within the soil profile. Quartz is resistant to weathering and may accumulate over time in the regolith, particularly if it is not being mobilized either in dissolved form or as suspended or bedload particles. This would further reduce the fraction of Si exported via the river system, relative to the bulk rock composition.

As a result, this part of our methodology likely underestimates the total solid-phase Si export from primary minerals. However, even assuming a more conservative value of $e_{river}^{Si}$ = 0.62—matching the quartz fraction of the bedrock—the amount of Si exported via harvesting ($h_{river}^{Si}$) remains larger than the dissolved Si flux, supporting our conclusion that agricultural harvesting is a major component of Si export in the system. We will include a discussion of this issue and the associated references in the revised manuscript.

Minor comments:

L56: Either more recent revisions of the Si budget (e.g. Treguer et al) and/or the 'original' river Si flux estimates (e.g. Dürr et al/Beusen et al) might be appropriate here.

**Response:** Thank you for the suggestion. We have added Dürr et al. (2011) and Beusen et al. (2009) for the values of dissolved Si (6.2 ± 1.8 Tmol Si yr-1), Frings et al. (2016) for the values of dissolvable amorphous silica (1.9 ± 1.0 Tmol Si yr-1), and Tréguer et al. (2021) for the most recent review. Additionally, we have removed the reference to groundwater to better align with the cited studies.

L84: To avoid overstating the novelty of this contribution, maybe already mention here that some previous work has identified that plant biomass as a whole doesn't seem to discriminate against Ge as much as the phytolith-based estimates cited here would suggest.

**Response:** We have modified the text to acknowledge this broader perspective "While some studies suggest that Ge is discriminated against Si during vascular plant uptake (Blecker et al., 2007; Delvigne et al., 2009; Derry et al., 2005; Lugolobi et al., 2010; Meek et al., 2016), other research indicates that plant biomass as a whole does not exhibit as strong a discrimination (Delvigne et al., 2009; Frings et al., 2021; Kaiser et al., 2020; Rains et al., 2006; Sparks et al., 2011)".

L158: if the bedrock comprises bedding of different lithologies, is this one sample enough to capture the heterogeneity? Even in plutonic rocks variability in 'immobile' element content can be large (which becomes important for e.g. the mass-balances and the 'tau' values later).

**Response:** We acknowledge that relying on a single bedrock sample is not ideal for capturing potential lithological heterogeneity. To address this limitation, we incorporated data from Denis and Dabard (1988), who compiled chemical analyses (n = 9) from the same geological unit that underlies our study basin. These samples, collected along a transect approximately 32 km southeast of our site, report $TiO_2$ concentrations with a mean of 0.71 ± 0.19 wt.% (SD), which closely matches the $TiO_2$ concentration measured in our own bedrock sample (0.74 ± 0.04 wt.%).

To better account for both natural variability and analytical uncertainty in the bedrock Ti concentration, we implemented a Monte Carlo simulation when calculating $\tau_{prim}$, $e_{sec}^{Si}$ and $e_{org}^{Si}$. Specifically, we modeled the Ti concentration in the bedrock as a normal distribution with a mean of 0.74 wt.% and a standard deviation of 0.19 wt.%, reflecting the variability reported by Denis and Dabard (1988). For soil samples, we assumed a 5% relative uncertainty in Ti concentration. This approach integrates both measurement uncertainty and spatial heterogeneity in schist bedrock, thereby improving the robustness of our mass-balance calculations.

Importantly, despite this more rigorous treatment of uncertainty, the resulting values are nearly identical to those previously reported. This is because $\tau_{prim}$, $e_{sec}^{Si}$ and $e_{org}^{Si}$ in Eq. 10 were already calculated using the mean values across all soil profile samples, which inherently incorporate substantial natural variability, some of it being most likely the result of variability in bedrock composition. The original values were $\tau_{prim}$ = –0.42 ± 0.15, $e_{sec}^{Si}$ = 0.088 ± 0.042, and $e_{org}^{Si}$ = 0.0064 ± 0.0033, while the updated Monte Carlo-derived estimates are $\tau_{prim}$ = –0.42 ± 0.15, $e_{sec}^{Si}$ = 0.088 ± 0.043, and $e_{org}^{Si}$ = 0.0065 ± 0.0035. These consistent results lend support to the robustness of our initial estimates and demonstrate that the updated method supports the same conclusions, while providing a more comprehensive representation of uncertainty.

L180: What is precision/long term reproducibility on the elemental data? Were any secondary reference materials included in the analyses?

**Response:** Yes, the river water standard SLRS-5 (National Research Council, Canada) was systematically analyzed, with a long-term analytical precision of better than 5%. This will be added in the method section

Fig 2: presumably cmbs on the y-axis, not mbs. Greek letter mu (not u) on Ge/Si x-axis.

**Response:** Thank you for pointing out this error. In the new manuscript we have corrected "mbs" to "cm b.s." on the y-axis and replaced "umol mol$^{-1}$" with "μmol mol$^{-1}$" on the Ge/Si x-axis.

L286: "compared"

**Response:** Thank you for pointing out this error. In the new manuscript we will correct this mistake.

345: 'show a positive correlation' / 'are positively correlated'

**Response:** Thank you for pointing out this error. In the new manuscript we will correct this mistake.

L415 – also Baronas et al 2020 GBC would be appropriate to cite here?

**Response:** Thank you for the suggestion. We have added Baronas et al. (2020) to acknowledge their findings that a significant portion of Si is taken up by vegetation fbioSi = 39 ± 14%, which aligns with the range reported by Frings et al. (2021). "These contributions are also higher than those observed in four other non-agricultural catchments, where $e_{org-iso}^{Si}$ ranged from 0.20 ± 0.1 to 0.42 ± 0.23 (Baronas et al., 2020; Frings et al., 2021)."

L483 – actually relatively high?

**Response:** We are not sure what the reviewer is referring to here — is it that the fraction of Si exported in the dissolved form (0.12 ± 0.06) is high compared to the value determined by the isotopic mass balance? If so, we underline that the isotopic mass balance approach yields values ranging from 0.15 ± 0.08 to 0.22 ± 0.11, which are within the uncertainties of the value obtained by the gauging method. Assuming this is what the reviewer is suggesting, we will add: "The mean fraction of Si exported from the catchment in the dissolved form is 0.12 ± 0.06, which is consistent with the values calculated using the isotopic mass balance when considering their respective uncertainties (0.15 ± 0.08 to 0.22 ± 0.11)."

L519: eSi_sec repeated here – presumably should be eSi_org?

**Response:** Thank you for catching this error. We have corrected $e_{sec}^{Si}$ to $e_{org}^{Si}$ in the revised manuscript.

L520: If this is a schist bedrock, how variable is the Ti content, and how are uncertainties propagated?

**Response:** We thank the reviewer for this related comment. As discussed in our response to Comment L158, we addressed the potential variability in Ti content within schist bedrock by incorporating external geochemical data from Denis and Dabard (1988), who reported $TiO_2$ concentrations (n = 9) with a mean of 0.71 ± 0.19 wt.% in the same geological unit underlying our study site. This variability was integrated into our calculations through a Monte Carlo simulation that models Ti concentration in the bedrock as a normal distribution (mean = 0.74 wt.%, SD = 0.19 wt.%) and includes a 5% relative uncertainty for the soil Ti concentrations. This framework allowed us to propagate uncertainties in Ti through all downstream calculations of $\tau_{prim}, e_{sec}^{Si}$ and $e_{org}^{Si}$, ultimately providing a more comprehensive assessment of uncertainty. As noted previously, this improved treatment does not significantly change the final estimates, reinforcing the robustness of our original results.

L530: What is the justification for using stream water rather than soil solutions to define e_prec?

**Response:** We chose to use stream water rather than soil solutions because it integrates contributions from the entire catchment, providing a more representative estimate of $\varepsilon_{prec}^{Si}$ at the catchment scale rather than reflecting localized variations. We will add a sentence justifying this choice.

L544: "to be inert"

**Response:** Thank you for correcting this error.

L567: Why are these values so low compared to previous two estimates?

**Response:** First, we would like to clarify that the values $e_{org}^{Si}$ and $e_{org-iso}^{Si}$ do not represent the same quantity. The term $e_{org-iso}^{Si}$ refers to the fraction of silicon that is both eroded naturally and exported through harvesting. In contrast, term $e_{org}^{Si}$ specifically represents the silicon associated with soil organic matter—i.e., the phytoliths that remain in the soil after harvesting. This distinction, which stems from the way the different mass balance equations are set up, explains why the value of $e_{org}^{Si}$ is lower than that of $e_{org-iso}^{Si}$. In the revised manuscript, we will rename $e_{org}^{Si}$ to make this distinction clearer and include a sentence to explicitly state the difference between the two terms.

Regarding $e_{clay-iso}^{Si}$ and $e_{sec}^{Si}$, these metrics are intended to represent the same process but determined with different approaches. In particular, estimating $e_{sec}^{Si}$ based on elemental metrics (eq. 15) is particularly challenging due to the complexity and spatial heterogeneity of soil processes. For instance, the preferential erosion of fine-grained, clay-rich material can remove a significant portion of the secondary Si pool from the soil. As $e_{sec}^{Si}$ is calculated using [Si]sec, reflecting the amount of soil Si contained in secondary minerals, any Si loss by clay erosion leads to an underestimation of [Si]sec. Consequently, the resulting $e_{sec}^{Si}$ value is biased toward lower estimated. We will include a discussion about these potential issues, acknowledging in particular that an ideal approach would require a more detailed characterization of the soil profiles and mineralogy.

L595: fractionation factors are not 'heavy' or 'light'; better to talk about magnitude. In general, fractionation factor normally refers to so-called 'alpha' notation, and just 'fractionation' alone to 'epsilon' notation – see Coplen 2011 DOI: 10.1002/rcm.5129.

**Response:** Thank you for providing clarification on the use of fractionation factor for alpha notation and isotopic fractionation for the epsilon notation. We have corrected to talk about the magnitude of the isotopic fractionation. Additionally, we have revised the sections 5.5.1 and 5.5.4 replacing the "fractionation factor" by "isotopic fractionation".

L634: See also Vandervenne et al 2013 Proc Royal Soc B.

**Response:** Thank you for the suggestion. We have now included Vandevenne et al. (2013) as a reference to highlight the role of grazing animals in accelerating the return of biogenic Si to the soil and enhancing its reactivity and dissolvability.

L708: Does the very low number of acceptable iterations (e.g. 0.2% for scenario 2) simply imply that an assumption underpinning the mass-balance or endmember assignments is incorrect?

**Response:** Thank you for your thoughtful comment. You are correct that the very low number of acceptable iterations in Scenario 2 (e.g., 0.2%) originally suggested a potential issue with our mass-balance assumptions or endmember assignments. In our initial approach, we assumed congruent dissolution of the bedrock. We have now revised the model to consistently reflect incongruent dissolution, assuming that only muscovite, albite, and chamosite actively contribute to weathering. This update includes the use of mineral-specific $\delta^{30}Si$ and Ge/Si signatures based on a literature compilation.

As a result, the $\delta^{30}Si$ value assigned to the dissolving rock has changed significantly, leading to a much higher number of valid iterations across all scenarios. Previously, out of 6 million iterations, only 78,658 were valid for Scenario 1, 13,524 for Scenario 2, and 1,541,432 for Scenario 3. In the revised model, approximately 590,000 iterations are now valid for Scenario 1, 464,000 for Scenario 2, and 1,603,000 for Scenario 3. These improvements result in a more robust and

internally consistent set of model outputs, indicating that mass balance is "more likely" to be achieved using these updated values for the solution produced by rock dissolution

**References:**

Baronas, J. J., West, A. J., Burton, K. W., Hammond, D. E., Opfergelt, S., Pogge Von Strandmann, P. A. E., James, R. H., and Rouxel, O. J.: Ge and Si Isotope Behavior During Intense Tropical Weathering and Ecosystem Cycling, Global Biogeochemical Cycles, 34, e2019GB006522, https://doi.org/10.1029/2019GB006522, 2020.

Beusen, A. H. W., Bouwman, A. F., Dürr, H. H., Dekkers, A. L. M., and Hartmann, J.: Global patterns of dissolved silica export to the coastal zone: Results from a spatially explicit global model, Global Biogeochemical Cycles, 23, 2008GB003281, https://doi.org/10.1029/2008GB003281, 2009.

Blecker, S. W., King, S. L., Derry, L. A., Chadwick, O. A., Ippolito, J. A., and Kelly, E. F.: The ratio of germanium to silicon in plant phytoliths: quantification of biological discrimination under controlled experimental conditions, Biogeochemistry, 86, 189–199, https://doi.org/10.1007/s10533-007-9154-7, 2007.

Bouchez, J., Gaillardet, J., France-Lanord, C., Maurice, L., and Dutra-Maia, P.: Grain size control of river suspended sediment geochemistry: Clues from Amazon River depth profiles, Geochem Geophys Geosyst, 12, 2010GC003380, https://doi.org/10.1029/2010GC003380, 2011.

Bouchez, J., Von Blanckenburg, F., and Schuessler, J. A.: Modeling novel stable isotope ratios in the weathering zone, American Journal of Science, 313, 267–308, https://doi.org/10.2475/04.2013.01, 2013.

Delvigne, C., Opfergelt, S., Cardinal, D., Delvaux, B., and André, L.: Distinct silicon and germanium pathways in the soil-plant system: Evidence from banana and horsetail: DISTINCT SI AND GE PATHWAYS IN PLANTS, J. Geophys. Res., 114, n/a-n/a, https://doi.org/10.1029/2008JG000899, 2009.

Denis, E. and Dabard, M. P.: Sandstone petrography and geochemistry of late proterozoic sediments of the armorican massif (France) — A key to basin development during the cadomian orogeny, Precambrian Research, 42, 189–206, https://doi.org/10.1016/0301-9268(88)90017-4, 1988.

Derry, L. A., Kurtz, A. C., Ziegler, K., and Chadwick, O. A.: Biological control of terrestrial silica cycling and export fluxes to watersheds, Nature, 433, 728–731, https://doi.org/10.1038/nature03299, 2005.

Douthitt, C. B.: The geochemistry of the stable isotopes of silicon, Geochimica et Cosmochimica Acta, 46, 1449–1458, https://doi.org/10.1016/0016-7037(82)90278-2, 1982.

Dürr, H. H., Meybeck, M., Hartmann, J., Laruelle, G. G., and Roubeix, V.: Global spatial distribution of natural riverine silica inputs to the coastal zone, Biogeosciences, 8, 597–620, https://doi.org/10.5194/bg-8-597-2011, 2011.

Frings, P. J., Clymans, W., Fontorbe, G., De La Rocha, C. L., and Conley, D. J.: The continental Si cycle and its impact on the ocean Si isotope budget, Chemical Geology, 425, 12–36, https://doi.org/10.1016/j.chemgeo.2016.01.020, 2016.

Frick, D. A., Remus, R., Sommer, M., Augustin, J., Kaczorek, D., and Von Blanckenburg, F.: Silicon uptake and isotope fractionation dynamics by crop species, Biogeosciences, 17, 6475–6490, https://doi.org/10.5194/bg-17-6475-2020, 2020.

Frings, P. J., Schubring, F., Oelze, M., and Von Blanckenburg, F.: Quantifying biotic and abiotic Si fluxes in the Critical Zone with Ge/Si ratios along a gradient of erosion rates, Am J Sci, 321, 1204–1245, https://doi.org/10.2475/08.2021.03, 2021.

Kaiser, S., Wagner, S., Moschner, C., Funke, C., and Wiche, O.: Accumulation of germanium (Ge) in plant tissues of grasses is not solely driven by its incorporation in phytoliths, Biogeochemistry, 148, 49–68, https://doi.org/10.1007/s10533-020-00646-x, 2020.

Lugolobi, F., Kurtz, A. C., and Derry, L. A.: Germanium–silicon fractionation in a tropical, granitic weathering environment, Geochimica et Cosmochimica Acta, 74, 1294–1308, https://doi.org/10.1016/j.gca.2009.11.027, 2010.

Lupker, M., France-Lanord, C., Galy, V., Lavé, J., Gaillardet, J., Gajurel, A. P., Guilmette, C., Rahman, M., Singh, S. K., and Sinha, R.: Predominant floodplain over mountain weathering of Himalayan sediments (Ganga basin), Geochimica et Cosmochimica Acta, 84, 410–432, https://doi.org/10.1016/j.gca.2012.02.001, 2012.

Meek, K., Derry, L., Sparks, J., and Cathles, L.: 87Sr/86Sr, Ca/Sr, and Ge/Si ratios as tracers of solute sources and biogeochemical cycling at a temperate forested shale catchment, central Pennsylvania, USA, Chemical Geology, 445, 84–102, https://doi.org/10.1016/j.chemgeo.2016.04.026, 2016.

Méheut, M. and Schauble, E. A.: Silicon isotope fractionation in silicate minerals: Insights from first-principles models of phyllosilicates, albite and pyrope, Geochimica et Cosmochimica Acta, 134, 137–154, https://doi.org/10.1016/j.gca.2014.02.014, 2014.

Méheut, M., Lazzeri, M., Balan, E., and Mauri, F.: Structural control over equilibrium silicon and oxygen isotopic fractionation: A first-principles density-functional theory study, Chemical Geology, 258, 28–37, https://doi.org/10.1016/j.chemgeo.2008.06.051, 2009.

Rains, D. W., Epstein, E., Zasoski, R. J., and Aslam, M.: Active Silicon Uptake by Wheat, Plant Soil, 280, 223–228, https://doi.org/10.1007/s11104-005-3082-x, 2006.

Savage, P. S., Armytage, R. M. G., Georg, R. B., and Halliday, A. N.: High temperature silicon isotope geochemistry, Lithos, 190–191, 500–519, https://doi.org/10.1016/j.lithos.2014.01.003, 2014.

Sparks, J. P., Chandra, S., Derry, L. A., Parthasarathy, M. V., Daugherty, C. S., and Griffin, R.: Subcellular localization of silicon and germanium in grass root and leaf tissues by SIMS: evidence for differential and active transport, Biogeochemistry, 104, 237–249, https://doi.org/10.1007/s10533-010-9498-2, 2011.

Tréguer, P. J., Sutton, J. N., Brzezinski, M., Charette, M. A., Devries, T., Dutkiewicz, S., Ehlert, C., Hawkings, J., Leynaert, A., Liu, S. M., Llopis Monferrer, N., López-Acosta, M., Maldonado, M., Rahman, S., Ran, L., and Rouxel, O.: Reviews and syntheses: The biogeochemical cycle of silicon in the modern ocean, Biogeosciences, 18, 1269–1289, https://doi.org/10.5194/bg-18-1269-2021, 2021.

Vandevenne, F. I., Barão, A. L., Schoelynck, J., Smis, A., Ryken, N., Van Damme, S., Meire, P., and Struyf, E.: Grazers: biocatalysts of terrestrial silica cycling, Proc. R. Soc. B., 280, 20132083, https://doi.org/10.1098/rspb.2013.2083, 2013.

---

## Author Comment (AC3)

**Final answer to the reviewers:**

We thank both reviewers for their insightful and constructive comments. We greatly appreciate your recognition of the manuscript's overall quality and welcome the opportunity to further improve our work.

Three main concerns were raised during the review process:

- Inconsistencies regarding the assumption of congruent versus incongruent dissolution (Reviewer 1).

- The assumption that all primary minerals share the same $\delta^{30}Si$ and Ge/Si signatures (Reviewer 1).

- Insufficient discussion and figure support for the results presented in Section 5.1 (Reviewer 2).

We have revised the manuscript to consistently assume incongruent dissolution, treating quartz as inert. This required correcting the Si isotope composition ($\delta^{30}Si$) and Ge/Si ratio of the solubilized solution following mineral dissolution and thus incorporating mineral-specific isotopic and elemental signatures.

Because the bedrock in our catchment is composed of fine-grained siltstone, we were unable to physically separate individual minerals for direct analysis. Therefore, we compiled a dataset of mineral-specific $\delta^{30}Si$ and Ge/Si values from the literature, which will be included in the Supplementary Material. These values were used to correct the isotopic signatures of secondary clays in the clay-sized fraction, to estimate the composition of the initial solution after rock dissolution and were consistently applied in both the elemental and isotopic mass balance calculations. We have also conducted a sensitivity analysis to evaluate how the selection of mineral signatures affects flux estimates, which will be provided in the Appendix.

While our primary objective is to quantify the agricultural footprint on the Si cycle, we agree with Reviewer 2 that Section 5.1 would benefit from further clarification. To maintain the overall conciseness of the main text, we will revise Section 5.1 to include additional context and precision regarding the Si isotopic signatures and their relevance. However, to avoid an overly lengthy manuscript, we will address the more detailed discussion—including pathways of plant uptake, isotopic fractionation, and the observed vertical gradients—in the Appendix. These aspects will be illustrated with the supporting figures (e.g., Appendix C, Fig. C1). Additionally Figure 3 will also be improved to include Critical Zone endmembers for clarity.

The revised manuscript shows that although the fraction of Si exported via harvesting has decreased in both the isotopic model and the riverine Si flux approach—and remained stable in the regolith-based approach—harvesting is no longer the dominant flux but now closely competes with erosion. Nonetheless, it still exceeds the dissolved Si flux and remains a key contributor to the Si cycle, highlighting the persistent and significant anthropogenic influence on the silicon budget in this agricultural catchment.

---

## Author Response (AR1)

**Reviewer 1:**

Review of manuscript egusphere-2025-78 submitted to EGUsphere by Sofía López-Urzúa and colleagues: Quantifying the agricultural footprint on the silicon cycle: Insights from silicon isotopes and Ge/Si ratios

With apologies to the authors and editor for this late review.

López-Urzúa and colleagues present the results of a comprehensive Si (isotope) budget for a small agricultural budget in France. Using different mass-balance approaches to quantify the amount of Si exported from the catchment in harvested crops, they find that it exceeds by a large amount the export of dissolved Si in streamwater, providing a demonstration of anthropogenic impacts on catchment Si cycling.

In general, I find this a solid manuscript worthy of publication after minor revisions. It is well written with clear figures and appropriate referencing, and deals with a topic that I think will be interesting to many in the community. The methods used are appropriate and the data seem of good quality. I have some suggestions or questions the authors may wish to consider in a revised version of the manuscript, that I detail in rough order of appearance.

**Response:** We thank Reviewer 1 for their thoughtful and constructive review. We greatly appreciate this recognition of the manuscript's overall quality, and we welcome the opportunity to improve our work further.

The two main concerns raised were:

- Inconsistencies across the manuscript regarding the assumption of congruent versus incongruent dissolution.
- The assumption that all primary minerals share the same Si isotope and Ge/Si signatures.

To address these points, we revised the manuscript to consistently assume incongruent bedrock dissolution, considering quartz as inert. This change required correction of the Si isotope composition ( $\delta^{30}$ Si) and Ge/Si ratio of the initial solution after mineral dissolution, which in turn necessitated the use of mineral-specific signatures. Because the bedrock in our catchment consists of fine-grained siltstone, it was not possible to physically separate and analyze individual minerals to determine their specific signatures. Therefore, we compiled a mineral-specific dataset of  $\delta^{30}$ Si and Ge/Si values from the literature, which we have now included in the Supplementary material. These mineral-specific signatures were also used to correct the signatures of secondary clays minerals in the clay-sized fraction and were applied consistently throughout both the elemental and isotopic mass balance calculations.

To assess the impact of estimates of mineral-specific isotope signatures on our flux calculations, we conducted a sensitivity analysis, now included in the appendix.

The revised results show that, although we now obtain a slightly lower fraction of Si exported via harvesting using the isotope-based mass balance model and a similar one using the regolith-based mass balance model, our main conclusion remains unchanged: Si export through crop harvesting continues to be an important flux—exceeding the dissolved Si flux and comparable to the erosion flux—underscoring the strong anthropogenic imprint on the silicon cycle in agricultural catchments.

Below, we provide point-by-point responses to the reviewer's detailed comments and clarify our methodology and interpretations accordingly.

Perhaps the weakest part of the dataset – as acknowledged by the authors (e.g. around L595) is the small number of total plant and clay samples, and that they are limited to only the leaves and not the full plant biomass. Much of the data interpretation relies on the plant and clay Si isotope fractionations/differences between fractionations for the difference species, but I feel these are not

so well constrained. If there is the possibility to provide more data here this would greatly help strengthen the paper.

**Response:** We appreciate the reviewer's thoughtful comment and fully acknowledge the limitations of our dataset, particularly the small number of leaf (n = 4) and clay (n = 2) samples, as noted in the manuscript. Despite these constraints, we took several measures to integrate uncertainty and ensure our interpretations remained robust, particularly with regard to plant Si isotope signatures.

- Using Method 1 (section 5.2: Determining Si export from the catchment as biomass using an isotopic mass balance of Ge/Si and  $\delta^{30}$ Si:  $e^{Si}_{org}$ ), we implemented a conservative approach in Scenario 1 by applying a uniform distribution across the full range of measured plant  $\delta^{30}$ Si values (-0.40% to +0.83%) as an input for our Monte Carlo estimates of uncertainty. This ensured that the modeled outcomes reflected the full range of natural variability observed in our dataset.
- Using Method 3 (section 5.5: quantifying Si export through harvesting using Si loss indexes and isotopic fractionation:  $h_{regolith}^{Si}$ ), we calculated the isotopic fractionation for each type of leaf sample. These were consistent with published values in literature. To incorporate uncertainty, we again applied a uniform distribution, assigning equal probability to the full range of observed fractionation values among species.

While our analysis included only two maize and two wheat samples, these represented the dominant crops grown in the catchment. Moreover, Si isotopic fractionation is more strongly influenced by plant functional type than by intraspecific variability (Frick et al., 2020). Thus, we believe the selected species captured the relevant functional variability for the purpose of this study.

Regarding the clay samples, we analyzed two samples from contrasting soil conditions—one from a hydromorphic soil and one from a well-drained soil. Both exhibited very similar  $\delta^{30}$ Si signatures, suggesting that the isotopic composition of secondary clays remains relatively consistent across variable soil environments in the studied catchment. This observation supported our use of a representative value for clays contributing to the erosion flux.

We agree that expanding the dataset—especially with more comprehensive plant and soil sampling—would further strengthen our conclusions. Nonetheless, we are confident that our conservative modeling strategy, use of scenario testing, and reliance on published values for comparison provided robust and defensible constraints on Si isotope behavior and fluxes in this catchment. In the revised manuscript, we included a brief discussion before introducing the scenarios in section 5.2 to clarify that the modeled outcomes captured the full range of natural variability observed in our dataset as well in section 5.5.4.

Changes: L429: "One limitation of our dataset is the small number of total plants (n= 4) and clay fractions (n=2) and that the samples are limited to only the leaves and not the full plant biomass. To integrate these uncertainties and ensure interpretations remain robust, particularly regarding plant Si isotope signatures, we implemented three scenarios. For each scenario, normal distributions were assigned to Ge/Si ratios and  $\delta^{30}Si$ , for secondary clays, bedrock and stream (Table 1). The organic fraction was treated differently across scenarios: in Scenario 1, we applied a uniform distribution based on the whole range of measured plant isotopic signatures to capture their high variability for the dominant crop species (maize and wheat); in Scenario 2, we used a normal distribution based on measurements made on wheat leaves only; and in Scenario 3, we applied a normal distribution derived from measurements on maize only (details in Appendix D). For the clay endmember, the consistent  $\delta^{30}Si$  values observed across contrasting soil types support the representativeness of our values. Although expanding the dataset would further improve these estimates, our modeling framework ensures that the results encompass the full range of plausible variability".

L650: "As previously discussed, (Section 5.2), while our dataset includes only two maize and two wheat samples, these represent the dominant crops cultivated in the catchment, and Si isotopic fractionation is generally more strongly influenced by plant functional type than by intraspecific

variability (Frick et al., 2020). We therefore consider that the selected species capture the relevant functional variability for this analysis."

Related – in some cases the uncertainty propagation seems unrealistically small, in particular for the clay fractionation (Table 4 gives it as ±0.07‰; presumably 1sd?), but I can't make this fit with the data from table 1. Also, an uncertainty of only 0.01‰ is used for the secondary clay itself, but this is after a series of corrections for the 'contamination' of the clay size fraction with primary minerals. How is it possible that this correction process (detailed in appendix B) results in a narrower uncertainty? And is it justifiable that a single clay sample taken at ca. 60cm depth (Fig. 2) is representative of the clay that will eventually be eroded?

**Response:** We thank the reviewer for pointing out the issue with the uncertainty propagation in Table 4. An error was present in the original calculation. As pointed out earlier, in the revised manuscript we applied mineral-specific signatures to correct the clay-sized fraction from the contribution of primary minerals, which resulted in the updated values in Table 1. Additionally, to better capture uncertainty in this correction process, we revised our approach as follows:

- We first estimated the mineralogical composition of the clay-sized fraction through a mixing model and implemented a Monte Carlo simulation to propagate uncertainties. We assumed normal distributions for Si, Al, and K concentrations with a standard deviation of 5%, consistent with the long-term analytical precision of elemental analysis (<5%).
- We then corrected Si and Al concentrations to account for kaolinite contributions, using Gaussian propagation in the estimation of the uncertainty resulting from this correction.
- Finally, for  $\delta^{30}$ Si and Ge/Si ratio corrections, we performed a second Monte Carlo simulation using mineral-specific  $\delta^{30}$ Si and Ge/Si values compiled from the literature for primary minerals (quartz, muscovite, and microcline). This updated procedure yielded significantly larger and more realistic uncertainties (e.g.,  $\pm 0.17\%$  for  $\delta^{30}$ Si and  $\pm 0.74$  for Ge/Si) compared to the previously underestimated  $\pm 0.04\%$ :

| Sample   | δ 30 Si (‰) | Ge/Si (µmol mol-1) |
|----------|------------------------|--------------------|
| 240 clay | -1.20 ± 0.16           | 4.78 ± 0.41        |
| 288 clay | -1.25 ± 0.17           | 8.31 ± 0.74        |

Regarding Table 1, the previously reported uncertainty of  $\pm 0.01\%$  for secondary values is now updated to  $\pm 0.04\%$ . This uncertainty corresponds to the standard deviation of the mean calculated from the two corrected  $\delta^{30}$ Si values ( $-1.20\pm0.16\%$  and  $-1.25\pm0.17\%$ ) but it does not incorporate the uncertainties from the correction procedures applied in Appendix B. To avoid confusion, we added a footnote in Table 1 to clarify that the reported uncertainty represents the descriptive variation (mean  $\pm$  SD) between corrected values, not the total analytical or propagated uncertainty used in modeling.

Concerning the representativeness of the clay sample taken at ~60 cm depth: we analyzed two clay samples from contrasting pedological conditions—one from a hydromorphic soil and one from a well-drained soil. Both yielded very similar  $\delta^{30}$ Si values, suggesting minimal isotopic variation in secondary clays across soil types and depths. Moreover, the mineralogical and geochemical profiles of the sampled soils are relatively homogeneous, supporting the assumption that these clays are representative of those being mobilized via erosion from more superficial horizons. While we agree that expanding the dataset would be ideal, we are confident that the revised methodology and expanded uncertainty analysis now provide a more accurate and defensible treatment of the clay corrections and their influence on the mass balance model.

**Changes:** Appendix B (L743): We have revised Appendix B to reflect the new methodology for calculating clay fraction signatures. A new Table B2 has been added, listing the selected  $\delta^{30}$ Si and Ge/Si values for each mineral present in the bedrock.

Table 1 (L255): The mean  $\delta^{30}$ Si and Ge/Si values for the clay fraction were updated to -1.23 ± 0.04 % and a value of 6.55 ± 2.50 µmol mol-1, respectively. A footnote was added to table 1: "(c) The lower SD arises from nearly identical values; however, each individual measurement has an uncertainty of ±0.16."

The mass-balances approaches detailed here explicitly or implicitly require steady-state, but I wonder how justifiable that is for this heavily anthropgenised catchment. E.g. the Clymans et al. reference that is cited details how the soil pools of Si change over decadal to centennial tiemscales in response to land cover change. This is a bit of an easy criticism to make but perhaps some discussion on how transient increases or decreases in the size of internal soil pools of Si (phytoliths, amorphous Si, clays, ...) might impact the interpretation would be warranted?

**Response:** Thank you for raising this point. We acknowledge that strict steady-state conditions may not always hold, particularly in a catchment with significant anthropogenic influence. Transient changes in soil Si reservoirs —such as phytoliths, amorphous silica, and clays—could indeed influence short-term mass-balance interpretations. However, we argue that a quasi-steady-state framework is appropriate for our study for several reasons:

- 1) Temporal integration and isotopic consistency of dissolved Si export across multi-year time scales: Our silicon isotope dataset spans multiple hydrological years and seasons (2015, 2017, 2021, 2022, 2023; see Zenodo dataset). Despite differences in sampling time and discharge conditions,  $\delta^{30}$ Si values in soil solution, groundwater, and river water remain consistent in both absolute values and isotopic trends. This temporal coherence indicates that our estimates of the Si element and isotope dissolved efflux averages over short-term (i.e., seasonal and interannual) represents a steady state. Our reasoning aligns with that of Bouchez et al. (2013), who argue that time-integrated datasets (e.g., sediment depth profiles, multi-year average fluxes) provide a solid basis for applying a quasi-steady-state assumption in isotope mass balance approaches.
- 2) Long-term (multi-decadal) changes in Si content and isotope composition in secondary phases: Over long time scales, the additional removal of Si through harvesting will likely progressively deplete all "solid" compartments of the Critical Zone in Si, in particular in the lightest Si isotopes. As a result, in an agricultural catchment today's soil solutions are most likely isotopically heavier than in the past. However, modern soil still contain solid phases such as clay minerals and organic matter formed partly under pre-agriculture conditions still retain an "inherited", lighter isotopic composition. Consequently, clay minerals sampled today as a whole integrate inherited Si isotope composition that is lighter than what would be expected if they had formed solely from present-day (i.e., influenced by agriculture) water-rock interactions. A similar reasoning might apply to soil organic matter. Note that this reasoning does not readily apply to the plant pool, which in this environment is harvested yearly, leaving the possibility for this pool to "reset" its isotopic composition as time passes and as water itself evolves. If this legacy effect were fully corrected for (i.e., if the whole of soil secondary solids were in equilibrium with today's heavier waters), the isotopic signature of soil solutions would likely be even heavier, further reinforcing our conclusion that harvesting is a major driver of light Si isotope export.

In light of these points, while we recognize that the agricultural weathering system of Kervidy-Naizin may not be in perfect steady state, we consider a quasi-steady-state approach both justifiable and robust for our mass-balance interpretation. In the revised manuscript, we included a brief discussion in Section 5.2 to explicitly acknowledge these limitations and clarify the assumptions underlying our approach.

**Changes:** L402: "This mass balance approach requires a steady state assumption. We recognize that in a human-impacted catchment, internal Si reservoirs—such as amorphous silica, phytoliths, and secondary clays—may experience transient shifts in response to land-use changes. While these changes could impact short-term fluxes, the isotopic coherence observed over multiple hydrological years and seasons (2015, 2017, 2021, 2022, 2023) in the water samples indicates that the system is

buffered against such variations in dissolved flux signatures. Additionally, the integration of time-averaged measurements from stream water, groundwater, and soil solutions captures long-term system dynamics, supporting the validity of this assumption (Bouchez et al., 2013). Over longer time scales, the continuous export of Si through biomass harvesting likely depletes solid-phase pools in the lighter isotopes, leading to progressively heavier  $\delta^{30}$ Si values in soil solutions. However, the solid phases currently present (e.g., clays, soil organic matter) formed at least partly under predisturbance conditions, and thus retain isotopically lighter signatures that are out of equilibrium with today's heavier waters. This legacy effect contributes to the observed isotopic contrast between solids and dissolved Si. If this effect were fully corrected for, the dissolved Si pool would appear even heavier, further supporting our interpretation that harvesting drives light Si isotope export. We therefore consider the quasi-steady-state framework a reasonable and robust approximation for interpreting catchment-scale Si fluxes."

Regarding the vertical gradients in [Si] and d30Si, there doesn't seem to be much discussion of a simple mixing between Si-deplete rainwater and Si-rich 'weathering' water. Could this be part of the interpretation?

**Response:** While mixing between Si-depleted rainwater and Si-rich weathering-derived water could explain some of the vertical gradients in [Si], it cannot fully account for the observed  $\delta^{30}$ Si signatures. To our knowledge, no silicon isotope values for rainwater have been reported in this region. If mixing were the dominant process, it would require rainwater to have a significantly heavier  $\delta^{30}$ Si signature to explain the enrichment in soil solutions. Instead, the observed  $\delta^{30}$ Si enrichment is best explained by biological processes, particularly plant uptake and subsequent harvesting, which preferentially remove lighter Si isotopes from the system. In the revised manuscript, we added a sentence clarifying that rainwater cannot be a significant Si input to the soil system and therefore cannot account for the observed concentration and isotope profiles.

**Changes:** L377: "While simple mixing between Si-depleted rainwater and Si-rich weathering-derived water could contribute to the observed [Si] gradients, it is unlikely to explain the  $\delta^{30}$ Si enrichment in soil solutions. This would require rainwater to have a significantly heavier  $\delta^{30}$ Si signature, which is improbable, and no such data exist for this region."

The authors assume that the bedrock is dissolving congruently (e.g. L311, but somewhat contradicted on L541), and that all primary minerals have the same Si isotope signature (e.g. Appendix B, L682). But how justifiable are these assumptions? A growing body of work demonstrates that minerals have specific d30Si signatures. Probably of minor importance here, but perhaps worth considering.

**Response:** Thank you for highlighting this important point. As noted earlier, we acknowledge an inconsistency in our original approach. In Section 5.1 (L311), we initially assumed congruent bedrock dissolution—i.e., that the water released reflects the bulk rock Si/Al ratio (7.54),  $\delta^{30}$ Si (-0.13 ± 0.05‰), and Ge/Si ratio (1.33 µmol mol-1). However, in our calculation of  $h_{regolith}^{Si}$  we treated quartz as inert, implying incongruent dissolution.

To address this, we revised the manuscript to consistently reflect incongruent bedrock dissolution, assuming that only muscovite, albite, and chamosite actively dissolve. We also incorporated mineral-specific  $\delta^{30}$ Si and Ge/Si signatures, based on a literature compilation, as summarized below:

| Mineral               | δ 30 Si (‰) | Comments                                   | Ge/Si (µmol mol-1) | comments                           |
|-----------------------|------------------------|--------------------------------------------|--------------------|------------------------------------|
| Quartz                | -0.06 ± 0.10 n=11      | Mean of data heavier than - 0.13 (bedrock) | 0.72 ± 0.30 n=5    | Mean of all quartz values          |
| Microcline and albite | -0.29 ± 0.14 n=17      | Mean for feldspars and plagioclase         | 2.55 ± 0.77 n=8    | Mean for feldspars and plagioclase |

Muscovite

Mean of available values

Chamosite -0.68 ± 0.18 n=6 Estimated using biotite values  $4.55 \pm 1.24 \text{ n} = 6$ Estimated using biotite values

For  $\delta^{30}$ Si, we selected only quartz data with values heavier than the bedrock average ( $-0.13 \pm 0.05\%$ ), as including lighter quartz values would not reproduce the bulk bedrock signature, given that other minerals exhibit even lower  $\delta^{30}$ Si. This "filtered" literature compilation yielded a guartz  $\delta^{30}$ Si of -0.06± 0.10‰. For microcline and albite, we adopted the average value reported for feldspars and plagioclases (-0.29 ± 0.14%). For chamosite, a member of the chlorite group, we used values from biotite (-0.68 ± 0.18%), and for muscovite, where direct data are lacking, we applied the average between biotite and feldspar (–0.49  $\pm$  0.11%), based on known trends relating  $\delta^{30}$ Si fractionation to polymerization (Douthitt, 1982; Savage et al., 2014) and interlayer cation effects (Méheut et al., 2009; Méheut and Schauble, 2014).

For Ge/Si ratios, we used  $0.72 \pm 0.30 \,\mu\text{mol} \,\text{mol}^{-1}$  for quartz,  $2.55 \pm 0.77 \,\mu\text{mol} \,\text{mol}^{-1}$  for albite and microcline,  $2.13 \pm 0.21 \,\mu$ mol mol-1 for muscovite, and  $4.55 \pm 1.24 \,\mu$ mol mol-1 for chamosite.

We acknowledged in the revised text that these values are derived from literature sources at other sites and are associated with uncertainty. However, this is currently the best approach available, as we only have one bulk bedrock sample and mineral-specific data are scarce. These kinds of assumptions are common in geochemical modeling, and we have taken care to test that our main outcomes are not overly sensitive to these choices.

Using these revised values, we recalculated the composition of the dissolving fluid, obtaining a Si/Al = 1.8,  $\delta^{30}$ Si = -0.44 ± 0.08% and Ge/Si = 2.8  $\mu$ mol mol-1. These corrections did not affect the main trends or interpretations of our  $\delta^{30}$ Si and Ge/Si dynamics in the catchment, but yet improved the internal consistency of the modeling framework and have been clearly documented in the revised manuscript and Supplementary Material.

Changes: In Supplementary material, we have added table S1 that includes the literature compilation on  $\delta^{30}$ Si and Ge/Si ratios. In the Appendix B (L743) we have added a table with the selected values and the explanation of how they were selected.

L779: "Using the values form Table B2 we have calculated the bedrock  $\delta^{30}$ Si and Ge/Si ratios corrected for quartz, which is assumed to be inert. To account for the water signature resulting from incongruent dissolution, and assuming a silicon mass balance with contributions solely from muscovite, albite, and chamosite (based on the values in Table B2), a Montecarlo simulation yielded  $\delta^{30}$ Sirock-qtz = -0.44 ± 0.08 % and Ge/Sirock-qtz = 2.80 ± 0.43 µmol mol-1."

L310: "Assuming incongruent dissolution of the bedrock —where silicon is preferentially released from more reactive minerals such as plagioclase, muscovite, and chlorite, while quartz is considered inert—and after correcting for quartz content, the initial water composition resulting from primary mineral dissolution is expected to reflect the non-quartz fraction of the rock. This is characterized by a Si/Al ratio (1.80),  $\delta^{30}$ Si (-0.44  $\pm$  0.11%), and a Ge/Si ratio (2.80  $\pm$  0.43  $\mu$ mol mol-1), detailed quartz correction in Appendix B". Finally, Figure 3 was updated to the new values of the initial fluid composition resulting from primary mineral dissolution."

There are three different approaches applied here: 1) a d30Si+Ge/Si mass balance, 2) a mass balance based on river Si fluxes, and 3) a mass balance based on soil geochemistry. Although they are designed to predict slightly different aspects of Si export, I was surprised not to see a more explicit comparison (e.g. in a table or a figure).

Response: We appreciate the reviewer's suggestion to more explicitly compare the three different approaches used to estimate Si export. We agree that a clearer side-by-side comparison would strengthen the manuscript. To address this, we moved Tables 2 and 5 to the appendix and introduced

a new summary table in the main text (now Table 2) that directly compares the estimates from all three approaches.

Changes: line 444: See new table 2 (L446) and Appendix Table D.1 (L806) and Table F.1 (L847).

The fractional export value for e\_Si in approach 2 (stream water + sediment based) is 0.36 (L498). As far as I understand, this includes E\_org, E\_sec and E\_prim - but is this inconsistent with a bedrock dominated by quartz? (which they assume elsewhere to be inert, e.g. 541 – if quartz is not dissolving then a minimum value for e\_Si would be the quartz fraction of the bedrock)

**Response:** We thank the reviewer for this thoughtful observation. It is correct that if quartz is assumed to be inert, the minimum theoretical value for  $e^{Si}_{river}$  should reflect the quartz fraction of the bedrock. Given that quartz comprises approximately 62% of the bedrock, this suggests a lower bound for  $e^{Si}_{river}$  around 0.62.

The lower value we obtained (0.36) using the stream water + sediment approach likely reflects methodological limitations. Our "gauging" estimate is based on turbidity-derived suspended sediment concentrations, which primarily capture the fine sediment fraction. Coarse and dense minerals such as quartz are more likely to be transported as bedload and are thus underrepresented in turbidity-based estimates. This grain-size bias in suspended sediment sampling is well documented (Bouchez et al., 2011; Lupker et al., 2012). In addition, it is also possible that some portion of the quartz is retained within the soil profile. Quartz is resistant to weathering and may accumulate over time in the regolith, particularly if it is not being mobilized either in dissolved form or as suspended or bedload particles. This would further reduce the fraction of Si exported via the river system, relative to the bulk rock composition. We acknowledge that this accumulation would violate the steady-state assumption regarding this particular component of soils, but in a way that does not affect the isotope mass balance equations used in the study (Bouchez et al., 2013).

As a result, this part of our methodology likely underestimates the total solid-phase Si export from primary minerals. However, even assuming a more conservative value of  $e_{river}^{Si}$  = 0.62—matching the quartz fraction of the bedrock—the amount of Si exported via harvesting ( $h_{river}^{Si}$ ) remains larger than the dissolved Si flux, supporting our conclusion that agricultural harvesting is a major component of Si export from the kervidy-Naizin catchment. We now include a discussion of this issue and the associated references in the revised manuscript.

**Changes:** L538: "It is important to note that we assume quartz to be inert; therefore, the minimum theoretical value for  $e^{Si}_{river}$  should reflect the quartz fraction of the bedrock, which is 0.62 (Table 1), suggesting a lower bound for  $e^{Si}_{river}$  of approximately 0.62. The lower value we obtained ( $e^{Si}_{river} = 0.36 \pm 0.21$ ) likely reflects methodological limitations associated with the estimation of suspended sediment fluxes. Specifically, the approach based on turbidity-derived suspended sediment concentrations predominantly captures the fine sediment fraction. Coarse and dense minerals such as quartz are preferentially transported as bedload and are underrepresented in turbidity-based estimates, a known grain-size bias in suspended sediment sampling (Bouchez et al., 2011; Lupker et al., 2012). Additionally, a portion of the quartz may be retained within the soil profile, as quartz is highly resistant to weathering and can accumulate over time in the regolith if not mobilized either in dissolved form or as suspended or bedload particles. These factors likely contribute to an underestimation of the total solid-phase Si export from primary minerals."

L557: "To further assess the sensitivity of this result to assumptions regarding  $e^{Si}_{river}$  we conducted an additional Monte Carlo simulation in which  $e^{Si}_{river}$  was set to 0.62 ± 0.21, corresponding to the quartz fraction of the bedrock under the assumption that quartz is inert and retained in the solid phase. In this conservative scenario, the resulting mean  $h^{Si}_{river}$  is 0.30 ± 0.18 (Fig. 6b). Importantly, even under this assumption of higher particulate Si export, harvesting remains a major Si flux in the catchment,

exceeding the dissolved export fraction ( $w_{river}^{Si}$ = 0.12 ± 0.06). These results reinforce the conclusion that agricultural harvesting plays a dominant role in the overall Si export from this system."

L563: Figure 6 was updated to include the second simulation.

**Minor comments:**

L56: Either more recent revisions of the Si budget (e.g. Treguer et al) and/or the 'original' river Si flux estimates (e.g. Dürr et al/Beusen et al) might be appropriate here.

**Response:** Thankyou for the suggestion. We added Dürr et al. (2011) and Beusen et al. (2009) for the values of dissolved Si (6.2 ± 1.8 Tmol Si yr-1), Frings et al. (2016) for the values of dissolvable amorphous silica (1.9 ± 1.0 Tmol Si yr-1), and Tréguer et al. (2021) for the most recent review. Additionally, we removed the reference to groundwater to better align with the cited studies.

**Changes:** L53: "Global agricultural activities are estimated to remove approximately 7.8 Tmol of Si per year from landscapes (Matichenkov and Bocharnikova, 2001), a quantity almost equivalent to the 8.1 Tmol of dissolved Si and dissolvable amorphous silica transferred from continents to oceans via rivers and groundwater (Beusen et al., 2009; Dürr et al., 2011; Frings et al., 2016; Tréguer et al., 2021)."

L84: To avoid overstating the novelty of this contribution, maybe already mention here that some previous work has identified that plant biomass as a whole doesn't seem to discriminate against Ge as much as the phytolith-based estimates cited here would suggest.

Response: We modified the text to acknowledge this broader perspective

**Changes:** L81: "While some studies suggest that Ge is discriminated against Siduring vascular plant uptake (Blecker et al., 2007; Derry et al., 2005; Lugolobi et al., 2010; Meek et al., 2016), other research indicates that plant biomass as a whole does not exhibit as strong a discrimination (Delvigne et al., 2009; Frings et al., 2021b; Kaiser et al., 2020; Rains et al., 2006; Sparks et al., 2011)".

L158: if the bedrock comprises bedding of different lithologies, is this one sample enough to capture the heterogeneity? Even in plutonic rocks variability in 'immobile' element content can be large (which becomes important for e.g. the mass-balances and the 'tau' values later).

**Response:** We acknowledge that relying on a single bedrock sample is not ideal for capturing potential lithological heterogeneity and has potential implications for the calculation of soil weathering indices. To address this limitation, we incorporated data from Denis and Dabard (1988), who compiled chemical analyses (n = 9) from the same geological unit that underlies our study basin. These samples, collected along a transect approximately 32 km southeast of our site, report  $TiO_2$  concentrations with a mean of  $0.71 \pm 0.19$  wt.% (SD), which closely matches the  $TiO_2$  concentration measured in our own bedrock sample ( $0.74 \pm 0.04$  wt.%).

To better account for both natural variability and analytical uncertainty in the bedrock Ti concentration, we implemented a Monte Carlo simulation when calculating  $\tau_{prim}$ ,  $e_{sec}^{Si}$  and  $e_{org}^{Si}$ . Specifically, we modeled the Ti concentration in the bedrock as a normal distribution with a mean of 0.74 wt.% and a standard deviation of 0.19 wt.%, reflecting the variability reported by Denis and Dabard (1988). For soil samples, we assumed a 5% relative uncertainty in Ti concentration. This approach integrates both measurement uncertainty and spatial heterogeneity in schist bedrock, thereby improving the robustness of our mass-balance calculations.

Importantly, despite this more rigorous treatment of uncertainty, the resulting values are nearly identical to those previously reported. This is because  $\tau_{prim}$ ,  $e_{sec}^{Si}$  and  $e_{org}^{Si}$  in Eq. 10 were already calculated using the mean values across all soil profile samples, which inherently incorporate substantial natural variability, some of it being most likely the result of variability in bedrock

composition. The original values were  $\tau_{prim}$  = -0.42 ± 0.15,  $e_{sec}^{Si}$  = 0.088 ± 0.042, and  $e_{org}^{Si}$  = 0.0064 ± 0.0033, while the updated Monte Carlo-derived estimates are  $\tau_{prim}$  = -0.42 ± 0.15,  $e_{sec}^{Si}$  = 0.088 ± 0.043, and  $e_{org}^{Si}$  = 0.0065 ± 0.0035. These consistent results lend support to the robustness of our initial estimates and demonstrate that the updated method supports the same conclusions, while providing a more comprehensive representation of uncertainty.

Changes: L611: "Relying on a single bedrock sample may not fully capture potential lithological heterogeneity in the parent rock, especially regarding the Ti concentrations. To address this, we performed a sensitivity test using the dataset of Denis and Dabard (1988), which includes nine bedrock samples from the same geological unit as the one underlying the Kervidy-Naizin catchment. The results of this sensitivity test (Appendix G) demonstrate that the estimated  $\tau_{prim}$  values are robust to this source of variability, with the same mean value obtained when incorporating this broader dataset."

L849: We added an Appendix with the results of this sensibility analysis: "Appendix G: Sensitivity analysis on bedrock lithological variability for the calculation of  $\tau_{nrim}^{Si}$ "

L180: What is precision/long term reproducibility on the elemental data? Were any secondary reference materials included in the analyses?

**Response:** Yes, the river water standard SLRS-5 (National Research Council, Canada) was systematically analyzed, with a long-term analytical precision of better than 5%. This was added in the method section.

**Changes:** L181: "The analytical precision for elemental analysis is < 5% based on the long-term measurement of the SLRS-5 reference material (National Research Council, Canada)".

Fig 2: presumably cmbs on the y-axis, not mbs. Greek letter mu (not u) on Ge/Si x-axis.

**Response:** Thank you for pointing out this error.

**Changes:** In the new manuscript we have corrected "mbs" to "cm b.s." on the y-axis and replaced "umol mol-1" with "µmol mol-1" on the Ge/Si x-axis (see Figure 2, L267).

L285: "compared"

**Response:** Thank you for pointing out this error. We have corrected this mistake.

Changes: L286: we have added "compared"

345: 'show a positive correlation' / 'are positively correlated'

Response: Thank you for pointing out this error. We have corrected this mistake.

Changes: L361: we have changed "show a positive correlation" to "are positively correlated"

L415 – also Baronas et al 2020 GBC would be appropriate to cite here?

**Response:** Thank you for the suggestion. We have added Baronas et al. (2020) to acknowledge their findings that a significant portion of Si is taken up by vegetation fbioSi =  $39 \pm 14\%$ , which aligns with the range reported by Frings et al. (2021).

**Changes:** L453: we have added Baronas et al. 2020 and corrected the sentence: "These contributions are also higher than those observed in four other non-agricultural catchments, where  $e_{org-iso}^{Si}$  ranged from 0.20 ± 0.1 to 0.42 ± 0.23 (Baronas et al., 2020; Frings et al., 2021).

L483 – actually relatively high?

**Response:** We are not sure what the reviewer is referring to here — is it that the fraction of Si exported in the dissolved form  $(0.12 \pm 0.06)$  is high compared to the value determined by the isotopic mass balance? If so, we underline that the isotopic mass balance approach yields values ranging from 0.15  $\pm$  0.08 to 0.22  $\pm$  0.11, which are within the uncertainties of the value obtained by the gauging method. Assuming this is what the reviewer is suggesting.

**Changes:** L521: "The mean fraction of Si exported from the catchment in the dissolved form is 0.12  $\pm$  0.06, which is consistent with the values calculated using the isotopic mass balance when considering their respective uncertainties (0.15  $\pm$  0.08 to 0.22  $\pm$  0.11)."

L519: eSi\_sec repeated here - presumably should be eSi\_org?

**Response:** Thank you for catching this error. We have corrected  $e_{sec}^{Si}$  to  $e_{org}^{Si}$  in the revised manuscript.

**Changes:** L574: we have changed  $e_{sec}^{Si}$  to  $e_{org}^{Si}$

L520: If this is a schist bedrock, how variable is the Ti content, and how are uncertainties propagated?

**Response:** We thank the reviewer for this related comment. As discussed in our response to Comment L158, we addressed the potential variability in Ti content within schist bedrock by incorporating external geochemical data from Denis and Dabard (1988), who reported  $TiO_2$  concentrations (n = 9) with a mean of  $0.71 \pm 0.19$  wt.% in the same geological unit underlying our study site. This variability was integrated into our calculations through a Monte Carlo simulation that models Ti concentration in the bedrock as a normal distribution (mean = 0.74 wt.%, SD = 0.19 wt.%) and includes a 5% relative uncertainty for the soil Ti concentrations. This framework allowed us to propagate uncertainties in Ti through all downstream calculations of  $\tau_{prim}$ ,  $e_{sec}^{Si}$  and  $e_{org}^{Si}$ , ultimately providing a more comprehensive assessment of uncertainty. As noted previously, this improved treatment does not significantly change the final estimates, reinforcing the robustness of our original results.

**Changes:** L611: Relying on a single bedrock sample may not fully capture potential lithological heterogeneity in the parent rock, especially regarding the Ti concentrations. To address this, we performed a sensitivity test using the dataset of Denis and Dabard (1988), which includes nine bedrock samples from the same geological unit as the one underlying the Kervidy-Naizin catchment. The results of this sensitivity test (Appendix G) demonstrate that the estimated  $\tau_{prim}$  values are robust to this source of variability, with the same mean value obtained when incorporating this broader dataset."

L849: We have added an Appendix with the results of this sensibility analysis: "Appendix G: Sensitivity analysis on bedrock lithological variability for the calculation of  $au_{prim}^{Si}$ "

L530: What is the justification for using stream water rather than soil solutions to define e\_prec?

**Response:** We chose to use stream water rather than soil solutions because the former integrates contributions from the entire catchment, providing a more representative estimate of  $\varepsilon_{prec}^{Si}$  at the catchment scale rather than reflecting local variations.

**Changes:** L582: "Stream was used instead of soil solutions, because the former integrates contributions from the entire catchment, providing a more representative estimate of  $\varepsilon_{prec}^{Si}$  at the catchment scale, rather than reflecting local variations."

L544: "to be inert"

Response: Thank you for correcting this error.

Changes: L602: we have added "be".

L567: Why are these values so low compared to previous two estimates?

**Response:** First, we would like to clarify that the values  $e^{Si}_{org}$  and  $e^{Si}_{org-iso}$  do not represent the same quantity. The term  $e^{Si}_{org-iso}$  refers to the fraction of silicon that is both eroded naturally and exported through harvesting. In contrast, term  $e^{Si}_{org}$  specifically represents the silicon associated with soil organic matter—i.e., the phytoliths that remain in the soil after harvesting, and that are eroded naturally. This distinction, which stems from the way the different mass balance equations are set up, explains why the value of  $e^{Si}_{org}$  is lower than that of  $e^{Si}_{org-iso}$ .

Regarding  $e^{Si}_{clay-iso}$  and  $e^{Si}_{sec}$ , these metrics are intended to represent the same process but determined with different approaches. In particular, estimating  $e^{Si}_{sec}$  based on elemental metrics (eq. 15) is particularly challenging due to the complexity and spatial heterogeneity of soil processes. For instance, the preferential erosion of fine-grained, clay-rich material can remove a significant portion of the secondary Si pool from the soil. As  $e^{Si}_{sec}$  is calculated using [Si]sec, reflecting the amount of soil Si contained in secondary minerals, any Si loss by clay erosion leads to an underestimation of [Si]sec. Consequently, the resulting  $e^{Si}_{sec}$  value is biased toward lower estimated. We now include a discussion about these potential issues, acknowledging in particular that an ideal approach would require a more detailed characterization of the soil profiles and mineralogy.

**Changes:** L632: "It is worth noting that, although these values ( $e^{Si}_{sec}$  and  $e^{Si}_{org}$ ) are lower than those obtained from the isotopic mass balance (e.g.,  $e^{Si}_{sec-iso}$  and  $e^{Si}_{org-iso}$ ), they reflect different processes. Specifically,  $e^{Si}_{org}$  quantifies the fraction of Si retained in soil organic matter after harvest (i.e., residual phytoliths), rather than total plant export. Similarly,  $e^{Si}_{sec}$  represents the estimated soil Si pool contained in secondary minerals, which can be biased low due to the unaccounted loss of fine, clayrich material during erosion. This conceptual distinction, as well as differences in methodological approaches, explains the lower magnitude of these values relative to the isotopically derived fluxes."

L595: fractionation factors are not 'heavy' or 'light'; better to talk about magnitude. In general, fractionation factor normally refers to so-called 'alpha' notation, and just 'fractionation' alone to 'epsilon' notation – see Coplen 2011 DOI: 10.1002/rcm.5129.

**Response:** Thank you for providing clarification on the use of fractionation factor for alpha notation and isotopic fractionation for the epsilon notation. We have corrected to talk about the magnitude of the isotopic fractionation. Additionally, we have revised the sections 5.5.1 and 5.5.4 replacing the "fractionation factor" by "isotopic fractionation".

Changes: see lines with highlighted changes in all the manuscript.

L634: See also Vandervenne et al 2013 Proc Royal Soc B.

**Response:** Thank you for the suggestion. We have now included Vandevenne et al. (2013) as a reference to highlight the role of grazing animals in accelerating the return of biogenic Si to the soil and enhancing its reactivity.

**Changes:** L701: "Grazing animals can influence Si cycling by accelerating the return of biogenic Si to the soil through feces, thereby enhancing its reactivity and dissolvability (Vandevenne et al., 2013)".

L708: Does the very low number of acceptable iterations (e.g. 0.2% for scenario 2) simply imply that an assumption underpinning the mass-balance or endmember assignments is incorrect?

**Response:** Thank you for your thoughtful comment. You are correct that the very low number of acceptable iterations in Scenario 2 (e.g., 0.2%) originally suggested a potential issue with our mass-balance assumptions or end member assignments. In our initial approach, we assumed congruent dissolution of the bedrock. As explained above we have now revised the model to consistently reflect incongruent dissolution, assuming that only muscovite, albite, and chamosite actively contribute to weathering. This update includes the use of mineral-specific  $\delta^{30}$ Si and Ge/Si signatures based on a literature compilation.

As a result, the  $\delta^{30}$ Si value assigned to the dissolving rock has changed significantly, leading to a much higher number of valid iterations across all scenarios. Previously, out of 6 million iterations, only 1.31% were valid for Scenario 1, 0.23% for Scenario 2, and 25.69% for Scenario 3. In the revised model, 7.27% were valid for Scenario 1, 1.92% for Scenario 2, and 51.59% for Scenario 3. These improvements result in a more robust and internally consistent set of model outputs, indicating that mass balance is "more likely" to be achieved using these updated values for the solution produced by rock dissolution.

**Changes:** L805: "Out of the 6 million iterations, 7.27% were valid for Scenario 1, 1.92% for Scenario 2, and 51.59% for Scenario 3."

**References:**

Baronas, J. J., West, A. J., Burton, K. W., Hammond, D. E., Opfergelt, S., Pogge Von Strandmann, P. A. E., James, R. H., and Rouxel, O. J.: Ge and Si Isotope Behavior During Intense Tropical Weathering and Ecosystem Cycling, Global Biogeochemical Cycles, 34, e2019GB006522, https://doi.org/10.1029/2019GB006522, 2020.

Beusen, A. H. W., Bouwman, A. F., Dürr, H. H., Dekkers, A. L. M., and Hartmann, J.: Global patterns of dissolved silica export to the coastal zone: Results from a spatially explicit global model, Global Biogeochemical Cycles, 23, 2008GB003281, https://doi.org/10.1029/2008GB003281, 2009.

Blecker, S. W., King, S. L., Derry, L. A., Chadwick, O. A., Ippolito, J. A., and Kelly, E. F.: The ratio of germanium to silicon in plant phytoliths: quantification of biological discrimination under controlled experimental conditions, Biogeochemistry, 86, 189–199, https://doi.org/10.1007/s10533-007-9154-7, 2007.

Bouchez, J., Gaillardet, J., France-Lanord, C., Maurice, L., and Dutra-Maia, P.: Grain size control of river suspended sediment geochemistry: Clues from Amazon River depth profiles, Geochem Geophys Geosyst, 12, 2010GC003380, https://doi.org/10.1029/2010GC003380, 2011.

Bouchez, J., Von Blanckenburg, F., and Schuessler, J. A.: Modeling novel stable isotope ratios in the weathering zone, American Journal of Science, 313, 267–308, https://doi.org/10.2475/04.2013.01, 2013.

Delvigne, C., Opfergelt, S., Cardinal, D., Delvaux, B., and André, L.: Distinct silicon and germanium pathways in the soil-plant system: Evidence from banana and horsetail: DISTINCT SI AND GE PATHWAYS IN PLANTS, J. Geophys. Res., 114, n/a-n/a, https://doi.org/10.1029/2008JG000899, 2009.

Denis, E. and Dabard, M. P.: Sandstone petrography and geochemistry of late proterozoic sediments of the armorican massif (France) — A key to basin development during the cadomian orogeny, Precambrian Research, 42, 189–206, https://doi.org/10.1016/0301-9268(88)90017-4, 1988.

Derry, L. A., Kurtz, A. C., Ziegler, K., and Chadwick, O. A.: Biological control of terrestrial silica cycling and export fluxes to watersheds, Nature, 433, 728–731, https://doi.org/10.1038/nature03299, 2005.

Douthitt, C. B.: The geochemistry of the stable isotopes of silicon, Geochimica et Cosmochimica Acta, 46, 1449–1458, https://doi.org/10.1016/0016-7037(82)90278-2, 1982.

- Dürr, H. H., Meybeck, M., Hartmann, J., Laruelle, G. G., and Roubeix, V.: Global spatial distribution of natural riverine silica inputs to the coastal zone, Biogeosciences, 8, 597–620, https://doi.org/10.5194/bg-8-597-2011, 2011.
- Frings, P. J., Clymans, W., Fontorbe, G., De La Rocha, C. L., and Conley, D. J.: The continental Si cycle and its impact on the ocean Si isotope budget, Chemical Geology, 425, 12–36, https://doi.org/10.1016/j.chemgeo.2016.01.020, 2016.
- Frick, D. A., Remus, R., Sommer, M., Augustin, J., Kaczorek, D., and Von Blanckenburg, F.: Silicon uptake and isotope fractionation dynamics by crop species, Biogeosciences, 17, 6475–6490, https://doi.org/10.5194/bg-17-6475-2020, 2020.
- Frings, P. J., Schubring, F., Oelze, M., and Von Blanckenburg, F.: Quantifying biotic and abiotic Si fluxes in the Critical Zone with Ge/Si ratios along a gradient of erosion rates, Am J Sci, 321, 1204–1245, https://doi.org/10.2475/08.2021.03, 2021.
- Kaiser, S., Wagner, S., Moschner, C., Funke, C., and Wiche, O.: Accumulation of germanium (Ge) in plant tissues of grasses is not solely driven by its incorporation in phytoliths, Biogeochemistry, 148, 49–68, https://doi.org/10.1007/s10533-020-00646-x, 2020.
- Lugolobi, F., Kurtz, A. C., and Derry, L. A.: Germanium—silicon fractionation in a tropical, granitic weathering environment, Geochimica et Cosmochimica Acta, 74, 1294–1308, https://doi.org/10.1016/j.gca.2009.11.027, 2010.
- Lupker, M., France-Lanord, C., Galy, V., Lavé, J., Gaillardet, J., Gajurel, A. P., Guilmette, C., Rahman, M., Singh, S. K., and Sinha, R.: Predominant floodplain over mountain weathering of Himalayan sediments (Ganga basin), Geochimica et Cosmochimica Acta, 84, 410–432, https://doi.org/10.1016/j.gca.2012.02.001, 2012.
- Meek, K., Derry, L., Sparks, J., and Cathles, L.: 87Sr/86Sr, Ca/Sr, and Ge/Si ratios as tracers of solute sources and biogeochemical cycling at a temperate forested shale catchment, central Pennsylvania, USA, Chemical Geology, 445, 84–102, https://doi.org/10.1016/j.chemgeo.2016.04.026, 2016.
- Méheut, M. and Schauble, E. A.: Silicon isotope fractionation in silicate minerals: Insights from first-principles models of phyllosilicates, albite and pyrope, Geochimica et Cosmochimica Acta, 134, 137–154, https://doi.org/10.1016/j.gca.2014.02.014, 2014.
- Méheut, M., Lazzeri, M., Balan, E., and Mauri, F.: Structural control over equilibrium silicon and oxygen isotopic fractionation: A first-principles density-functional theory study, Chemical Geology, 258, 28–37, https://doi.org/10.1016/j.chemgeo.2008.06.051, 2009.
- Rains, D. W., Epstein, E., Zasoski, R. J., and Aslam, M.: Active Silicon Uptake by Wheat, Plant Soil, 280, 223–228, https://doi.org/10.1007/s11104-005-3082-x, 2006.
- Savage, P. S., Armytage, R. M. G., Georg, R. B., and Halliday, A. N.: High temperature silicon isotope geochemistry, Lithos, 190–191, 500–519, https://doi.org/10.1016/j.lithos.2014.01.003, 2014.
- Sparks, J. P., Chandra, S., Derry, L. A., Parthasarathy, M. V., Daugherty, C. S., and Griffin, R.: Subcellular localization of silicon and germanium in grass root and leaf tissues by SIMS: evidence for differential and active transport, Biogeochemistry, 104, 237–249, https://doi.org/10.1007/s10533-010-9498-2, 2011.
- Tréguer, P. J., Sutton, J. N., Brzezinski, M., Charette, M. A., Devries, T., Dutkiewicz, S., Ehlert, C., Hawkings, J., Leynaert, A., Liu, S. M., Llopis Monferrer, N., López-Acosta, M., Maldonado, M., Rahman, S., Ran, L., and Rouxel, O.: Reviews and syntheses: The biogeochemical cycle of silicon in the modern ocean, Biogeosciences, 18, 1269–1289, https://doi.org/10.5194/bg-18-1269-2021, 2021.

Vandevenne, F. I., Barão, A. L., Schoelynck, J., Smis, A., Ryken, N., Van Damme, S., Meire, P., and Struyf, E.: Grazers: biocatalysts of terrestrial silica cycling, Proc. R. Soc. B., 280, 20132083, https://doi.org/10.1098/rspb.2013.2083, 2013.

**Reviewer 2:**

**Peer Review Report**

Manuscript Number: EGUSPHERE-2025-78

**Title: Quantifying the agricultural footprint on the silicon cycle: Insights from silicon isotopes and Ge/Si ratios**

Review of manuscript EGUSPHERE-2025-78 submitted to BG by Sofía López-Urzúa and colleagues:

With apologies to the authors and editor for the delayed review. The manuscript couple silicon isotopes and Ge/Si data of different critical zone compartments to quantify the Si export from the catchment. The authors identify vertical gradient in water pools, with a heavier Si isotopic composition in soil porewater and lighter composition in groundwater interpreted as a result of plant uptake in shallow soil profiles. Using two independent quantitative approach the authors identify plant uptake to be the largest Si export flux from the catchment. The manuscript is generally well written, methodologically sound, and adds valuable insights into terrestrial Si cycling. The results highlighted in the study aligns well with the scope of BG and I recommend the manuscript for publication after considering the following comments.

I have one suggestion regarding the entire section 5.1. One of the key highlights of the manuscript is that the authors have made considerable effort in measuring  $\delta^{30}$ Si and Ge/Si from different critical zone compartments, with an objective to decipher what controls the Si cycle in the catchment. However, the results are not well depicted in the figures and discussed. I understand the focus is more on the quantification of the Si export, but I would suggest a bit more detail to be included especially in 5.3 about the plant uptake and Si isotopic fractionation pathways linking to vertical gradient (e.g. Appendix C, Fig. C1 nested piezometers and soil solutions).

**Response:** We thank the reviewer for this thoughtful suggestion. It is absolutely right to highlight that one of the strengths of the manuscript lies in the effort to measure  $\delta^{30}$ Si and Ge/Si across various compartments of the Critical Zone. While our primary objective has been to quantify the agricultural footprint on the Si cycle, we recognize that better integrating the isotopic patterns across compartments—particularly in relation to vertical gradients and plant uptake—significantly enhance the interpretive depth of the manuscript. To maintain the overall conciseness of the main text, we revised Section 5.1 to include additional context and detail regarding the plant uptake and Si isotopic fractionation pathways linking to vertical gradient.

**Changes:** We have revised Section 5.1 accordingly. Specific additions were made in response to the more detailed minor comments outlined below, particularly focusing on improving the connection between isotopic signals and hydrological/biogeochemical processes along depth profiles.

**Minor comments**

l83-86: The authors have introduced the potential of Ge/Si ratio in decoupling plant uptake vs. weathering here without commenting on the results from Frings et al., (2021b), which they have discussed in l369-371. I would recommend to introduce the key highlights from Frings et al., (2021b) as well, since the validity of Ge discrimination against Si during plant uptake is under question.

**Response:** We have revised the text to reflect this broader perspective and to introduce the findings of Frings et al. (2021b) earlier in the manuscript.

Changes: L81: "While some studies suggest that Ge is discriminated against Si during vascular plant uptake (Blecker et al., 2007; Derry et al., 2005; Lugolobi et al., 2010; Meek et al., 2016), other research indicates that plant biomass as a whole does not exhibit as strong a discrimination (Delvigne et al., 2009; Frings et al., 2021; Kaiser et al., 2020; Rains et al., 2006; Sparks et al., 2011)".

l137: Any irrigation practices?

**Response:** No, irrigation is not practiced in the catchment. We have clarified this in the revised manuscript.

**Changes:** L140: "There is no irrigation in the catchment, and crop production relies entirely on natural rainfall."

l124: Please add details of the general climate of the catchment, especially rainfall.

Response: We have added climate information to the revised manuscript.

**Changes:** L108: "The climate is temperate and humid, with an average annual temperature of  $11.2 \pm 0.6$  °C and annual precipitation of  $853 \pm 210$  mm, falling exclusively as rain. Estimated Penman potential evapotranspiration is  $697 \pm 57$  mm yr -1, and runoff is approximately  $340 \pm 169$  mm yr -1 (Fovet et al., 2018)."

l182: Add the uncertainty and certified reference used for ICP-MS, especially for traces (Al, Fe).

**Response:** We have included this detail in the revised methods section.

**Changes:** L181: "The analytical precision for elemental analysis is < 5% based on the long-term measurement of the SLRS-5 reference material (National Research Council, Canada)".

l185-193: The sentences here are not clear here. I suggest you re-write to clarify phases and protocol. If I understand correctly, you target here amorphous aluminosilicates, crystalline Al and Fe oxides? The amorphous aluminosilicates can be clay precursors with different fractionation factors than adsorption onto oxyhydroxides.

**Response:** Thank you for pointing this out. We have reorganized and clarified the paragraph in the revised manuscript.

**Changes:** L188: "Amorphous Al-Si phases were assessed using extraction by oxalic acid and ammonium oxalate solution buffered to pH 3 (Tamm, 1922). Approximately 1.25 g of soil ground to 250 µm was agitated for 4 hours in the presence of 50 ml of reagent, at 20°C in the dark. Crystalline Fe and Al oxides were extracted using a sodium citrate-bicarbonate-dithionite (CBD) solution at elevated temperatures (Mehra & Jackson, 1960). This method targets Fe oxides and oxyhydroxides, with minimal dissolution (<5%) of Fe present in silicate minerals (Jeanroy, 1983). Approximately 0.5 g of soil ground to 250 µm was mixed with 25 ml of the extraction solution. After adding 1.5 ml of the reducing solution, the mixture was heated to 80°C in a water bath for 30 minutes with intermittent agitation. After cooling, the volume was adjusted to 50 ml, homogenized, and filtered."

l334: Again, here you mention dry and wet season and redox processes but we have no clue about the rainfall variability or seasonality of the study site. Interestingly, I could see that groundwater sampled do not exhibit any significant changes in  $\delta^{30}$ Si (maybe ±0.2) over the time period of sampling (8 years?)

**Response:** Thank you for pointing this out. We agree that providing more context on rainfall variability and the seasonality of the study site will help readers better understand the discussion on redox processes and seasonal dynamics. We included a brief description of the rainfall regime (see comment above) and hydrological periods in the revised manuscript and clearly link it to Appendix Figure A1, which illustrates water level fluctuations during the dry and wet seasons. Regarding the  $\delta^{30}$ Si variability, we appreciate your observation. Indeed, the deep groundwater samples show relatively stable  $\delta^{30}$ Si values (within ±0.2%) over the sampling period, while more variability is observed in shallow groundwater, likely reflecting the stronger impact of near-surface processes for these samples. We will make this distinction clearer in the revised text.

**Changes:** L108: "The climate is temperate and humid, with an average annual temperature of  $11.2 \pm 0.6$  °C and annual precipitation of  $853 \pm 210$  mm, falling exclusively as rain. Estimated Penman potential evapotranspiration is  $697 \pm 57$  mm yr -1, and runoff is approximately  $340 \pm 169$  mm yr -1 (Fovet et al., 2018)."

L345: "The catchment exhibits marked seasonal variability, characterized by distinct hydrological periods (Humbert et al., 2015; Molenat et al., 2008). Period B (high flow) is associated with heavy rainfall and rising water tables, which enhance hydrological connectivity across the catchment. In contrast, Period C (recession) marks the onset of drier conditions, with progressively falling water levels. These seasonal dynamics are shown in Appendix Figure A1.  $\delta^{30}$ Si values in deep groundwater remain relatively stable (±0.2‰ over 8 years), indicating limited seasonal influence, while shallow groundwater shows more variability, likely due to stronger coupling with surface processes."

l380: You mention in results the significant differences in  $\delta^{30}$ Si between the Gueriniec and Kerroland transect (l285-288) but I don't see any discussion related to that? What can be the drivers of such differences? I can see it is consistent in shallow as well as deep groundwater, with a higher Si/Al in Kerroland.

**Response:** Thank you for highlighting this important observation. We agree that more discussion is needed regarding the differences in  $\delta^{30}$ Si between the Guériniec and Kerroland transects. One possibility is that the steeper slope in the Guériniec transect results in shorter residence times, limiting clay precipitation and thus affecting isotopic signatures. Another factor could be differences in bedrock mineralogy, with Guériniec potentially containing a higher proportion of lighter minerals. Despite these differences, the overall vertical trend observed in both transects—particularly the signal attributed to plant uptake—remains consistent. This supports the validity of the main mechanisms controlling silicon dynamics discussed in the manuscript. While we do not currently have enough evidence to provide a definitive explanation for the difference between the two profiles, we included a brief discussion proposing potential explanations.

**Changes:** L335: "In addition to vertical gradients, systematic differences between transects are observed, with groundwater from the Kerroland transect consistently exhibiting heavier  $\delta^{30}$ Si values and higher Si/Alratios than Guériniec (Fig. 2, Table 1). Several factors may contribute to this contrast. The steeper slope in Guériniec likely results in shorter water residence times, limiting the extent of clay precipitation and associated isotopic fractionation. Additionally, subtle differences in bedrock composition between the transects—such as variations in muscovite and chamosite content—may influence the initial isotopic signature of the dissolved Si pool. While current data does not allow us to fully disentangle these factors, both Kerroland and Guériniec display consistent vertical trends in  $\delta^{30}$ Si and Si/Al ratios. This indicates that the same underlying processes are operating across the catchment, despite spatial variability in  $\delta^{30}$ Si values."

l519: Repetition here, please change instead of eSiorg?

**Response:** Thank you for catching this oversight. We have corrected the redundant notation in the revised manuscript by replacing  $e_{sec}^{Si}$  with  $e_{org}^{Si}$ .

**Changes:** L574: we have changed  $e_{sec}^{Si}$  to  $e_{ora}^{Si}$

l567-l570: Here, I have a query regarding the assumption of organic matter Si. In Table S2, I can see you mention about the OM and also the Si content associated, which is ~0.1%? Is that the Si bound to organic matter? If yes, could you justify the assumption of 2.3%?

**Response:** We appreciate this insightful question. In our calculation of silicon export associated with organic matter  $(e^{Si}_{org})$ , we assumed that soil organic matter primarily derives from plant material—specifically, in this agricultural context, from wheat residues. We used a Si concentration

of 2.3% by weight, based on values measured in wheat leaves (Hodson et al., 2005), as a proxy for the Si content associated with organic matter. This approach assumes limited alteration or loss of biogenic Si during the decomposition of plant residues and their transformation into soil organic matter. We acknowledge that this simplification may overestimate the true Si content bound to OM, but we opted for a consistent plant-based reference given the agricultural character of the catchment. We have clarified this assumption in the revised manuscript.

**Changes:** L629: "We assumed that the soil organic matter primarily originates from plant material—specifically, in this agricultural context, from wheat residues—and therefore assumed that 2.3% of this organic matter consists of Si, as reported for wheat (Hodson et al., 2005)."

**Figures**

Fig. 2: Correct the mistake in the depth unit mentioned in y axis, should be cmbs and please expand for reader ease.

**Response:** Thank you for pointing out this error.

**Changes:** L267: In the new manuscript we have corrected "mbs" to "cm b.s." on the y-axis and replaced "umol mol-1" with "µmol mol-1" on the Ge/Si x-axis, see Figure 2).

Fig. 3: Please include some endmembers from the critical zone here rather than presumed trends perhaps. Are you still certain about the plant uptake trend in Ge/Si vs  $\delta^{30}$ Si relationship? The plant leaf samples indicate Ge/Si close to or greater than the bedrock/water, pointing to a more no discrimination or selective uptake of Ge relative to Si.

**Response:** Thank you for this thoughtful suggestion. In the revised manuscript, we now include representative endmembers from the local Critical Zone, such as measured values for secondary clay and vegetation, to better contextualize the observed trends. We acknowledge that the plant data exhibit Ge/Si ratios similar to—or even exceeding—those of the bedrock and water, which challenges the assumption of strong discrimination against Ge during plant uptake However, the  $\delta^{30}$ Si and Si/Al patterns still suggest a plant uptake signal. To better reflect this ambiguity, we have removed the uptake/formation arrows from panels A and C in the revised figure and clarified the caption accordingly.

Changes: L323: See Figure 3.

**References:**

Blecker, S. W., King, S. L., Derry, L. A., Chadwick, O. A., Ippolito, J. A., and Kelly, E. F.: The ratio of germanium to silicon in plant phytoliths: quantification of biological discrimination under controlled experimental conditions, Biogeochemistry, 86, 189–199, https://doi.org/10.1007/s10533-007-9154-7, 2007.

Delvigne, C., Opfergelt, S., Cardinal, D., Delvaux, B., and André, L.: Distinct silicon and germanium pathways in the soil-plant system: Evidence from banana and horsetail: DISTINCT SI AND GE PATHWAYS IN PLANTS, J. Geophys. Res., 114, n/a-n/a, https://doi.org/10.1029/2008JG000899, 2009.

Derry, L. A., Kurtz, A. C., Ziegler, K., and Chadwick, O. A.: Biological control of terrestrial silica cycling and export fluxes to watersheds, Nature, 433, 728–731, https://doi.org/10.1038/nature03299, 2005.

Fovet, O., Ruiz, L., Gruau, G., Akkal, N., Aquilina, L., Busnot, S., Dupas, R., Durand, P., Faucheux, M., Fauvel, Y., Fléchard, C., Gilliet, N., Grimaldi, C., Hamon, Y., Jaffrezic, A., Jeanneau, L., Labasque, T., Le Henaff, G., Mérot, P., Molénat, J., Petitjean, P., Pierson-Wickmann, A.-C., Squividant, H., Viaud, V., Walter, C., and Gascuel-Odoux, C.: AgrHyS: An Observatory of Response Times in Agro-Hydro Systems, Vadose Zone Journal, 17, 1–16, https://doi.org/10.2136/vzj2018.04.0066, 2018.

Frings, P. J., Schubring, F., Oelze, M., and Von Blanckenburg, F.: Quantifying biotic and abiotic Si fluxes in the Critical Zone with Ge/Si ratios along a gradient of erosion rates, Am J Sci, 321, 1204–1245, https://doi.org/10.2475/08.2021.03, 2021.

Hodson, M. J., White, P. J., Mead, A., and Broadley, M. R.: Phylogenetic Variation in the Silicon Composition of Plants, Annals of Botany, 96, 1027–1046, https://doi.org/10.1093/aob/mci255, 2005.

Humbert, G., Jaffrezic, A., Fovet, O., Gruau, G., and Durand, P.: Dry-season length and runoff control annual variability in stream DOC dynamics in a small, shallow groundwater-dominated agricultural watershed, Water Resources Research, 51, 7860–7877, https://doi.org/10.1002/2015WR017336, 2015.

Jeanroy, E.: Diagnostic des formes du fer dans les pédogénèses tempérées, Ph.D. thesis, Université de Nancy, France, 47–49, 1983.

Kaiser, S., Wagner, S., Moschner, C., Funke, C., and Wiche, O.: Accumulation of germanium (Ge) in plant tissues of grasses is not solely driven by its incorporation in phytoliths, Biogeochemistry, 148, 49–68, https://doi.org/10.1007/s10533-020-00646-x, 2020.

Lugolobi, F., Kurtz, A. C., and Derry, L. A.: Germanium–silicon fractionation in a tropical, granitic weathering environment, Geochimica et Cosmochimica Acta, 74, 1294–1308, https://doi.org/10.1016/j.gca.2009.11.027, 2010.

Meek, K., Derry, L., Sparks, J., and Cathles, L.: 87Sr/86Sr, Ca/Sr, and Ge/Si ratios as tracers of solute sources and biogeochemical cycling at a temperate forested shale catchment, central Pennsylvania, USA, Chemical Geology, 445, 84–102, https://doi.org/10.1016/j.chemgeo.2016.04.026, 2016.

Mehra, O. P., and Jackson, M. L.: Iron oxide removal from soils and clays by a dithionite-citrate system buffered with sodium bicarbonate, in: Clays and Clay Minerals, Proc. 7th Natl. Conf., Washington, D.C., 1958, edited by: Swineford, A., Pergamon Press, New York, 317–327, 1960.

Molenat, J., Gascuel-Odoux, C., Ruiz, L., and Gruau, G.: Role of water table dynamics on stream nitrate export and concentration in agricultural headwater catchment (France), Journal of Hydrology, 348, 363–378, https://doi.org/10.1016/j.jhydrol.2007.10.005, 2008.

Rains, D. W., Epstein, E., Zasoski, R. J., and Aslam, M.: Active Silicon Uptake by Wheat, Plant Soil, 280, 223–228, https://doi.org/10.1007/s11104-005-3082-x, 2006.

Sparks, J. P., Chandra, S., Derry, L. A., Parthasarathy, M. V., Daugherty, C. S., and Griffin, R.: Subcellular localization of silicon and germanium in grass root and leaf tissues by SIMS: evidence for differential and active transport, Biogeochemistry, 104, 237–249, https://doi.org/10.1007/s10533-010-9498-2, 2011.

Tamm, O.: Um bestämning ow de organiska komponenterna i markens gelkomplex, Meddelanden Fran Statens Skogsförsöksanstalt, 19, 385–404, 1932.